

# Random matrix theory of the isospectral twirling

**Salvatore F. E. Oliviero**[1][*], **Lorenzo Leone**[1],
**Francesco Caravelli**[2] **and Alioscia Hamma**[1]

**1** Physics Department, University of Massachusetts Boston, 02125, USA
**2** Theoretical Division (T-4), Los Alamos National Laboratory, 87545, USA

[*] s.oliviero001@umb.edu

## Abstract

We present a systematic construction of probes into the dynamics of isospectral ensembles of Hamiltonians by the notion of Isospectral twirling, expanding the scopes and methods of ref. [1]. The relevant ensembles of Hamiltonians are those defined by salient spectral probability distributions. The Gaussian Unitary Ensembles (GUE) describes a class of quantum chaotic Hamiltonians, while spectra corresponding to the Poisson and Gaussian Diagonal Ensemble (GDE) describe non chaotic, integrable dynamics. We compute the Isospectral twirling of several classes of important quantities in the analysis of quantum many-body systems: Frame potentials, Loschmidt Echos, OTOCs, Entanglement, Tripartite mutual information, coherence, distance to equilibrium states, work in quantum batteries and extension to CP-maps. Moreover, we perform averages in these ensembles by random matrix theory and show how these quantities clearly separate chaotic quantum dynamics from non chaotic ones.



# 1  Introduction

Few concepts are more fascinating than the one expressed by the word *Chaos* ($\chi\acute{\alpha}o\varsigma$). Originally meaning 'the Abyss', but soon opposed to the notion of an ordered universe [2], the concept of chaos lives up to its etymology by being a very challenging notion to define precisely. In classical mechanics, chaotic systems are those displaying high sensitivity to the initial conditions. A related notion is that of the butterfly effect, which relates the amplification of small local perturbations to the spreading of a large effect involving the whole system.

In quantum unitary dynamics, different initial states cannot lead to exponentially separated trajectories, as an elementary result from the fact that the inner product between different states is conserved by unitary dynamics. To circumvent this elusiveness, quantum chaos has traditionally been associated to salient properties of those systems which were quantized from classical chaotic systems. Such systems were found to possess statistics of energy levels spacings corresponding to the predictions of Random Matrix Theory (RMT) [3–14], as well as statistics of eigenvectors [15–20]. Such a measure of chaos is impractical for a quantum many-body system, as it requires a perfect knowledge of the spectrum of the Hamiltonian, and is quite indirect. After all, we still would not know what quantum chaos means [21–28].

In recent years, more direct measures of chaos have been proposed. The one that is most closely related to the butterfly effect is the expectation value (in the thermal state) of commutators of spatially separated local operators. A quantum version of the butterfly effect means

that dynamics spreads operators around the system in a way that measuring one would have non trivial effect on the other [29]; this behavior in turns corresponds to the decay of four-point out-of-time order correlation functions (OTOCs) [30–52]. This behavior has been confirmed for several theories believed to be chaotic, for instance, theories holographically dual to gravity [53–56] or random unitary evolutions modelling the scrambling behavior of black holes [57–61], to non integrable quantum many-body systems [22,62–71]. The scrambling of information in a quantum many-body systems consists in the delocalization of quantum information by a quantum channel over the entire system in the sense that no information about the input can be gained by any local measurement of the output. It has been a very insightful and remarkable result [72] the realization that the butterfly effect as measured by rapid decay of the OTOCs implies the scrambling of quantum information. When different properties proposed to define a notion come to a unification, this means that one is onto something.

Quantum chaos is also invoked as an explanation of thermalizing behavior in closed quantum many-body systems [23,73–92] and the emergence of irreversibility [93] resulting from the establishment of a universal pattern of entanglement corresponding to that of a random state in the Hilbert space. The sensitivity of the dynamics can also be captured by the Loschmidt Echo (LE), and it has been shown that for bipartite systems LE and OTOCs coincide [94]. Entanglement dynamics is also invoked as an explanation for the onset of quantum chaos, from the aforementioned typicality of entanglement to complexity of the entanglement itself [95], and the fact that scrambling itself can be measured through quantum correlations between different parts of the system [48,72,96,97]. With a very insightful concept, the authors in [98] posit that the web of quantum chaos diagnostics should be unified and study further connections of these probes with complexity.

The need for a unifying framework is pressing, and if all the probes to quantum chaos can be unified in a single notion this would constitute a decisive step towards a theory of quantum chaos. The framework for unification has been introduced in [1], where it was shown that several probes of quantum chaos can be cast in the form of the $2k$-Isospectral twirling (IT). In this paper, we extend these results to obtain a more complete classification. Consider a unitary evolution $U = \exp(-iHt)$ generated by the Hamiltonian $H$. All the moments of polynomials of order $2k$ that depend on such evolution can be constructed starting from $U^{\otimes k,k} \equiv U^{\otimes k} \otimes U^{\dagger \otimes k}$. The Isospectral twirling consists in averaging over all the possible *U with a given spectrum*, that is, the Haar average of a $k$-fold unitary channel [99]. After the averaging, one obtains the probe as the expectation value in the thermal state of the IT with an appropriate permutation of the desired operator. After this average, this quantity only depends on the spectrum $\mathrm{Sp}(H)$ of the Hamiltonian generating the dynamics.

All the proposed probes to quantum chaos share a similar heuristics, that is, one has a desired characteristic that quantum chaos should have, say, the butterfly effect, and then defines a quantity that is able to characterize that effect. Moreover, one tries to show that physical systems thought to be chaotic do indeed possess that characteristic. In this very enlightening framework, though, one piece is missing. How does one know that these quantities and properties are not possessed also by systems that are far from being chaotic? The probability of return [100] was one of the first quantities conjectured to describe the high sensitivity of the dynamics. However, also integrable systems like diagonalizable quantum spin chains feature a rapid decay of such probability, see [101]. In the domain of quantum many-body physics, it has been shown that some OTOCs may indeed behave similarly also for such systems [49,102]. In order to complete the program of characterizing quantum chaos, one needs a tool that at the very least distinguishes clearly between chaotic and integrable systems. The Isospectral twirling does exactly this. Since it is a quantity that is completely characterized by the spectrum of the Hamiltonian generating the dynamics, one can evaluate it for Hamiltonians with spectra given by random matrix theory, e.g., the Gaussian Unitary Ensemble (GUE)

and for integrable Hamiltonians with spectra belonging to the ensembles Poisson or the Gaussian Diagonal Ensemble (GDE) and compare. If the probes behave distinctly in the different ensembles, then one has truly characterized quantum chaos. In [103] the OTOCs are averaged over the GUE. Comparison between different spectra requires the use of random matrix theory for all the ensembles of Hamiltonians.

In this paper, we show that random matrix theory applied to the Isospectral twirling of a unitary quantum channel shows that all the proposed probes to quantum chaos clearly separate dynamics generated by GUE Hamiltonians, that are chaotic, from integrable ones, that certainly are not chaotic. The distinction can be seen in the salient values and corresponding timescales of certain spectral functions that are averaged over the corresponding ensembles of isospectral Hamiltonians. These values and time scales clearly separate the ensemble of GUE Hamiltonians that are chaotic from the Poisson and GDE ensembles in the way they depend on the dimension $d$ of the Hilbert space. A complete table of the results is shown in Table 2. While the isospectral twirling has been introduced in [1], here we provide the calculations and techniques to obtain the random matrix theory results for several ensembles of spectra. Moreover, here we provide a concentration bound for the Isospectral twirling, generalize the notion of IT to general quantum channel, show the behavior of quantum coherence in quantum chaotic evolutions, include applications to quantum thermodynamics, and finally prove a general theorem regarding Schwartzian spectra distributions and universality at late times.

The paper is organized as follows: in Sec. 2 we introduce the steps involved in the calculation of the Isospectral twirling. In Sec. 3 we study the Isospectral twirling for the frame potential, OTOCs and Loschmidt echo, coherence and Wigner-Yanase-Dyson skew information, the mutual and tripartite mutual information, and quantum batteries. In Sec. 4 and the Appendices the details of the calculation of the Isospectral twirling and the random matrix theory is presented. Conclusions follow.

## 2 Definitions and strategy

### 2.1 General approach and notation

We now briefly describe the structure of this paper and provide a birdeye view to its scopes, techniques and results. The first goal of this paper is to show that many of the proposed probes to quantum chaos can be cast in the form of the expectation value of the Isospectral twirling times a suitable operator characterizing the probe. More specifically, we consider:

- measures of randomness and adherence to Haar measure, like the frame potential (see Sec. 3.1).

- measures of the butterfly effect as OTOC and Loschmidt Echos(LE) (in Sec. 3.2),

- measures of quantum correlations and direct measures of scrambling as the tripartite mutual information (in Sec. 3.3),

- measure of coherence (in Sec. 3.4),

- quantum thermodynamics measures of chaos (in Sec. 3.5).

We shall denote these measures by the notation $\mathcal{P}_{\mathcal{O}}(U)$. Here $\mathcal{P}$ is a linear functional of an operator $\mathcal{O}$ whose expectation value may be of interest as a probe to quantum chaos, and $U$ is the unitary evolution describing the dynamics of interest. We will show that, after averaging over all the possible dynamics with a *given spectrum*, all the probes to quantum chaos assume

the general form

$$\langle \mathcal{P}_{\mathcal{O}}(U) \rangle_G = \text{tr}\left( \tilde{T}_{\pi}^{(2k)} \mathcal{O} \hat{\mathcal{R}}^{(2k)}(U) \right). \tag{1}$$

The average $\langle \cdot \rangle_G$ is the average over all the unitary channels with a given spectrum. The quantity $\hat{\mathcal{R}}^{(2k)}(U)$ is the Isospectral twirling of $U$, that is, the average over its $2k$-fold channel [1, 104, 105]. The $\tilde{T}_{\pi}^{(2k)} = T_{\pi}^{(2k)}/\text{tr}(T_{\pi}^{(2k)})$ is a rescaled permutation operator corresponding to the element $\pi$ of the permutation group $S_{2k}$ (in the cycle representation, see App. 4.2 for details). More precisely, the $2k$-Isospectral twirling of a unitary channel is the average of $(G^{\dagger}UG)^{\otimes k,k}$ over $G$ sampled uniformly from the unitary group. Let us define it formally. Be $\mathcal{H} \simeq \mathbb{C}^d$ a $d$-dimensional Hilbert space and let $U \in \mathcal{U}(\mathcal{H})$ be a unitary quantum channel. The $2k$-Isospectral twirling of $U$ is defined as

$$\hat{\mathcal{R}}^{(2k)}(U) := \int dG \, G^{\dagger \otimes 2k} \left( U^{\otimes k,k} \right) G^{\otimes 2k}. \tag{2}$$

Above, $dG$ represents the Haar measure over the unitary group $\mathcal{U}(d)$ and we recall that

$$U^{\otimes k,k} \equiv U^{\otimes k} \otimes U^{\dagger \otimes k}, \tag{3}$$

which naturally arises from the linearization of the expectation values of polynomials of $U^{\dagger}XU$. The Haar average will be computed by the Weingarten functions method, discussed in Sec. 4 resulting in a weighted sum over permutation operators:

$$\hat{\mathcal{R}}^{(2k)}(U) = \sum_{\pi\sigma} (\Omega^{-1})_{\pi\sigma} \text{tr}(T_{\pi}^{(2k)} U^{\otimes k,k}) T_{\sigma}^{(2k)}, \tag{4}$$

where the matrix with components $\Omega_{\pi\sigma} = \text{tr}(T_{\pi}T_{\sigma})$ is the inverse of the Weingarten functions, see Sec. 4.1 for details.

The connection between the unitary channel and the dynamics is given by the Hamiltonian of the system, that is, $U = \exp(-iHt)$. Through this connection, the Isospectral twirling becomes a function of time $\hat{\mathcal{R}}^{(2k)}(t)$. Effectively, the Haar average performed by the Isospectral twirling corresponds to averaging over all the possible eigenstates of $H$. The result of the twirl only depends on the spectrum of $H$.

From Eq. (4) we see that the dependence of the Isospectral twirling on the spectrum of the unitary channel $U$ is stored in the functions known as *spectral form factors* [8]

$$(\text{tr} \, T_{\pi}^{(2k)})^{-1} c_{\pi}^{(2k)}(U) \equiv \tilde{c}_{\pi}^{(2k)}(U) := \text{tr}(\tilde{T}_{\pi}^{(2k)} U^{\otimes k,k}), \tag{5}$$

and therefore we have

$$\hat{\mathcal{R}}^{(2k)}(U) = \sum_{\pi\sigma} (\Omega^{-1})_{\pi\sigma} c_{\pi}^{(2k)}(U) T_{\sigma}^{(2k)}. \tag{6}$$

### 2.1.1 Probes to Chaos

Let us now show the connection between the typical probes of quantum chaos and Eq. (1). The definitions of the quantities on the left hand side are given in Sec. 3. For every quantity we show the Isospectral twirling in the form of Eq. (1) and its asymptotic evaluation for large dimension $d$. The temporal dependence is contained in the coefficients $c_{\pi}^{(2k)}(U)$ which also contain all the information about the spectrum of the Hamiltonian generating the dynamics. The large $d$ expression for some of the quantities above is too cumbersome to be reported in this section. We systematically compute and analyze all these quantities in Sec. 3.

The meaning of the above formulae is the following. We pick a random matrix within an ensemble of Hamiltonians defined by some spectral properties: for example, for a qubit system,

Table 1: Table of the Isospectral twirling for the different probes to chaos presented, and its large $d$ limit. Here $\mathscr{A} \in \mathcal{B}(\mathcal{H}^{\otimes 2})$, $\mathscr{A} = A^\dagger \otimes A$ is a bounded operator on $\mathcal{H}^{\otimes 2}$ and all the $T_\pi$ operators are some permutations; for a precise definition see the corresponding section.

| Probe | Isospectral twirling | Large $d$ limit |
|---|---|---|
| Frame potential $\hat{\mathcal{F}}^{(k)}_{\mathcal{E}_H}$ | $\text{tr}\,(T_{1\to 2}\hat{\mathcal{R}}^{\dagger(2k)}(U) \otimes \hat{\mathcal{R}}^{2k}(U))$ | $d^2\tilde{c}_4(t)+1$ |
| Loschmidt-Echo (1) $\mathcal{L}_1(t)$ | $\text{tr}\,(\hat{\mathcal{R}}^{(2)}(t)\psi^{\otimes 2})$ | $\tilde{c}_2(t)$ |
| Loschmidt-Echo (2) $\mathcal{L}(t)$ | $\text{tr}\,(\tilde{T}_{(14)(23)}\mathscr{A}^{\otimes 2}\hat{\mathcal{R}}^{(4)}(U))$ | $\tilde{c}_4(t)+d^{-2}$ |
| $\text{OTOC}_2(t)$ | $\text{tr}\,(\tilde{T}\mathscr{A}\hat{\mathcal{R}}^{(2)}(t))$ | $\tilde{c}_2(t)\|A\|^2_2$ |
| $\text{OTOC}_4(t)$ | $\text{tr}\,(\tilde{T}_{(1423)}(\mathscr{A}\otimes\mathscr{B})\hat{\mathcal{R}}^{(4)}(U))$ | $\tilde{c}_4(t)-d^{-2}$ |
| Entanglement $S_2(\psi_t)$ | $-\log\text{tr}\,\left(T_{(13)(24)}\hat{\mathcal{R}}^{(4)}(U)\psi^{\otimes 2}\otimes T_{(A)}\right)$ | $-\log\left[\frac{2}{\sqrt{d}}+\tilde{c}_4(t)\left(\text{tr}\,(\psi^2_A)-\frac{2}{\sqrt{d}}\right)\right]$ |
| TMI $I_{3_{(2)}}(t)$ | $\log d + \log\text{tr}\,(\tilde{T}_{(13)(24)}\hat{\mathcal{R}}^{(4)}(U)T^{\otimes 2}_{(C)})+$ $\log\text{tr}\,(\tilde{T}_{(13)(24)}\hat{\mathcal{R}}^{(4)}(U)T_{(C)}\otimes T_{(D)})$ | $\log_2(2-3\tilde{c}_4(t)+2\,\text{Re}\,\tilde{c}_3(t))+$ $\log_2(\tilde{c}_4(t)+(2-\tilde{c}_4(t))d^{-1})$ |
| Coherence $\mathcal{C}_B(\psi_U)$ | $1-\text{tr}\,\left(T_{(13)(24)}(\mathcal{D}_B\otimes\psi^{\otimes 2})\hat{\mathcal{R}}^{(4)}(U)\right)$ | $1-\tilde{c}_4(t)\text{tr}\,(D_B\psi)^2$ |
| Quantum work $\delta W(t)$ | $\text{tr}\,[T_2(\psi\otimes H_0)\hat{\mathcal{R}}^{(2)}(K)]$ | $\frac{d^2(1-\tilde{c}_2(t))}{d^2-1}\delta W_0$ |
| Free energy | $\beta^{-1}\log\text{tr}\,\left(T_{(13)(24)}\hat{\mathcal{R}}^{(4)}(U)(\psi^{\otimes 2}\otimes T_{(A)})\right)$ $+\text{tr}\,\left((H_A\otimes\psi)\hat{\mathcal{R}}^{(2)}(U)\right)$ | $\epsilon^{-1}\log\frac{(1+\tilde{c}_4(t)(d_A-1))}{d_A}+$ $\frac{d^2(1-\tilde{c}_2(t))}{d^2-1}$ |

we can consider all the Hamiltonians with the spectrum of a Pauli string. We ask: how will the dynamics generated by a Hamiltonian in this ensemble behave as described by quantities like mutual information, OTOCs, entanglement, coherence, etc. Knowing the spectrum of the Hamiltonian $H$, one can compute the coefficients $c^{(2k)}_\pi(U)$ that depend on the time $t$ through $U$. The dynamics of the probes is on average specified solely in terms of the functions $c^{(2k)}_\pi(t)$ in the Table 1. In the following, we will be interested by ensembles of Hamiltonians whose spectrum is fixed by some given probability distribution.

## 2.2 Spectral average of the Isospectral twirling: Random Matrix Theory

The second important goal of this paper is to show that the aforementioned probes do demonstrate *peculiar characteristics* of quantum chaos. To this end, we should show that a behavior that is possessed by an ensemble describing chaotic Hamiltonians is not possessed by an ensemble describing non-chaotic, e.g., integrable Hamiltonians. We can obtain this insight by looking at the average behavior of the probes $\langle \mathcal{P}_{\mathcal{O}}(U)\rangle_G$ within that ensemble and showing that that behavior is typical.

The Isospectral twirling is capable of being a tell-tale quantity for chaotic behavior because it only depends on the spectrum of the Hamiltonian, and we can define ensembles of chaotic or integrable Hamiltonians by spectral properties. For instance, for integrable systems typical energy distributions are known, the Gaussian Diagonal Ensemble (GDE) or the Poisson distribution (P). Similarly, for chaotic systems a typical spectrum is given by the Gaussian Unitary Ensemble (GUE) or the Gaussian Orthogonal Ensemble (GOE).

In the following we will denote by E the ensemble of Hamiltonians of interest and by

$\overline{R^{(2k)}(t)}^{E}$ the average behavior of the Isospectral twirling in that ensemble. In turn, this will become an average over the ensemble E for the spectral functions $c_{\pi}^{(2k)}(t)$. We will use random matrix theory for the ensembles E $\equiv$ GUE, P, GDE. These calculations are presented in detail in Sec. 4.3. The spectral functions $c_{\pi}^{(2k)}(t)$ depend on the permutation element. We will show they take the form

$$c_{\pi}^{(2k)}(U) = C_d (\operatorname{tr} U^{\alpha})^{\gamma} (\operatorname{tr} U^{\dagger \beta})^{\delta}, \tag{7}$$

where $\alpha, \beta, \gamma, \delta \in \mathbb{N}$ and depend on $k$ and $\pi$ and $C_d$ is a constant depending on the Hilbert space dimension $d$, see Table 3 and Table 4 for $c_{\pi}^{(2)}(U)$ and $c_{\pi}^{(4)}(U)$ respectively. Since we perform averages of these functions with respect to spectrum statistics of certain isospectral classes of Hamiltonians $H$, it is worth introducing some definitions. Let $H = \sum_i E_i |i\rangle\langle i|$ be a spectral decomposition of the Hamiltonian, with the assumption $E_{i+1} \ge E_i$. Throughout the paper we assume $H$ to be time-independent and use a decomposition of the form $U = \sum_j e^{-iE_j t} \Pi_j$, with $\Pi_j^2 = \Pi_j$ orthogonal projectors. The spectral functions Eq. (5) one obtains, up to $k = 2$ are the following:

$$
\begin{aligned}
c_2(t) &= \operatorname{tr}(U)\operatorname{tr}(U^{\dagger}) = \sum_{kl} e^{i(E_k - E_l)t}, \\
c_3(t) &= \operatorname{tr}(U^2)(\operatorname{tr} U^{\dagger})^2 = \sum_{kln} e^{i(2E_k - E_l - E_n)t}, \\
c_4(t) &= \operatorname{tr}(U)^2 \operatorname{tr}(U^{\dagger})^2 = \sum_{mnkl} e^{i(E_k + E_l - E_m - E_n)t}.
\end{aligned} \tag{8}
$$

In the above expression we have adopted a convenient notation for these functions which appear all around in the paper: $c_e^{(2)}(U) = c_2(t)$, $c_e^{(4)}(U) = c_4(t)$ and $c_{(12)}(U) \equiv c_3(t)$.[1] Since $c_{\pi}^{(2k)}(U)$ scale with the dimension of the Hilbert space, namely $c_{\pi}^{(2k)}(\mathbb{1}) = \operatorname{tr} T_{\pi}^{(2k)}$, it makes sense to introduce the tilded quantities for $a = 2, 3, 4$ defined above

$$\tilde{c}_a(t) = \frac{c_a(t)}{d^a} \le 1. \tag{9}$$

As the expressions Eq. (8) explicitly show, these coefficients are spectral functions of $U$; let us introduce the spectral average we use. Let E be an isospectral family of Hamiltonians $H$ with spectral distribution $P(\{E_k\}) \equiv P(E_1, \dots, E_d)$, the *spectral average* of the Isospectral twirling over E is defined as:

$$\overline{\hat{\mathcal{R}}^{(2k)}(U)}^{E} := \int d\{E_k\} P(\{E_k\}) \hat{\mathcal{R}}^{(2k)}(U), \tag{10}$$

where $d\{E_k\} \equiv dE_1 \cdots dE_d$; since the Isospectral twirling depends linearly from the coefficients Eq. (8), see Eq. (4), its spectral average with respect to E is a linear combination of the spectral average of $c_{\pi}(U)$s. One can immediately see that we can also express the coefficients in Eq. (8) in terms of (nearest) level spacings, $s_i \equiv E_{i+1} - E_i$; for this reason we also consider isospectral families E with given distribution of nearest level spacings $P(\{s_i\})$.

Notice that $\hat{\mathcal{R}}^{(2)}(U)$ depends only on $c_2(t)$, while $\hat{\mathcal{R}}^{(4)}(U)$ depends on $c_2(t)$, $c_3(t)$ and $c_4(t)$. As such, the higher the order of the Isospectral twirling, the higher the amount of information that it is contained and its ability to distinguish different types of dynamics cfr. Table 2 and Fig. 2. A technical but important definition that we use later is the following. The spectrum of a Hamiltonian Sp $(H)$ on $\mathcal{H} \simeq \mathbb{C}^d$ is said to be *generic* if for any $l$ s.t. $d \ge l \ge 1$:

$$\sum_{m=i}^{l} E_{n_i} - \sum_{j=1}^{l} E_{m_j} \ne 0, \tag{11}$$

unless $E_{n_i} = E_{m_j}$, $\forall i, j = 1, \dots, l$ and for some permutation of the indices $n_i, m_j$.

---

[1] Also, in some occasions also $c_2(2t)$ appears.

### 2.2.1 Eigenvalues distribution: GDE and GUE

We consider two types of eigenvalues distributions $P(\{E_k\})$, corresponding to the following classes of matrices: $\mathrm{GUE}(d, 0, 1/d^{1/2})$, random unitarily invariant $d \times d$ matrices whose entries have null mean and $d^{-1/2}$ as standard deviation and $\mathrm{GDE}(d, 0, 1/2)$ random diagonal matrices whose $d$ diagonal elements are stochastic independent variables with null mean and $1/2$ as standard deviation. The eigenvalue distributions read [8–10, 106]:

$$P_{\mathrm{GUE}}(\{E_i\}) \quad \propto \quad \exp\left\{-\frac{d}{2}\sum_i E_i^2\right\}\prod_{i<j}|E_i - E_j|^2, \tag{12}$$

$$P_{\mathrm{GDE}}(\{E_i\}) \quad \propto \quad \left(\frac{2}{\pi}\right)^{d/2}\exp\left\{-2\sum_i E_i^2\right\}, \tag{13}$$

the first distribution refers to a chaotic [9, 107], while the second is an integrable spectrum [108]. The difference between these eigenvalue distributions is the presence of level-repulsion term, namely $\prod_{i<j}|E_i - E_j|^2$. This term inhibits the presence of degeneracies and pops up also in the Wigner-Dyson distribution, as follows.

### 2.2.2 Nearest level spacings distributions

We have seen that the Isospectral twirling can be also regarded as a function of the nearest level spacings, $P_{\mathrm{E}}(\{s_i\})$. Because of the dependence of the Isospectral twirling for generic Hamiltonian on the level spacings, this paper focuses on performing averages with respect Poisson (P), Wigner-Dyson from the Gaussian Orthogonal Ensemble (WD-GOE), and Wigner-Dyson from Gaussian Unitary Ensemble (WD-GUE). The nearest level spacings distributions are given by [3, 8, 14, 109–113]:

$$\begin{aligned} P_{\mathrm{P}}(s) &= e^{-s} \quad \text{Poisson} \\ P_{\mathrm{WD\text{-}GOE}}(s) &= \frac{\pi}{2}s\,e^{-\frac{\pi}{4}s^2} \quad \text{Wigner-Dyson from GOE,} \\ P_{\mathrm{WD\text{-}GUE}}(s) &= \frac{32}{\pi^2}s^2\,e^{-\frac{4}{\pi}s^2} \quad \text{Wigner-Dyson from GUE,} \end{aligned} \tag{14}$$

which we use in the remainder of the paper, and whose distributions are shown Fig. 1. Although $P_{\mathrm{WD\text{-}GOE}}$ and $P_{\mathrm{WD\text{-}GUE}}$ are calculated for $d = 2$, they are seen to be a good approximation for $d \to \infty$ [8]. An important remark, which is cleared during the explicit calculations of random matrix theory (Sec. 4), is that for the distributions in Eq. (15) we are considering the nearest level spacings $s_i$ to be independent from each other; this assumption works well for the Poisson distribution.

### 2.2.3 Results and plots

In Sec. 4 we perform the computation of the spectral functions $\tilde{c}_a(t)$. Here we discuss the results and present some plots for the temporal evolution of these functions. Since the goal of this paper is to infer from the Isospectral twirling the dynamical properties of integrable vs. non-integrable spectra, the properties of $\tilde{c}_a(t)$ directly enter the Isospectral twirling. A striking feature of $\tilde{c}_a(t)$ is that both the magnitude of the oscillations with respect to the mean, the location of the first minimum and the typical timescale with which the asymptotic state is reached depend on the dimension $d$ of the Hilbert space. We present the salient features of the spectral function $\tilde{c}_2(t)$ and $\tilde{c}_4(t)$ in Table 2. One can observe that typically the GDE distribution reaches the asymptotic limit in a $\sqrt{\log d}$ time, while for the case of GUE and Poisson such timescale is a scaling law $d^\gamma$, with exponents $\gamma = 1$ and $\gamma = 1/2$ respectively.

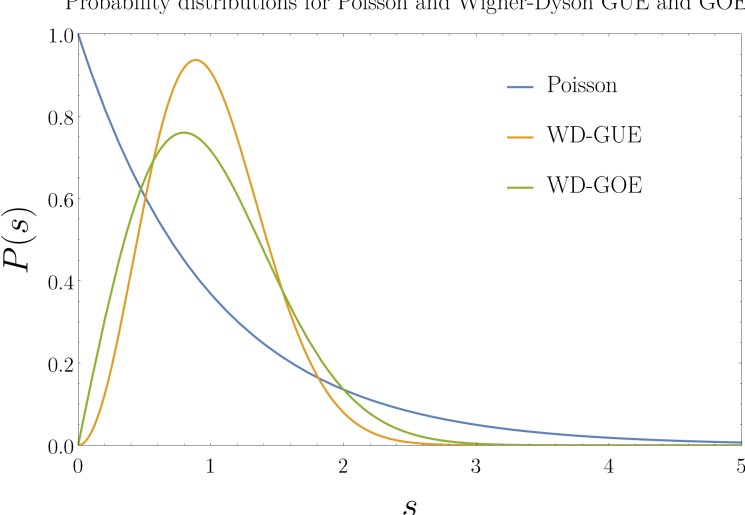

Figure 1: Plot of the nearest level spacing distributions Poisson, Wigner Dyson GUE and GOE reported in Eq. (15).

While we are able to distinguish different ensembles during the dynamics, the behavior in $t \to \infty$ is universal and does not depend on the ensemble (cfr. Table 2), see Proposition in Sec. 4.5. A detailed case by case analysis is presented below, while the spectral average of $c_2(t)$ and $c_4(t)$ are plotted in Fig. 2.

### 2.3 Typicality

In this section we show how typical is the behavior in the two ensembles of dynamical systems on which we perform the averages, namely, the typicality in the ensemble of isospectral Hamiltonians (that is, in the ensemble $\{G^\dagger H G, G \in \mathcal{U}(d)\}$, and in the ensemble of Hamiltonians with a given spectrum, namely E.

The Isospectral twirling is a tool to study quantum mechanical quantities depending only on the spectrum of the unitary generating the dynamic, and thus the details, such as the basis of the Hamiltonian, are washed away in the calculation [76, 83, 94, 114, 115]. As we show in this section, however, the Isospectral twirling exhibits typicality, e.g. a typical dynamical curve is close to the one obtained from the Isospectral twirling. In this paper, given the operator $\mathcal{P}_\mathcal{O}$, we analyze its Isospectral twirling $\langle \mathcal{P}_\mathcal{O} \rangle_G$ and then study the average behavior of such quantity over some well-known distribution of spectra and nearest level spacings distributions. Since we are taking two averages, there are two typicalities to talk about. With the use of Lévy lemma [116], we prove the typicality of the average with respect to the randomization of the eigenvectors, and from which the typicality of subsequent average with respect to the spectrum follows. Taking $\mathcal{P}_\mathcal{O}$ its Isospectral twirling can be written as $\langle \mathcal{P}_\mathcal{O} \rangle_G = \mathrm{tr}(\tilde{T}^{(2k)} \mathcal{O} \hat{\mathcal{R}}^{(2k)}(U))$. Then:

$$\Pr\left(|\mathcal{P}_\mathcal{O} - \langle \mathcal{P}_\mathcal{O} \rangle_G| \ge \delta\right) \le 4 \exp\left\{-\frac{d\delta^2 \mathrm{tr}^2(T^{(2k)})}{72k^2 \|\mathcal{O}\|_1^2 \pi^3}\right\}. \tag{15}$$

A proof of the upper bound above is provided in Sec. 4. It follows that if $O(\|\mathcal{O}\|_1) \le O(d^{1/2} \mathrm{tr}(T^{(2k)}))$, in the large $d$ limit the probability of deviating from the average drops to zero exponentially fast, and the Isospectral twirling shows strong typicality.

Now let us turn to the typicality within the ensembles E. We exploit the linearity of $\hat{\mathcal{R}}^{(2k)}(U)$ in the coefficients $\tilde{c}_\pi(U)$, specializing the discussion for the case $k = 1, 2$. As mentioned above,

Table 2: Timescales and values for $\overline{\tilde{c}_2(t)}^{\text{E}}$ (*Left*) and $\overline{\tilde{c}_4(t)}^{\text{E}}$ (*Right*) obtained by the time evolution $U = \exp(-iHt)$ for the ensembles of Hamiltonian: Poisson, GUE, GDE, while the last one is $U$ drawn from the Haar measure. The reported scaling values can be explicitly computed from the envelope curves of the spectral average of $\tilde{c}_a(t)$ for $a = 2, 4$. The details of the calculations for these quantities are presented in Sec. 4.

| | $\tilde{c}_2(t)$ | | $\tilde{c}_4(t)$ | |
| --- | --- | --- | --- | --- |
| Ensemble | Scaling | Time | Scaling | Time |
| **Poisson** | first minimum $O(d^{-1})$ | $O(1)$ | first minimum $O(d^{-2})$ | $O(1)$ |
| | envelope $O(d^{-1/2})$ | $O(d^{1/4})$ | envelope $O(d^{-1})$ | $O(d^{1/4})$ |
| | envelope $O(d^{-1})$ | $O(d^{1/2})$ | envelope $O(d^{-2})$ | $O(d^{1/2})$ |
| | equilibrium $d^{-1}$ | $O(d^{1/2})$ | equilibrium $(2d-1)d^{-3}$ | $O(d^{1/2})$ |
| **GUE** | first minimum $O(d^{-1})$ | $O(1)$ | first minimum $O(d^{-2})$ | $O(1)$ |
| | envelope $O(d^{-1/2})$ | $O(d^{1/6})$ | envelope $O(d^{-1})$ | $O(d^{1/6})$ |
| | envelope $O(d^{-1})$ | $O(d^{1/3})$ | envelope $O(d^{-2})$ | $O(d^{1/3})$ |
| | dip $O(d^{-3/2})$ | $O(d^{1/2})$ | envelope $O(d^{-3})$ | $O(d^{1/2})$ |
| | equilibrium $d^{-1}$ | $O(d)$ | equilibrium $(2d-1)d^{-3}$ | $O(d)$ |
| **GDE** | equilibrium $d^{-1}$ | $O(\sqrt{\log d})$ | equilibrium $(2d-1)d^{-3}$ | $O(\sqrt{\log d})$ |
| **Haar** | value $d^{-2}$ | N.A. | value $2d^{-4}$ | N.A. |

for $k = 1$, $\hat{\mathcal{R}}^{(2)}(U)$ depends on $c_2(t)$ only. In this case, we can make use of the Chebyshev's inequality:

$$\Pr\left(\left|\tilde{c}_2(t) - \overline{\tilde{c}_2(t)}^{\text{E}}\right| \geq \delta\right) \leq \frac{\overline{\tilde{c}_4(t)}^{\text{E}} - \overline{\tilde{c}_2(t)}^{\text{E}\,2}}{\delta^2}, \tag{16}$$

the above bound holds for any $t \geq 0$. From Eq. (8) it is clear that $\tilde{c}_4(t) = \tilde{c}_2^2(t)$. Taking for example the asymptotic values, recalling table 2, we obtain:

$$\Pr\left(\left|\tilde{c}_2(t \to \infty) - d^{-1}\right| \geq \delta\right) \leq \frac{d^{-2}}{\delta^2}, \tag{17}$$

therefore in the thermodynamic limit the probability can be set arbitrarily small. For $k = 2$, $\hat{\mathcal{R}}^{(4)}(U)$ depends upon $\tilde{c}_2, \tilde{c}_4, \tilde{c}_3$. Here we present the discussion for $\tilde{c}_4$, while the tipicality for $\tilde{c}_3$ can be discussed using similar techniques. For $\tilde{c}_4(t)$ we make use of the Bathia-Davis inequality [117] together with the Chebyshev's inequality. Since $\tilde{c}_4(t) \in [0, 1]$, recall Eq. (9), we have:

$$\Pr\left(\left|\tilde{c}_4(t) - \overline{\tilde{c}_4(t)}^{\text{E}}\right| \geq \delta\right) \leq \frac{\overline{\tilde{c}_4(t)}^{\text{E}}}{\delta^2}, \tag{18}$$

we used $\overline{\tilde{c}_4^2}^{\text{E}} - \overline{\tilde{c}_4}^{\text{E}\,2} \leq (1 - \overline{\tilde{c}_4}^{\text{E}})\overline{\tilde{c}_4}^{\text{E}} \leq \overline{\tilde{c}_4}^{\text{E}}$. This bound holding for any $t \geq 0$, e.g. taking $t \to \infty$:

$$\Pr\left(\left|\tilde{c}_4(t \to \infty) - (2d-1)d^{-3}\right| \geq \delta\right) \leq \frac{2d^{-2}}{\delta^2}. \tag{19}$$

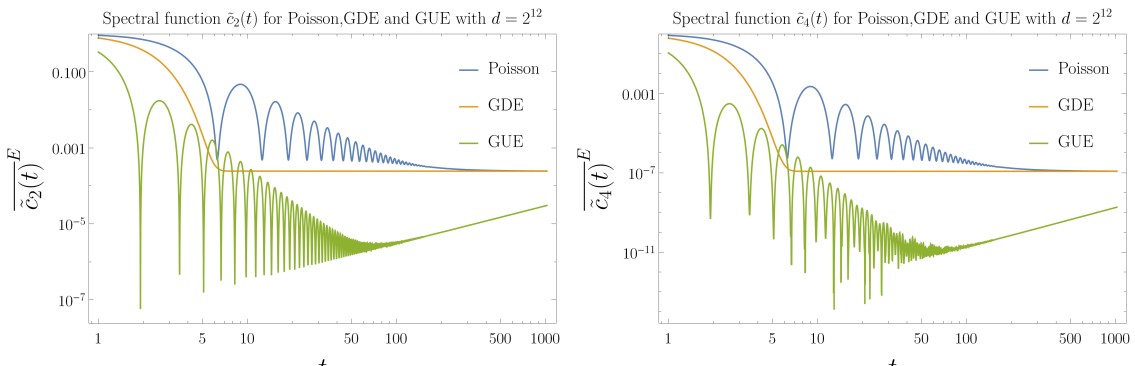

Figure 2: *Left*: Log-Log plot of the spectral function $\overline{\tilde{c}_2(t)}^E$ for different ensembles $E = P$, $E = GDE$ and $E = GUE$ for $d = 2^{12}$. The starting value is 1, while the asymptotic value is $d^{-1}$. For scalings see Table 2 (*Left*). *Right*: Log-Log plot of the spectral function $\overline{\tilde{c}_4(t)}^E$ for different ensemble $E = P$, $E = GDE$ and $E = GUE$ for $d = 2^{12}$. The starting value is 1, while the asymptotic value is $(2d - 1)d^{-3}$. From these plots and from scalings in Table 2(*Right*) we conclude that $c_4(t)$ distinguishes chaotic from non-chaotic dynamics more efficiently than $c_2$.

Once again, in the large $d$ limit we can set the probability to be arbitrary small. As a result of these calculations, the isospectral twirling presents typicality in both the averages over $G$ and over E.

## 3  Isospectral twirling as a universal quantity for quantum chaos

We have discussed in the Introduction how challenging is a direct characterization of quantum chaos. In classical systems, chaos can emerge when phase space trajectories diverge exponentially due to small differences in their initial states [118]. Such exponential growth leads to fast mixing in phase space, and thus the information on the initial state is quickly lost in time. In classical dynamical systems, chaotic behavior is thus a possible pathway towards ergodic mixing [119] and thermalization.

In quantum systems, the analog is not as obvious because of the linearity of the dynamics, which leads to unitary and seemingly information preserving evolution. Nonlinearity is an essential element of the theory of classical chaos, and thus there seems to be a certain tension between the two notions. However, the difference becomes thinner if the linear system is infinite dimensional. In quantum systems one can then intuitively see that in the case of many-body dynamics hints of rapid mixing between trajectories can occur. Such mixing in quantum systems is often referred to as *scrambling* [42, 59, 63, 120–125].

As we discussed earlier, many quantities have been proposed towards a diagnostic of quantum chaos [23, 88, 98, 126, 127], forming an intricate web [98] that necessitates a unifying framework.

A first probe to quantum chaos can be via the study of frame potentials, namely the adherence of an ensemble of unitary operators to the full unitary group. This quantity also serves as measure of randomness. We discuss the Isospectral twirling of the frame potential in Sec. 3.1.

A second possibility is to characterize quantum chaos through the growth of out-of-time-ordered correlation functions (OTOCs), thus analogous to the definition of diverging trajectories with the initial states [98]. OTOCs have various formulations, of which we discuss few in

this paper [38, 40, 41, 45, 50, 75, 128].

Loschmidt Echos have also been proposed to probe the high sensitivity of a quantum dynamics. Recently [1, 94], it has been understood that there is a direct connection between these quantities and the OTOCs. We study their Isospectral twirling in Sec. 3.2.

Entanglement and quantum correlations have been understood as important quantities for the description of complex quantum dynamics. Quantum chaotic dynamics is supposed to generate universal patterns of entanglement, on the one hand, leading to thermalization [57, 129], while, on the other, to high entanglement complexity and emergence of irreversibility [93]. Correlations, both quantum and classical, in a quantum many-body system are described by the mutual information. Moreover, a more complex form of correlations defined by the tripartite mutual information [72, 96] has been advocated as a measure of scrambling and thus of quantum chaos. These quantities are studied in Sec. 3.3 .

Recently, quantum coherence [130] has received a renewed interest as a resource theory [131, 132] for quantum algorithms and quantum thermodynamics. In this paper, we show that also quantum coherence can probe chaotic dynamics. Quantum coherence has been studied with a similar goal in the recent paper [133]. The Isospectral twirling of quantum coherence and skew information is shown in Sec. 3.4.

One of the most important motivations for the study of quantum chaos is quantum thermodynamics. We have already mentioned the aspects of thermalization and entropy production. Thermalization in closed quantum systems has been associated to the typicality of the entanglement generated by typical Hamiltonians [23, 73, 75, 76, 78, 79, 82, 83, 86, 87, 129]. More generally [74], even in presence of conserved quantities, equilibration in a closed quantum system can ensue in the form of typical expectation values for local observables. We show how this kind of approach to equilibrium tells apart chaotic from integrable Hamiltonians in Sec. 3.5.1. Moreover, an important topic in quantum thermodynamics is the understanding of quantum engines [129] and their capability of storing energy or performing work, e.g. quantum batteries [105]. These systems are discussed under the lens of the Isospectral twirling in Sec. 3.5.

Finally, one can consider more general quantum evolutions, like those described by generic CP-maps and quantum channels [134]. We show in Sec. 3.6 how one can take the Isospectral twirling of such channels.

For all these quantities, we show how to cast them in the form of Eq. (1). Then, applying the results from random matrix theory of the previous Sec. 2.2, we show how these quantities discriminate quantum chaotic from non chaotic dynamics.

## 3.1 Frame potential

Random unitary operators have been used in the literature to approximate quantum chaotic dynamics, for instance, in the context of black holes [135] to study the "scrambling" of information. An important question is how a given set of unitaries samples well the unitary group, in its capability of representing the group averages up to the $t$-th moment of the distribution. This forms the notion of unitary $t$-design [32, 128, 136–149]. Quantum chaotic dynamics is supposed to be able to realize high $t$-designs unlike a less ergodic, non-chaotic dynamics.

Given $\mathcal{E} = \{q_j, U_j\}$ an ensemble of unitary operators $U_j \in \mathcal{U}(\mathcal{H})$ weighted by $q_j$, the $k$-fold channel of $A \in \mathcal{B}(\mathcal{H}^{\otimes k})$ is defined as:

$$\Phi_{\mathcal{E}}(A) = \sum_j q_j U_j^{\dagger \otimes k} A U_j^{\otimes k}, \tag{20}$$

it is the weighted average of $A$ under the adjoint action of the $U_j$s. By definition, the ensemble

of unitaries $\mathcal{E}$ is a $k$-design iff the $k$-fold channel Eq. (20) equals the Haar $k$-fold channel:

$$\Phi_{\text{Haar}}(A) = \int dU U^{\dagger \otimes k} A U^{\otimes k}, \tag{21}$$

where $dU$ is the Haar measure over the unitary group. Recall that the Haar measure is the uniform measures over groups [150], see App. 4.1 for more details. In other words, a $k$-design reproduces k moments of the uniform distribution on $\mathcal{U}(\mathcal{H})$. A convenient characterization for $k$-designs is provided by the Frame potential $\mathcal{F}_{\mathcal{E}}$ of the ensemble $\mathcal{E}$, introduced in [136] and defined:

$$\mathcal{F}_{\mathcal{E}}^{(k)} = \sum_j \sum_k q_j q_k |\text{tr}\,(U_j U_k^{\dagger})|^{2k}. \tag{22}$$

This quantity measures the 2-norm distance between the $k$-fold channel of $\mathcal{E}$ Eq. (20) and the Haar $k$-fold channel Eq. (21). Scott [136] proved that the Frame potential is minimized, namely $\mathcal{F}_{\mathcal{E}}^{(k)} = k!$, iff $\mathcal{E}$ is a $k$-design. This makes the frame potential a natural measure of the randomness for ensembles $\mathcal{E}$.

Given one unitary $U$, we can construct the ensemble of all the unitaries with the same spectrum, namely $\{G^{\dagger}UG, G \in \mathcal{U}(d)\}$ and ask whether this ensemble realizes a unitary $t$-design. Since the unitaries are generated by a Hamiltonian, we can write $G^{\dagger}UG = \exp(-iG^{\dagger}HGt)$ and we can write the ensemble as

$$\mathcal{E}_H(t) = \{G^{\dagger} \exp(-iHt)G \,|\, G \in \mathcal{U}(\mathcal{H})\}. \tag{23}$$

We now compute the $k$-th frame potential of the ensemble $\mathcal{E}_H$ to quantify its randomness, i.e. to see how well $\mathcal{E}_H$ replicates the moments of the Haar distribution. The $k$-th frame potential of the ensemble $\mathcal{E}_H$ reads:

$$\mathcal{F}_{\mathcal{E}_H}^{(k)} = \int dG_1 \, dG_2 \left| \text{tr}\left(G_1^{\dagger}U^{\dagger}G_1 G_2^{\dagger}UG_2\right) \right|^{2k}. \tag{24}$$

As it turns out, the frame potential of $\mathcal{E}_H$ is the square Schatten 2-norm of the Isospectral twirling of Eq. (2) (see Proposition 1 in [1]):

$$\mathcal{F}_{\mathcal{E}_H}^{(k)} = \|\hat{\mathcal{R}}^{(2k)}(U)\|_2^2, \tag{25}$$

for completeness, a proof is reported in App. D.1. Recall that the computation of frame potential for the uniform Haar distribution returns $\mathcal{F}_{\text{Haar}}^{(k)} = k!$ and it provides a lower bound for Eq. (25), namely $k! \leq \mathcal{F}_{\mathcal{E}_H}^{(k)}$; this consideration, together with Eq. (25), implies that a lower value for the 2-norm of the Isospectral twirling Eq. (2) can be associated to higher randomness in the dynamics for $\mathcal{E}_H(t)$. One of the goal of this paper is to discriminate between chaotic and integrable Hamiltonians from spectral properties only; one of the key feature for the distinction comes from quantifying the randomness generated during the dynamics, i.e. as a function of $t$. A lower bound for the frame potential of $\mathcal{E}_H$ which turns useful as a measure of the deviation from the (Unitary) Haar average is the following (see Proposition 2 in [1]):

$$\mathcal{F}_{\mathcal{E}_H}^{(k)}(t) \geq d^{-2k} |\text{tr}\,(U)|^{4k}. \tag{26}$$

This bound holds for any $t \geq 0$, e.g. if one takes the infinite time average:

$$\mathbb{E}_T(\cdot) := \lim_{T \to \infty} T^{-1} \int_0^T (\cdot) dt \tag{27}$$

of the both sides of Eq. (26), it is possible to compute the lower bound for the asymptotic value of the frame potential, in the hypothesis of *generic spectrum* (see Sec. 2.2); from Proposition 3 in [1]:

$$\mathbb{E}_T\left[\mathcal{F}_{\mathcal{E}_H}^{(k)}(t)\right] \geq (2k)! + O(d^{-1}),\tag{28}$$

which is far from the Haar value $k!$. The non generic spectrum is defined in 2.2 and in practice means a particularly strong form of non resonance condition. The asymptotic value computed in Eq. 28 is the same for any Schwartzian spectral probability distribution, e.g. for GUE, GDE (cfr. Table 2), see Sec. 4.5; on the other hand, the spectral average of the frame potential $\mathcal{F}_{\mathcal{E}_H}^{(k)}$ is non trivial in its time evolution. For $k = 1$, we have :

$$\mathcal{F}_{\mathcal{E}_H}^{(1)} = \frac{d^2}{d^2-1}(d^2\tilde{c}_4(t) - 2\tilde{c}_2(t) + 1),\tag{29}$$

where the coefficients $\tilde{c}_a(t)$ do depend on the particular choice of the spectrum of $H$; they are defined in Eq. (8) and computed in Sec. 4. From [1] we report the plot of the spectral average of the frame potential Eq. (29) for GUE, GDE and Poisson, see Fig. 3. While for Poisson and GDE the frame potential has a similar behavior, these are quite distinct from the non-trivial dynamics in the GUE ensemble. The frame potential for Poisson and GDE never crosses the asymptotic value $3 + O(d^{-1})$, and it always stays away from the Haar value 1. This is consistent with Proposition 4 in [1], which states that

$$\overline{\mathcal{F}_{\mathcal{E}_H}^{(k)}}^{\text{GDE}}(t) \geq (2k)!,\tag{30}$$

i.e. the randomness in time of the GDE ensemble is always far from being equal to the Haar value; a proof of this result can be found in [1], here we present an alternative proof, see App. D. In the case of GUE instead, the frame potential reaches the Haar value 1, and remains close to it during the temporal interval $t \in [O(d^{1/3}), O(d)]$, only after reaching the asymptotic value 3. We conclude that, the chaotic spectra (GUE) present more randomness during the dynamics than the integrable ones (GDE, Poisson) which reach their maximum value of randomness asymptotically.

## 3.2 OTOCs and Loschmidt-Echos

In this section, we discuss two important classes of probes to chaos taking the form of correlation functions. In Sec. 3.2.1 we compute the Isospectral twirling of Loschmidt-Echos, while in Sec. 3.2.2 we compute the Isospectral twirling for the Correlators (OTOCs).

### 3.2.1 Loschmidt-Echo of the first and second kind

The Loschmidt Echo quantifies the non-trivial dynamics of a quantum system [100, 101, 151, 152], it can be used to study revivals occurring when an imperfect time-reversal procedure is applied to a complex quantum system [153]. The Loschmidt-Echo of the second kind is defined as the squared Hilbert-Schmidt scalar product between the unitary time evolution $e^{iHt}$ and the perturbed backward evolution $e^{-i(H+\delta H)t}$:

$$\mathcal{L}(t) = d^{-2}|\text{tr}(e^{iHt}e^{-i(H+\delta H)t})|^2.\tag{31}$$

It is interesting to use Hamiltonians $H$ and $\delta H$ such that the time evolution is non-trivial (for instance, chaotic), and thus the system is sensitive to the changes in initial conditions. In [94, 98], it was found that, under suitable conditions, the OTOC (which we discuss next) and LE are quantitatively equivalent. Via the Isospectral twirling it was shown in [1] that the

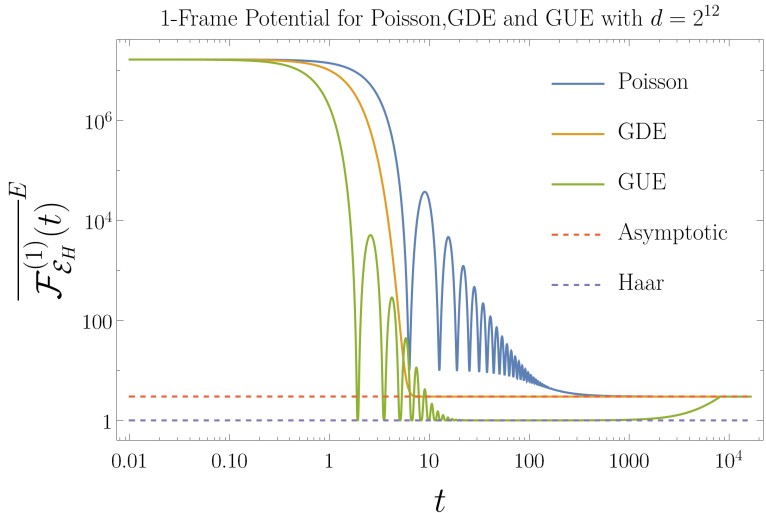

Figure 3: Log-Log plot of the ensemble average of frame potential for the Poisson, GUE and GDE ensemble with system size $d = 2^{12}$. The dashed lines represent the Haar value 1 and the asymptotic value $3 + O(d^{-1})$. While Poisson and GDE never go below the asymptotic value, GUE stays at the Haar value for the time window $t \in [O(d^{1/3}), O(d)]$ and at late times $t \geq O(d)$ GUE moves away from the Haar value [103] and reaches the asymptotic value. This plot is already shown in [1] and reported here for completeness.

OTOC and the LE are very similar measures. We comment on this issue in the next subsection when we discuss OTOCs.

For the Loschmidt Echo of the second kind, we now use the following connection to the Isospectral twirling (see Proposition 6 in [1]). Provided that $\text{Sp}(H) = \text{Sp}(H + \delta H)$ the Isospectral twirling of the LE is given by:

$$\langle \mathcal{L}(t) \rangle_G = \text{tr}\, (\tilde{T}_{(14)(23)} \mathscr{A}^{\otimes 2} \hat{\mathcal{R}}^{(4)}(U)), \tag{32}$$

where $\mathscr{A} = A^\dagger \otimes A$. We thus see that the 4-point Isospectral twirling enters directly into the Loschmidt echo as well. If one sets $\mathscr{A}$ to be a Pauli operator on qubits one gets in the large $d$ limit( see App. E):

$$\langle \mathcal{L}(t) \rangle_G = \tilde{c}_4(t) + d^{-2} + O(d^{-4}). \tag{33}$$

A plot of the Loschmidt-Echo of the second kind is in Fig. 4 (top).

A similar formula also applies to Loschmidt Echo of the first kind [101] $\mathcal{L}_1(t)$, which is defined as the modulus square the overlap between $|\psi\rangle$ and its time evolution $|\psi(t)\rangle \equiv U |\psi\rangle$, with $U \equiv \exp(-iHt)$. Defining $\psi = |\psi\rangle \langle\psi|$, it is straightforward to write it as:

$$\mathcal{L}_1(t) = |\text{tr}\, (U\psi)|^2. \tag{34}$$

We can now obtain the Isospectral twirling; with the usual trick $|\text{tr}\, (U\psi)|^2 = \text{tr}\, (U \otimes U^\dagger \psi^{\otimes 2})$, we obtain:

$$\langle \mathcal{L}_1(t) \rangle_G = \text{tr}\, (\hat{\mathcal{R}}^{(2)}(t) \psi^{\otimes 2}). \tag{35}$$

We thus see that a 2-point Isospectral twirling enters into the Loschmidt Echo of the first kind. If we insert in this formula the average, obtained in Eq. (106), we obtain

$$\langle \mathcal{L}_1(t) \rangle_G = \frac{d\tilde{c}_2 + 1}{d + 1}. \tag{36}$$

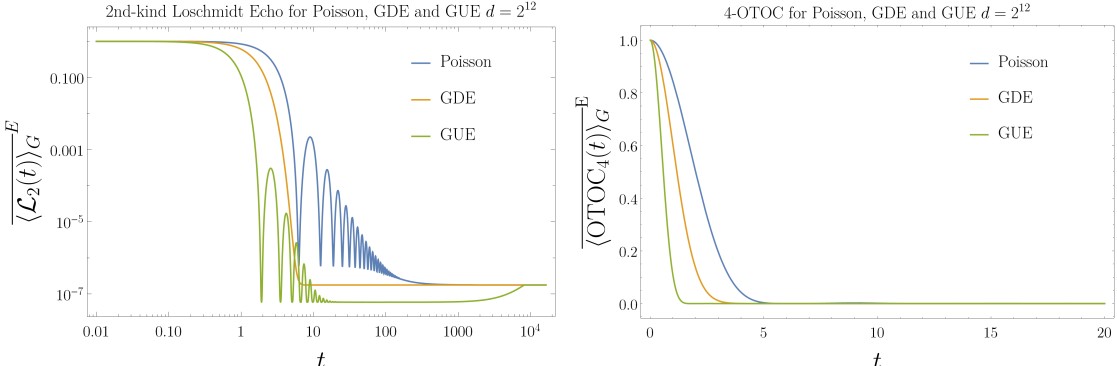

Figure 4: *Left*: Log-Log plot for the Loschmidt-Echo of the second kind reported in Eq. (36) for Poisson, GDE and GUE ensembles and $d = 2^{12}$. The Loschmidt-Echo for GDE and Poisson reaches the asymptotic value $3/d^2$ after a time $O(\sqrt{\log d})$ and $O(d^{1/2})$ respectively , GUE instead equals the offset $1/d^2$ for the temporal window $t \in [O(d^{1/3}), O(d)]$ and then reaches the same asymptotic value. *Right*: Plot of the 4-OTOC for Poisson, GUE, GDE ensemble for $d = 2^{12}$. The striking feature of the 4-OTOC is the following: while GUE drops to a negative value after a time $O(d^{1/3})$, GDE and Poisson never become negative and reach the asymptotic value $1/d^2$ in a time $O(\sqrt{\log d})$ and $O(d^{1/2})$ respectively.

We immediately note that $\mathcal{L}_1(t)$ depends on $\tilde{c}_2(t)$, while $\mathcal{L}(t)$ depends on $\tilde{c}_4(t)$ and therefore the Loschmidt-Echo of the second kind is more efficient in the distinction among chaotic and non-chaotic dynamics, cfr. Table 2. A plot of the quantity in Eq. ( 36) is shown in Fig. 5 (left); it is clear that we can distinguish chaotic from integrable time evolution.

### 3.2.2 Out-of-time-order correlators (OTOCs)

Among the many measures of information propagation in quantum many body physics, Out-of-Time-order Correlators (OTOC) have recently found many new applications, ranging from black holes to disordered spin systems [42]. In particular, the "Information Scrambling" can be measured either via the OTOC [21], which we discuss here, or the tripartite mutual information (TMI) [96] which we discuss later (however, the decay of OTOC with time implies the decay of the TMI [72, 96], and thus the two measures are related). In this section we show how the OTOCs are described by the Isospectral twirling. The definition of the $2k$-OTOC is the following. Let us consider $2k$ local, non-overlapping operators $A_l, B_l, l \in [1, k]$. Then, the $4k$-point OTOC is defined via the following expression:

$$\text{OTOC}_{4k}(t) := d^{-1}\text{tr}(A_1^\dagger(t)B_1^\dagger \cdots A_k^\dagger(t)B_k^\dagger A_1(t)B_1 \cdots A_k(t)B_k), \tag{37}$$

where operators evolve in the Heisenberg representation as $A_l(t) = e^{iHt}A_l e^{-iHt}$. Define $\mathscr{A}_l := A_l^\dagger \otimes A_l$ and similarly for $\mathscr{B}$. The Isospectral twirling enters the $4k$-point OTOC as (see Proposition 5 in [1]):

$$\left\langle \text{OTOC}_{4k}(t) \right\rangle_G = \text{tr}(\tilde{T}_\pi^{(4k)}(\otimes_{l=1}^k \mathscr{A}_l \otimes_{l=1}^k \mathscr{B}_l)\hat{\mathcal{R}}^{(4k)}(U)), \tag{38}$$

where $\tilde{T}^{(4k)}$ is a normalized permutation operator of $S_{4k}$ defined as $\tilde{T}_\pi^{(4k)} = S\tilde{T}_{1\,4k\cdots 2}S^\dagger$, with $S$ a permutation operator. For completeness we report the proof of Eq. (38) for $k = 1$ in App. E.1. For $k = 1$ we obtain the 4-point OTOC:

$$\left\langle \text{OTOC}_4(t) \right\rangle_G = \text{tr}(\tilde{T}_{(1423)}(\mathscr{A} \otimes \mathscr{B})\hat{\mathcal{R}}^{(4)}(U)). \tag{39}$$

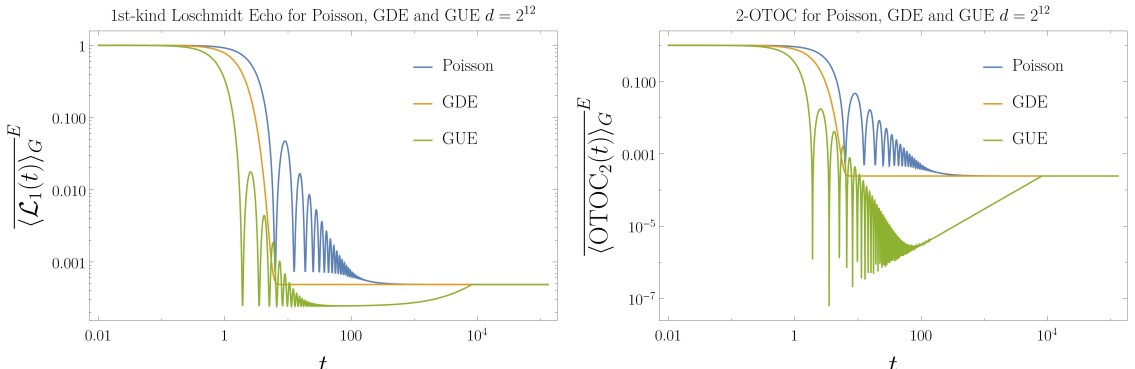

Figure 5: *Left*: Log Log plot for the Loschmidt-Echo of the first kind (see Eq. (36)) for Poisson, GDE and GUE ensemble and $d = 2^{12}$. Poisson and GDE being always greater than the asymptotic value $2/d$ they reach it in a time $O(d^{1/2})$ and $O(\sqrt{\log d})$, while GUE stays at the value $1/d$ for the whole temporal interval $t \in [O(d^{1/3}), O(d)]$ and then reaches the asymptotic value. *Right*: Log Log plot of the 2-OTOC see Eq. (43) for Poisson, GUE, GDE ensembles and $d = 2^{12}$ when $A$ is a Pauli operator. The behavior of the 2-OTOC is completely dictated by $\tilde{c}_2$: Poisson and GDE reaches the asymptotic value $1/d$ in a time $O(d^{1/2})$ and $O(\sqrt{\log d})$, while GUE in a time $O(d)$.

If one sets $A$ and $B$ to be non-overlapping Pauli operators on qubits, one finds in the large $d$ limit( see in App. E for details):

$$\left\langle \text{OTOC}_4(t) \right\rangle_G = \tilde{c}_4(t) - d^{-2} + O(d^{-4}). \tag{40}$$

As the expression above shows, the 4-point OTOCs distinguish chaotic from integrable behavior, through the timescales of $\tilde{c}_4(t)$ (see Table 2(*Right*)); we observe that the integrable distributions, GDE and Poisson, reach the asymptotic value $1/d^2$ in a time $O(\sqrt{\log d})$ and $O(d^{1/2})$ respectively; GUE instead reaches a negative value in a time $O(d^{1/2})$ and then the asymptotic value $1/d^2$ in a time of $O(1/d)$ ( see Fig. 4).

We now note that in this setting the LE and OTOC are equal to the spectral function $\tilde{c}_4(t)$ up to a constant $O(d^{-2})$. As a result, this implies that both the OTOC the LE are probing how scrambling the quantum system; in this sense, such an approach supports the findings of [103] on the quantitative agreement between OTOC and LE.

Another quantity of interest is the $\text{OTOC}_2(t)$. Let $A$ be a local operators on $\mathcal{B}(\mathcal{H})$, the 2-OTOC is defined as:

$$\text{OTOC}_2(t) = d^{-1} \text{tr}(A^\dagger(t)A), \tag{41}$$

where $A(t) = U^\dagger AU$, with $U \equiv \exp(-iHt)$. Taking the Isospectral twirling is straightforward to see:

$$\langle \text{OTOC}_2(t) \rangle_G = \text{tr}(\tilde{T} \mathscr{A} R^{(2)}(t)), \tag{42}$$

where we have defined $\mathscr{A} = A^\dagger \otimes A$. A straightforward application of Eq. (106) shows that:

$$\langle \text{OTOC}_2(t) \rangle_G = \frac{c_2(t) - 1}{d^2 - 1} \frac{\|A\|_2^2}{d} + \frac{d^2 - c_2(t)}{d^2 - 1} \frac{|\text{tr}(A)|^2}{d^2}. \tag{43}$$

The operator $A$ is just a general Hermitian operator. In particular we can consider the case that $A$ is a state. If we take $A \equiv \psi$ to be a pure state, we find that the 2-OTOC and the Loschmidt-Echo of the first kind are the same quantity. The plot of the 2-OTOC for Poisson, GDE and GUE are presented in Fig. 5 (*Right*) for $d = 2^{16}$. The convergence time can be observed to be different between the case of Poisson and GUE, and GDE.

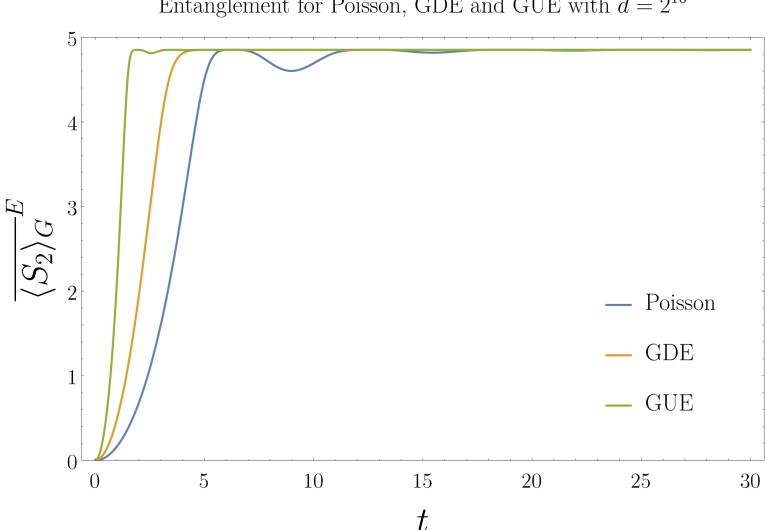

Figure 6: Plot of the r.h.s. of Eq. (45) with $\psi_A$ a pure state for Poisson, GDE and GUE ensembles and $d = 2^{16}$. GUE and Poisson reveal oscillations before reaching the equilibrium value in a time $O(d^{1/12})$ and $O(d^{1/8})$ respectively.

## 3.3 Entanglement and correlations

### 3.3.1 Entanglement

The Isospectral twirling also enters the evolution of entanglement [115, 154–166] assuming that the evolution can be well described by a random Hamiltonian with a given spectrum . The setup we consider is the following. The unitary time evolution of a state $\psi \in \mathcal{B}(\mathcal{H}_A \otimes \mathcal{H}_B)$ is given by $\psi \mapsto \psi_t \equiv U\psi U^\dagger$. Given a bipartition of the total Hilbert space $\mathcal{H} = \mathcal{H}_A \otimes \mathcal{H}_B$ of the Hilbert state, the entanglement of $\psi_t$ is computed by the 2-Rényi entropy $S_2 = -\log \operatorname{tr}(\psi_A(t)^2)$.

As proven in [1] (see *Proposition 7*, and App. F.1), the 2-Renyi entropy has the following structure in terms of the Isospectral twirling:

$$\langle S_2 \rangle_G \geq -\log \operatorname{tr}\left( T_{(13)(24)} \hat{\mathcal{R}}^{(4)}(U) \psi^{\otimes 2} \otimes T_{(A)} \right). \tag{44}$$

In the equation above, $T_{(A)} \equiv T_A \otimes \mathbb{1}_B^{\otimes 2}$, while $T_A$ is the swap operator on $\mathcal{H}_A$. If one sets $d_A = d_B = \sqrt{d}$, one gets, in the large $d$ limit, ( see the details in App. F.1):

$$\langle S_2 \rangle_G \geq -\log\left[ 2d^{-1/2} + \tilde{c}_4(t)\left( \operatorname{tr}(\psi_A^2) - 2d^{-1/2} \right) \right] + O(1/d). \tag{45}$$

The result we obtain using this technique is similar to the one obtained in [156]. It is thus clear that the time evolution of the entropy $\langle S_2 \rangle_G$ is dictated by $\tilde{c}_4(t)$. An analysis of the behavior of the lower bound can be obtained via the asymptotic analysis of the coefficient $\tilde{c}_4(t)$ (see Table 2), for clarity a plot of the lower bound is shown in Fig. 6, for $d = 2^{16}$. First, we note that fluctuations for GDE are suppressed, while the oscillations for Poisson and GUE are more pronounced; indeed the r.h.s. of Eq. (45) converges to the equilibrium value in a time $O(d^{1/12})$ and $O(d^{1/8})$ for GUE and Poisson respectively; these timescales can be explicitly calculated from the envelope curves of GUE and Poisson for $t = O(d^{1/2})$, see Sec. 4.4.2.

### 3.3.2 Mutual information

The mutual information (MI) of two random variables is a measure of the dependence of one variable on the other, and viceversa, and it quantifies how much information can be obtained from measuring a certain subsystem on its complementary [97, 134], e.g. a measure

of correlations. Given a certain joint distribution $p(x,y)$, the mutual information is defined as $\langle \log \frac{p(x,y)}{p(x)p(y)} \rangle$, and quantifies the level of factorization of a certain joint probability [167] with respect to two random variables $X$ and $Y$. Similarly, in quantum information theory, the quantum mutual information is a measure of correlation between subsystems and is defined via the von Neumann entropy $S(A) = \langle \log \psi \rangle_\psi$, where $\psi$ is the density matrix of a certain system $A$. It is the quantum mechanical analog of Shannon's mutual information, and is thus easily defined via the reduced density matrices of two complementary subsystems of the total Hilbert space. Specifically, let us thus consider a certain bipartition of the total Hilbert space $\mathcal{H} = \mathcal{H}_A \otimes \mathcal{H}_B$ and a state $\psi \in \mathcal{B}(\mathcal{H})$. The mutual information between the two partitions of the system $A$ and $B$ is quantified by the *mutual information*, defined as:

$$I(A : B) = S(A) + S(B) - S(AB),\qquad(46)$$

where $S(A)$ is a measure of quantum entropy of the reduced density matrix $\psi_A \equiv \mathrm{tr}_B(\psi)$, the same for $S(B)$. Instead of using von Neumann entropy, here we consider the 2-Rényi entropy $S_2(\psi)$ as a lower bound, for which we can perform the analysis analytically. We thus have:

$$I_2(A : B) = \log \mathrm{tr}\, \psi^2 - \log \mathrm{tr}\, \psi_A^2 - \log \mathrm{tr}\, \psi_B^2.\qquad(47)$$

If one sets $\psi = |\psi\rangle\langle\psi|$ to be a pure state, the mutual information reduces to the entropy $I(A : B)_\psi = 1 - 2S(A)_\psi = 1 - 2S(B)_\psi$. Therefore, taking a state $\psi$ with a given purity $\mathrm{tr}(\psi^2) = p$, let it evolve unitarily with a random unitary $U_G(t)$ with a given spectrum, from Eq. (44) we obtain the isospectral average:

$$\begin{aligned}\langle I(A : B)\rangle_G \geq \log p \quad &- \quad \log \mathrm{tr}\left(T_{(13)(24)}\hat{\mathcal{R}}^{(4)}(U)\psi^{\otimes 2} \otimes T_{(A)}\right)\\ &- \quad \log \mathrm{tr}\left(T_{(13)(24)}\hat{\mathcal{R}}^{(4)}(U)\psi^{\otimes 2} \otimes T_{(B)}\right).\end{aligned}\qquad(48)$$

The bound on the mutual information is similar to the lower bound on the entanglement introduced in Sec. 3.3.1, as for the entanglement the behavior of the bound on the mutual information is dictated by $\tilde{c}_4(t)$. If we consider a bipartite system $\mathcal{H} = \mathcal{H}_A \otimes \mathcal{B}$ with $d_A = \dim \mathcal{H}_A$ and $d_B = \dim \mathcal{H}_B$, we obtain that the ensemble average for GUE and Poisson of the lower bound reaches the asymptotic value $\log p + \log d_A + \log d_B$ in a time $O(d^{1/12})$ and $O(d^{1/8})$ respectively, these timescales can be calculated from the envelope of GUE and Poisson for $t = O(d^{1/2})$, see Sec. 4.4.2, while for GDE the same value is achieved in a time $O(\sqrt{\log d})$.

### 3.3.3 Tripartite mutual information.

Among the quantum measures of scrambling and chaotic behavior for many-body systems, the tripartite mutual information (TMI) is an information-theoretic measure that has been recently proposed in the literature [72, 96]. Consider a quantum channel that maps an input state in a bipartite system $\mathcal{H}_A \otimes \mathcal{H}_B$ into an output state $\mathcal{H}_C \otimes \mathcal{H}_D$. A good scrambling channel must be such that measurement on a local part of the output, say $C$, cannot detect any operation performed on a local part of the input, say $A$. For this reason, a good measure of scrambling is given by the tripartite mutual information [72] defined as

$$I_3(A : C : D) := I(A : C) + I(A : D) - I(A : CD),\qquad(49)$$

where $A, B$ and $C, D$ are the fixed bipartitions of past and future time slices respectively. To compute this quantity, one first maps the quantum channel into a state by the Choi isomorphism. Because of the unitarity of the evolution, the reduced density matrices $\rho_{AB}$ and $\rho_{CD}$ of the Choi state are maximally mixed, and thus $I(A : B) = I(C : D) = 0$. Then, since the tripartite mutual information is defined as a conditional mutual information, one necessarily

must have $I_3 \leq 0$, which is guaranteed thanks to the sub-additivity property of the entropy, whichever one decides to use. The unitary is thus minimally scrambling when $I_3 = 0$. While one can work with many measures of entropy, here we focus on the 2-Rényi entropy measure $I_{3_{(2)}}(U) = \log d + \log \operatorname{tr} \rho_{AC}^2 + \log \operatorname{tr} \rho_{AD}^2$ [1]. The tripartite mutual information has several properties, see App. F.2 for more details. More specifically, $\rho_{AC} = \operatorname{tr}_{BD}(\rho_U)$ and $\rho_{AD} = \operatorname{tr}_{BC}(\rho_U)$, where $\rho_U$ is the Choi state of the unitary evolution $U \equiv \exp(-iHt)$ and $H$ a random Hamiltonian with a given spectrum. Set $A = C$ and $B = D$, then, by defining $T_{(C)}^U := U^{\otimes 2} T_{(C)} U^{\dagger \otimes 2}$, $I_{3_{(2)}}$ can be written as [1]:

$$I_{3_{(2)}} = -3 \log d + \log \operatorname{tr}(T_{(C)}^U T_{(C)}) + \log \operatorname{tr}(T_{(C)}^U T_{(D)}). \tag{50}$$

The second term of Eq. (50) is reminiscent of the entanglement of quantum evolutions, which has been introduced in [156]. It should come at this point as no surprise that also TMI can be written in the form of an Isospectral twirling. In fact (following from Proposition 8 in [1]), one has that the Isospectral twirling of $I_{3_{(2)}}$ is upper bounded by

$$\left\langle I_{3_{(2)}} \right\rangle_G \leq \log d \quad + \quad \log \operatorname{tr}(\tilde{T}_{(13)(24)} \hat{\mathcal{R}}^{(4)}(U) T_{(C)}^{\otimes 2}) \tag{51}$$
$$+ \quad \log \operatorname{tr}(\tilde{T}_{(13)(24)} \hat{\mathcal{R}}^{(4)}(U) T_{(C)} \otimes T_{(D)}).$$

Given the fact that, as mentioned above, the TMI is a negative-definite quantity, the decay of the r.h.s. of Eq. (51) implies scrambling, that is, its value gets close to the minimum. In the large $d$ limit, Eq. (51) becomes (for details see App. F.2.3):

$$\left\langle I_{3_{(2)}}(t) \right\rangle_G \leq \quad \log_2(2 - 3\tilde{c}_4(t) + 2\operatorname{Re}\tilde{c}_3(t)) \tag{52}$$
$$+ \quad \log_2(\tilde{c}_4(t) + (2 - \tilde{c}_4(t))d^{-1}) + O(d^{-2}).$$

It follows that we can, also in this case, evaluate $\overline{\left\langle I_{3_{(2)}} \right\rangle_G}^E$ over the spectra GUE, GDE and Poisson. We set $d_C = \dim(\mathcal{H}_C)$ and $d_D = \dim \mathcal{H}_D$ and $d_C = d_D = \sqrt{d}$.

In general, the time evolution of $\overline{\left\langle I_{3_{(2)}} \right\rangle_G}^E$ depends on the spectral functions $\tilde{c}_a(t)$ which we have defined above, and calculated explicitly in Sec. 4. The results are however shown in Fig. 7, where we can quantitatively see how the chaotic and integrable behaviors are clearly different. Clearly, the time-behavior of $I_{3_{(2)}}(t)$ depend on the timescales of $\tilde{c}_4(t)$ Table. 2(*Right*). The equilibrium value of Eq. (52) can be calculated explicitly in our approach. We find that, for large $d$:

$$\lim_{t \to \infty} \overline{\left\langle I_{3_{(2)}}(t) \right\rangle_G}^E = 2 - \log_2 d + O(d^{-1}). \tag{53}$$

An interesting feature which can be observed in Fig. 7 is that the fluctuations on the mean value of the tripartite mutual information seem to decay with time. In order to quantify such decay, we fit the time scale of fluctuations (due to the oscillations around the mean) decay with a function of the form

$$t_{\text{fluct}} = a + b \log d, \tag{54}$$

obtaining the following values of $a, b$ in the case $d_C = d_D = \sqrt{d}$ for the GUE ensemble $a = -3.9, b = 0.8$, while for the Poisson distribution $a = -16.3, b = 3.2$. For GDE instead, there are no oscillations around the mean. It is possible to found similar behaviors in bipartitions of the system in which both $d_C$ and $d_D$ scales with the system size $d$, not necessarily being equals. When the system size of one of the partition is much greater than the other, the oscillations go to zero.

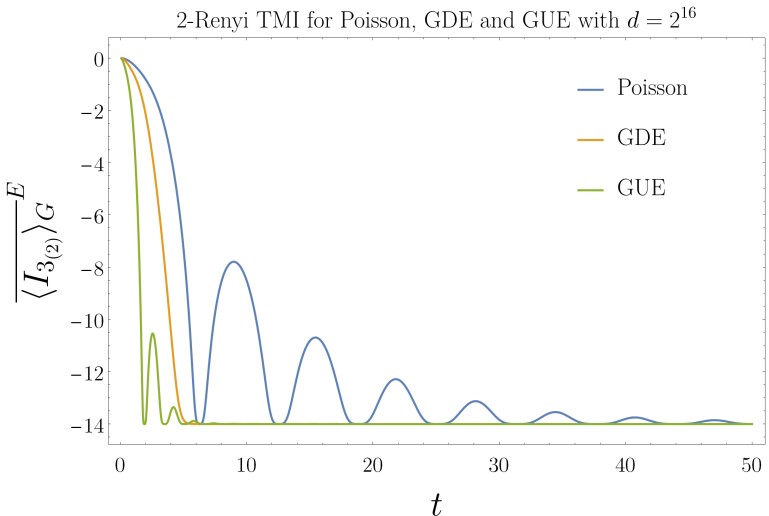

Figure 7: The upper bound for the tripartite mutual information, shown in Eq. (51) for the Poisson, GUE and GDE ensembles for $d = 2^{16}$. While GDE has no oscillations, GUE and Poisson exhibit oscillations which are reduced with the system size. An analysis of the oscillations is obtained in Eq. (53) and Eq. (54) respectively. This plot is already shown in [1] and reported here for completeness.

## 3.4 Coherence and Wigner-Yanase-Dyson skew information

### 3.4.1 Coherence

Another interesting measure to which the Isospectral twirling technique can be applied is quantum coherence [168–182]. Quantum coherence enters various aspects of the study of quantum mechanical systems, ranging from quantum chaos [133], to signature of quantum phase transitions [183] and as a resource in quantum thermodynamics [184]. Here we consider the Hilbert-Schmidt $l_2$ coherence, which even if it does not satisfy the monotones requirements [130] for a proper measure of coherence, it can be recast as an expectation value, i.e. it can be measured.

In order to show the connection between the Isospectral twirling and the Hilbert-Schmidt coherence, we define the dephasing superoperator $D_B(X) := \sum_i \Pi_i X \Pi_i$, where we introduced the spectral basis $B = \{\Pi_i\}$, with $\Pi_i$'s are rank-one projectors. This definition of coherence is natural, as it measures how the non-diagonal is an operator with respect to a certain basis defined by $\Pi$'s. We define $\psi_U = U\psi U^\dagger$ to be the state given a certain unitary transformation. Then, the $l_2$ measure of coherence in the basis $B$ is given by

$$\mathcal{C}_B(\psi_U) = \operatorname{tr} \psi_U^2 - \operatorname{tr}\left[(D_B\psi_U)^2\right]. \tag{55}$$

In the case in which the state $\psi$ is pure, then the first term is just one, otherwise, the quantity above is the purity of the initial state. Let us set $\operatorname{tr}(\psi_U^2) = 1$. The connection between the coherence and the Isospectral twirling of $\mathcal{C}_B$ has been derived in detail in App. G.1, and we have the following expression

$$\langle\mathcal{C}_B(\psi_U)\rangle_G = 1 - \operatorname{tr}\left(T_{(13)(24)}(\mathcal{D}_B \otimes \psi^{\otimes 2})\hat{\mathcal{R}}^{(4)}(U)\right), \tag{56}$$

where $\mathcal{D}_B \equiv \sum_i \Pi_i^{\otimes 2}$. It is interesting to see how the coherence depends on the dimension of the Hilbert space. The expression for the coherence for large $d$ takes the following asymptotic value:

$$\langle\mathcal{C}_B(\psi_U)\rangle_G = 1 - \tilde{c}_4(t)\operatorname{tr}\left[(D_B\psi)^2\right] + O(1/d). \tag{57}$$

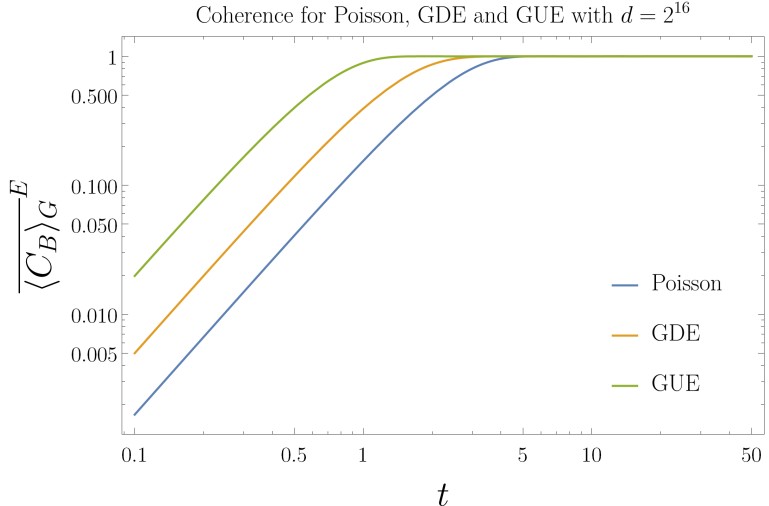

Figure 8: Log-Log plot of $\langle \mathcal{C}_B(\psi_U) \rangle_G$, with $\psi = \Pi_j$ for GUE, Poisson , GDE and $d = 2^{16}$. We do not observe any scaling difference between the ensembles, they equilibrate in a time $O(1)$.

As a function of time, a plot for $d = 2^{16}$ of the coherence is shown in Fig. 8 for the case of GUE, Poisson and GDE. We can see that while for $t \to \infty$ the asymptotic value approaches the maximum value 1 in all the three cases, the convergence to equilibrium is characterized by the behavior of $\tilde{c}_4(t)$; looking at Eq. (57), we note that the spectral coefficient $c_4(t)$ become negligible compared to 1 in a time $O(1)$ (cfr. Table 2), for this reason we do not observe any scaling difference between the three ensembles, see Fig. 8.

### 3.4.2 Wigner-Yanase-Dyson skew information

In this section, we consider also upper bounds to the Wigner-Yanase-Dyson (WYD) skew information, which was introduced in [185, 186]. The WYD has been historically studied in order to test how "hard" it is for an observable to be measured, but it has been used in a variety of contexts, including various generalizations of Heisenberg's uncertainty relations to mixed states [187]. The WYD is defined as

$$\mathcal{I}_\eta(\rho, X) = \text{tr}(X^2 \rho_t) - \text{tr}(X \rho_t^{1-\eta} X \rho_t^\eta). \tag{58}$$

It is easy to see that we can write the expression as

$$\mathcal{I}_\eta(\rho, X) = \text{tr}(X^2 U^\dagger \rho U) - \text{tr}(X U^\dagger \rho^{1-\eta} U X U^\dagger \rho^\eta U).$$

Taking the Isospectral twirling of such a quantity we obtain:

$$\left\langle \mathcal{I}_\eta(\rho, X) \right\rangle_G = \text{tr}\left(T_{(12)}(X^2 \otimes \rho)\hat{\mathcal{R}}^{(2)}(U)\right) - \text{tr}\left(T_{(1423)}(X^{\otimes 2} \otimes \rho^{1-\eta} \otimes \rho^\eta)\hat{\mathcal{R}}^{(4)}(U)\right). \tag{59}$$

The Isospectral twirling is evaluated in App. G.2, where we see that the Isospectral twirlings of order 2 and 4 enter.

Recently, it has been found [174] that the Wigner-Yanase-Dyson skew information can be set to be a good measure of quantum coherence. Let $B = \{\Pi_i\}$ be a basis of the Hilbert space $\mathcal{H}$, the coherence of $\rho$ can be measured as:

$$C(\rho) \equiv \sum_i \mathcal{I}_{1/2}(\rho, \Pi_i) = 1 - \text{tr}((D_B \sqrt{\rho})^2), \tag{60}$$

where $D_B(\cdot)$ is the dephasing super-operator on $B$. It is worth noting that for pure states $C(\rho) = \mathcal{C}_B(\rho)$ reduces to the previously defined $l_2$ measure of coherence. The Isospectral twirling of $C(\rho)$ follows immediately from the Isospectral twirling of $\mathcal{I}_\eta(\rho, X)$, see Eq. (59).

## 3.5 Quantum Thermodynamics

In this section we discuss some topics in quantum thermodynamics that result in a tell-tale of quantum chaotic behavior by means of the Isospectral twirling. First, we study the approach to equilibrium in a closed quantum system, following the setup of [104].

Then we turn our attention to quantum batteries. Among the quantum devices capable of performing work, quantum batteries are of central importance, also in view of the fact of the possible technological applications of nanoscale devices, and of possible quantum advantages in storing and extracting energy from the systems. Quantum batteries thus play a relevant role in the study of thermal machines and are an area of intense study [105, 188–195]. We apply the formalism of the Isospectral twirling to quantum batteries following the setup of [105] calculating average work and fluctuations. In the case of a closed quantum system, which we discuss in Sec. 3.5.2, the work is the amount of energy stored in the population levels defined of a certain Hamiltonian $H_0$. The population levels can be changed via the interactions, defined via a certain operator $V$. In Sec. 3.5.3 we discuss the framework of open systems, explicitly calculating the free energy in a given bipartition of the Hilbert space $\mathcal{H} = \mathcal{H}_A \otimes \mathcal{H}_B$ (and its convergence to equilibrium, following the setup of [104]). The application of the random matrix theory results of Sec. 2.2 to these quantities shows that the approach to equilibrium differs in chaotic and non-chaotic systems, and moreover that also the capability of storing and releasing energy in quantum batteries is affected by, and capable of discriminating, the quantum dynamics at play.

### 3.5.1 Convergence to Equilibrium

We are interested in how the Isospectral twirling enters the convergence to the equilibrium of a quantum many-body system. For this purpose, we follow the setup of [104], in which an equilibrium state $\omega$ in a bipartite quantum system $\mathcal{H} = \mathcal{H}_A \otimes \mathcal{H}_B$ is defined as $\omega = D_H(\psi)$, where $D_H(\cdot) = \sum_i \Pi_i(\cdot)\Pi_i$ is the dephasing super-operator in the Hamiltonian basis $\{\Pi_i\}_{i=1}^d$.

As it is well known, in a closed quantum system there is no equilibration at the level of the wave-function. What is possible is equilibration in probability for subsystems. In other words, local observables attain typical expectation values [76] or the reduced density matrix to a local subsystem has almost always a very small trace distance from some typical state [129]. Consider the reduced state $\psi_A(t) = \text{tr}_B \psi(t)$ of a global state evolving as $\psi(t) = U\psi U^\dagger$ and $U \equiv \exp(-iHt)$. We can compute the distance of the state of the subsystem $A$, $\psi_A(t)$ from the equilibrium reduced state $\omega_A = \text{tr}_B \omega$, through the Schatten 2-norm [116]:

$$\sqrt{f(t)} = \|\psi_A(t) - \omega_A\|_2. \tag{61}$$

It is preferable to study $f(t)$ the square of the 2-norm instead of $\sqrt{f(t)}$ to keep the calculations simple. The square of the norm can be written as:

$$f(t) = \text{tr}\left[ T_{(A)}\left( U^{\otimes 2}\psi^{\otimes 2}U^{\dagger\otimes 2} + D_H^{\otimes 2}\psi^{\otimes 2} - 2U\psi U^\dagger \otimes D_H\psi \right)\right]., \tag{62}$$

Since the Isospectral twirling randomizes over the eigenstates of $H$, leaving the spectrum invariant, we need to calculate the Isospectral twirling of the dephasing superoperator $D_H$; in Sec. 3.6 we introduce the Isospectral twirling for CP maps:

$$\hat{\mathcal{R}}^{(4)}(D_H) = \int dG\, G^{\dagger\otimes 4}(\mathcal{D}_H^{\otimes 2})G^{\otimes 4}, \tag{63}$$

where we define $\mathcal{D}_H \equiv \sum_i \Pi_i^{\otimes 2}$. In order to take the correct Isospectral twirling for $f(t)$, since we cannot consider the Isospectral twirling of $U^{\otimes 1,1}$ and the Isospectral twirling of $D_H$ to be independent, we need to define the following *hybrid* Isospectral twirling, which will be used *ad hoc* for the computation of $f(t)$:

$$\hat{\mathcal{R}}^{(4)}(U^{\otimes 1,1}, D_H) := \int \mathrm{d}G\, G^{\dagger \otimes 4}(U^{\otimes 1,1} \otimes \mathcal{D}_H)G^{\otimes 4}, \tag{64}$$

then we can take the Isospectral twirling of $f(t)$:

$$\langle f(t)\rangle_G = \mathrm{tr}\left(T_{(13)(24)}\hat{\Theta}\left(\psi^{\otimes 2} \otimes T_{(A)}\right)\right), \tag{65}$$

where we have defined:

$$\hat{\Theta} \equiv \hat{\mathcal{R}}^{(4)}(U) + T_{(23)}\hat{\mathcal{R}}^{(4)}(D_H)T_{(23)} - 2T_{(23)}\hat{\mathcal{R}}^{(4)}(U^{\otimes 1,1}, D_H)T_{(23)}. \tag{66}$$

From Eq. (166) and Eq. (65) we obtain the isospectral twirling for $f(t)$:

$$\begin{aligned}
\langle f(t)\rangle_G &= \frac{1}{d^2(d^2-1)(d+2)(d+3)d_A}(d_A-1)\Big[(d+2)\big((c_4(t)+2c_3(t) \\
&+ c_2(2t))(d-d_A)+(3+d)(1+d)d(d+d_A(d-1))\big) \\
&- 2c_2(t)\big(d^3(d_A+2)+d^2(3d_A+10)+d(6-4d_A)-6d_A\big)\Big].
\end{aligned} \tag{67}$$

The expression for large $d$ are presented below:

$$\langle f(t)\rangle_G \approx \tilde{c}_4(t)(1-\frac{1}{d_A})+\frac{1}{d_A d}(d_A^2-1)+O(1/d^2), \quad d_A = O(1) \tag{68}$$

$$\langle f(t)\rangle_G \approx \tilde{c}_4(t)+\frac{1}{\sqrt{d}}+O(1/d), \quad d_A = d_B = \sqrt{d}. \tag{69}$$

The behavior of the convergence to equilibrium, for $d_A = d_B = \sqrt{d}$ is governed by the coefficient $\tilde{c}_4(t)$ and its relationship with the offset $1/\sqrt{d}$: while GDE equilibrates in a time $O(\sqrt{\log d})$, as found in [104], GUE and Poisson equilibrate in a time $O(d^{1/12})$ and $O(d^{1/8})$ respectively, timescales that can be estimated from the envelope curves of GUE and Poisson for $t = O(d^{1/2})$, see Sec. 4.4.2. A plot is presented in Fig. 9. Equilibration in closed systems can also happen in integrable systems [76]. We see from our result that indeed the approach to equilibrium does not separate the behavior in GUE from that of Poisson. A similar analysis can be done for Eq. (68).

### 3.5.2 Quantum Batteries in Closed Systems

The model of quantum batteries we discuss in this section is that of a Hamiltonian $H_0$ which sets the scale of the energy via its quantum levels and of a quantum state $\rho$ evolving in time as $C_t(\rho) = \rho_t$ by means of a quantum evolution. The work on the battery, since the evolution is unitary and the entropy constant, results from populating the levels of $H_0$ in a different way from the initial state [196].

A quantum battery can be modeled by a Hamiltonian of the form $H = H_0 + V(t)$. Let $\psi_t = \mathcal{U}(\rho) = U\psi U^\dagger$ the state at the time $t$ obtained by unitary evolution $U$ induced by the Hamiltonian $H(t)$ [105]. The work extracted by (or stored in) the quantum battery is defined as the difference between expectation value of $H_0$ at time 0 and $t$:

$$W(t) = \mathrm{tr}(\psi H_0) - \mathrm{tr}(\psi U^\dagger H_0 U). \tag{70}$$

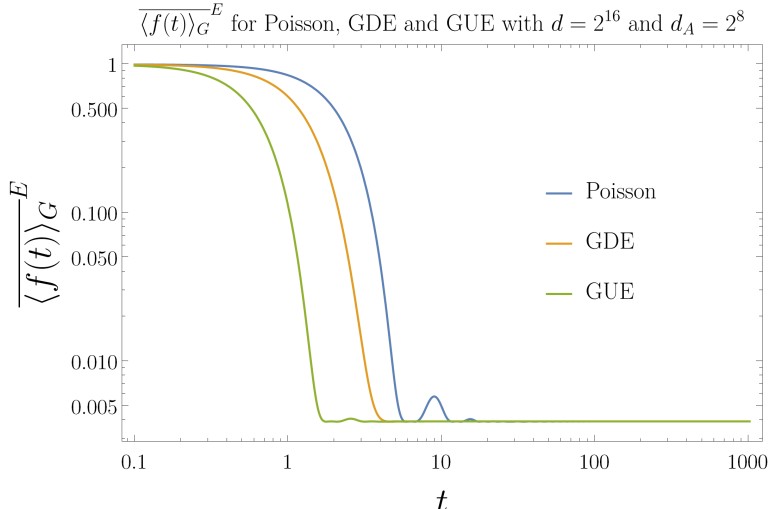

Figure 9: Log-Log plot of $\overline{\langle f(t)\rangle_G}^{\mathrm{E}}$, see Eq. (69), for $d = 2^{16}$ and $d_A = 2^8$ averaged over Poisson, GDE and GUE ensemble. The convergence to equilibrium is governed by $\tilde{c}_4$; we observe that GDE, GUE and Poisson equilibrate differently: in a time $O(\sqrt{\log d})$, $O(d^{1/12})$ and $O(d^{1/8})$ respectively (cfr. Table 2).

Following [105], one has that in the interaction picture the work can be written as

$$W(t) = \mathrm{tr}\,(\psi H_0) - \mathrm{tr}\,(\psi K^\dagger H_0 K), \tag{71}$$

where

$$K = \mathcal{T}\exp\left(-i\int_0^t V(s)ds\right). \tag{72}$$

We can compute an average work by averaging over isospectral unitary evolutions. We obtain (see the App. H.1 for details) the expression

$$\left\langle \tilde{W}(t)\right\rangle_G = \frac{\langle W(t)\rangle_G}{\delta W_0} = \frac{d^2 - c_2(t)}{d^2 - 1}, \tag{73}$$

where we have defined $\delta W_0 \equiv (E_0 - E_{HT})$, where $E_{HT} = \mathrm{tr}\,(H_0)/d$ and $E_0 = \mathrm{tr}\,(\psi H_0)$. The fluctuations of work are defined by $\left\langle \Delta W^2\right\rangle_G := \left\langle W^2(t)\right\rangle_G - \langle W(t)\rangle_G^2$ and are also related to $\hat{\mathcal{R}}^{(4)}(K)$ by

$$\left\langle \Delta W^2\right\rangle_G = \mathrm{tr}\left(T_{(13)(24)}\hat{\mathcal{R}}^4(K)(\psi^{\otimes 2}\otimes H_0^{\otimes 2})\right) - \langle W(t)\rangle_G^2. \tag{74}$$

Define the normalized work fluctuations as

$$\Delta\tilde{W}^2(t) := \Delta W^2(t)\left(\frac{\mathrm{tr}\,H_0^2}{d} - E_{HT}^2\right)^{-1}. \tag{75}$$

The normalized work fluctuations in the large $d$ limit are given by:

$$\Delta\tilde{W}^2(t) = h\tilde{c}_2(t) + \frac{1}{d} + O(d^{-2}), \tag{76}$$

where we have defined $h = 4E_0 E_{HT}\left(\frac{\mathrm{tr}\,H_0^2}{d} - E_{HT}^2\right)^{-1}$. See App. H.1. An analysis of the work can be done, similarly to what has been done previously, on the basis of the dependence on

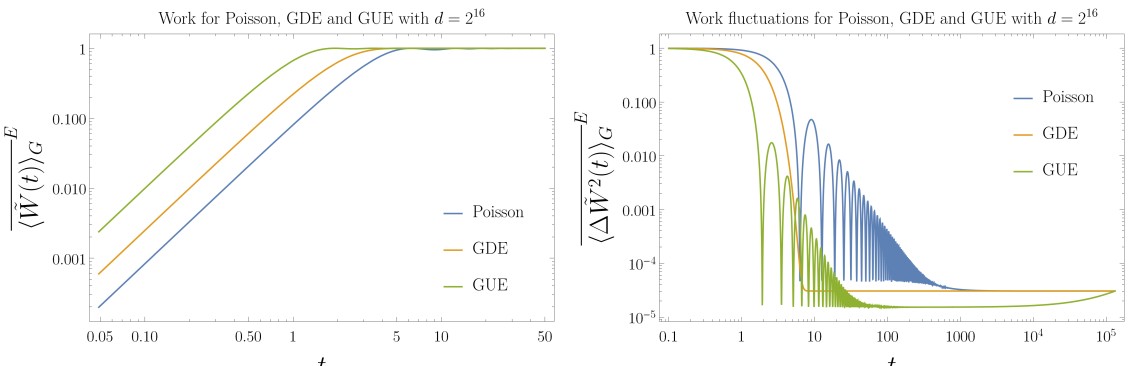

Figure 10: *Left:* Log-Log Plot of the normalized work in Eq. (73) for Poisson, GDE and GUE and $d = 2^{16}$. The normalized work starting from 0 at $t = 0$ and reaches the value 1 in a time $O(1)$ for the three ensembles. *Right:* Log-Log Plot of the quantity of Eq. (76) for Poisson, GDE, GUE ensembles, where $d = 2^{16}$ and $h = 1$. We observe that the timescales are dictated by the timescales of $\tilde{c}_2$, cfr. Table 2(*Left*).

the coefficient $c_2(t)$. In Fig. 10 we observe that the normalized work, defined as $\frac{W(t)}{\delta W_0}$ (and normalized between 0 and 1), reaches the asymptotic value 1 in a time $O(1)$; indeed, since the work depends on $c_2(t)$ we can see in Table 2 (left) that $c_2$ become negligible with respect to $d^2$ in a time $O(1)$. We conclude that the average work does not distinguish chaotic from integrable dynamics. Conversely, the fluctuactions around the mean, see Eq. (76), are able to discriminate between chaotic and integrable behaviors. Indeed while Poisson and GDE behave in a similar way: these are always greater than the asymptotic value $(h+1)/d$ and then drop after a typical time, $O(d^{1/2})$ for Poisson and $O(\sqrt{\log d})$ for GDE; GUE behaves differently: in time window $t \in [O(d^{1/3}), O(d)]$ the fluctuations sit on the value $1/d$, while only for typical times $t > O(d)$ it reaches the asymptotic value. More details in App. H.1.

### 3.5.3 Quantum Batteries in Open systems

An application for the Isospectral twirling is the study of quantum batteries in open systems. In the case of open systems storable and extractable energy does not correspond to extractable work [197], and thus in addition to the stored energy one needs also to consider the entropy [198], leading to the free energy. The free energy operator can be written as:

$$\hat{\mathcal{F}} := H_A + \beta^{-1} \log \psi_{U,A}, \tag{77}$$

defined on a Hilbert space $\mathcal{H} = \mathcal{H}_A \otimes \mathcal{H}_B$, where $\beta = k_B T$. The $A$ subsystem represents the system from which one extracts work, while the $B$ subsystem is the bath at temperature $T$. The evolution of this system is induced by a Hamiltonian of the form $H = H_A + H_B + H_{AB}$, where the interaction term is possibly time-dependent. The reduced evolution for the subsystem $A$ is not unitary and reads $\psi_{U,A} = \text{tr}_B(U\psi U^\dagger)$. The free-energy operator is the generalization of the definition of work in an open system. We aim to study the extractable work defined as the expectation value of the free-energy operator on the state $\psi_{U,A}$. It can be written as:

$$\mathcal{F} := \text{tr}(\hat{\mathcal{F}}\psi_{U,A}) = \text{tr}\left((\hat{\mathcal{F}} \otimes \mathbb{1}_B)U\psi U^\dagger\right). \tag{78}$$

It is possible to apply the Isospectral twirling to the extractable work. In order to average with the Isospectral twirling we should take the average of the Von-Neumann entropy term, for this we choose instead to use the hierarchy of Rényi entropies to bound the Von-Neumann entropy

$$S_2(\psi) \le S_1(\psi) \le \log \text{rank}\psi, \tag{79}$$

consequently follows a bound for the extractable work, then we use the Jensen inequality to average and we obtain:

$$
\begin{aligned}
\langle \mathcal{F} \rangle_G &\leq \operatorname{tr}\left((H_A \otimes \psi)\hat{\mathcal{R}}^{(2)}(U)\right) + \beta^{-1} \log \operatorname{tr}\left(T_{(13)(24)}\hat{\mathcal{R}}^{(4)}(U)(\psi^{\otimes 2} \otimes T_{(A)})\right) \\
\langle \mathcal{F} \rangle_G &\geq \operatorname{tr}\left((H_A \otimes \psi)\hat{\mathcal{R}}^{(2)}(U)\right) - \beta^{-1} \log d_A.
\end{aligned} \tag{80}
$$

Details of calculations are in App. H.2. We study a system, where $d_B = \dim \mathcal{H}_B = O(d)$ and $d_A = \dim \mathcal{H}_A = O(1)$ and choose as initial state $\psi = \psi_A \otimes \psi_B$ a pure state. In this set-up is preferable to study the following quantity:

$$
\Delta \langle \mathcal{F} \rangle_G := \frac{\langle \mathcal{F} \rangle_G(t) - \langle \mathcal{F} \rangle_G(0)}{\delta W_0}, \tag{81}
$$

where $\delta W_0 \equiv (E_0 - \operatorname{tr}(H_0)/d)$ has been defined in Sec. 3.5.2. Defining $\epsilon \equiv \delta W_0 \beta$, a dimensionless parameter depending on the temperature $T$ of the bat, we get in the large $d$ limit( see App. H.2):

$$
\Delta \langle \mathcal{F} \rangle_G \leq \frac{d^2(1 - \tilde{c}_2(t))}{d^2 - 1} + \epsilon^{-1} \log\left[1 + \tilde{c}_4(t)(d_A - 1)\right] - \epsilon^{-1} \log d_A + O(1/d) \tag{82}
$$

$$
\Delta \langle \mathcal{F} \rangle_G \geq \frac{d^2(1 - \tilde{c}_2(t))}{d^2 - 1} - \epsilon^{-1} \log d_A. \tag{83}
$$

We see that the lower bound of the extractable work depends on the $\tilde{c}_2$ coefficient, while the upper bound on $\tilde{c}_2$ and $\tilde{c}_4$. It is possible to observe that the two bounds differ for the term containing $\tilde{c}_4(t)$; this term become negligible after a time $O(1)$ for all the three ensembles (cfr. Table 2). It is interesting to point out that when this term equals zero, the lower and upper bound equals each other and the extractable work is exactly determined and it reads:

$$
\langle \mathcal{F} \rangle_G = 1 - \epsilon^{-1} \log d_A + O(d^{-1}). \tag{84}
$$

The normalized extractable work after the equilbration time $O(1)$ of $\overline{\tilde{c}_2}^{\,\mathrm{E}}$ and $\overline{\tilde{c}_4}^{\,\mathrm{E}}$, will depend only from the temperature of the bath $\epsilon^{-1} = \beta^{-1}/\delta W_0$. A plot of the normalized free energy $\Delta \langle \mathcal{F} \rangle_G$ for the three different ensemble is reported in Fig. 11.

## 3.6 Isospectral twirling and CP maps

In this subsection, we briefly discuss the application of the formalism of the Isospectral twirling to completely positive (CP) maps. Given $Q(\cdot) = \sum_\alpha K_\alpha(\cdot)K_\alpha^\dagger$ a quantum map, i.e. a completely positive trace preserving map, with $K_\alpha$ Kraus operators, the $2k$-Isospectral twirling is defined as:

$$
\hat{\mathcal{R}}^{(2k)}(Q) := \int dG\, G^{\dagger \otimes 2k} \mathcal{K}^{\otimes k} G^{\otimes 2k}, \tag{85}
$$

where we defined

$$
\mathcal{K} \equiv \sum_\alpha K_\alpha \otimes K_\alpha^\dagger, \tag{86}
$$

is a non hermitian bounded operator on $\mathcal{H}^{\otimes 2}$; it is a generalization of $U \otimes U^\dagger$ for non-unitary quantum maps. As a consequence of the trace preserving condition on $Q(\cdot)$, taking $T \in \mathcal{B}(\mathcal{H}^{\otimes 2})$ the swap operator, then one has $\operatorname{tr}(T\mathcal{K}) = d$. We compute the Isospectral twirling for quantum maps with the usual techniques of the Haar average, see Sec. 4.1, and get:

$$
\hat{\mathcal{R}}^{(2k)}(Q) = \sum_{\pi\sigma} (\Omega^{-1})_{\pi\sigma} \operatorname{tr}(T_\pi^{(2k)} \mathcal{K}^{\otimes k}) T_\sigma^{(2k)}, \tag{87}
$$

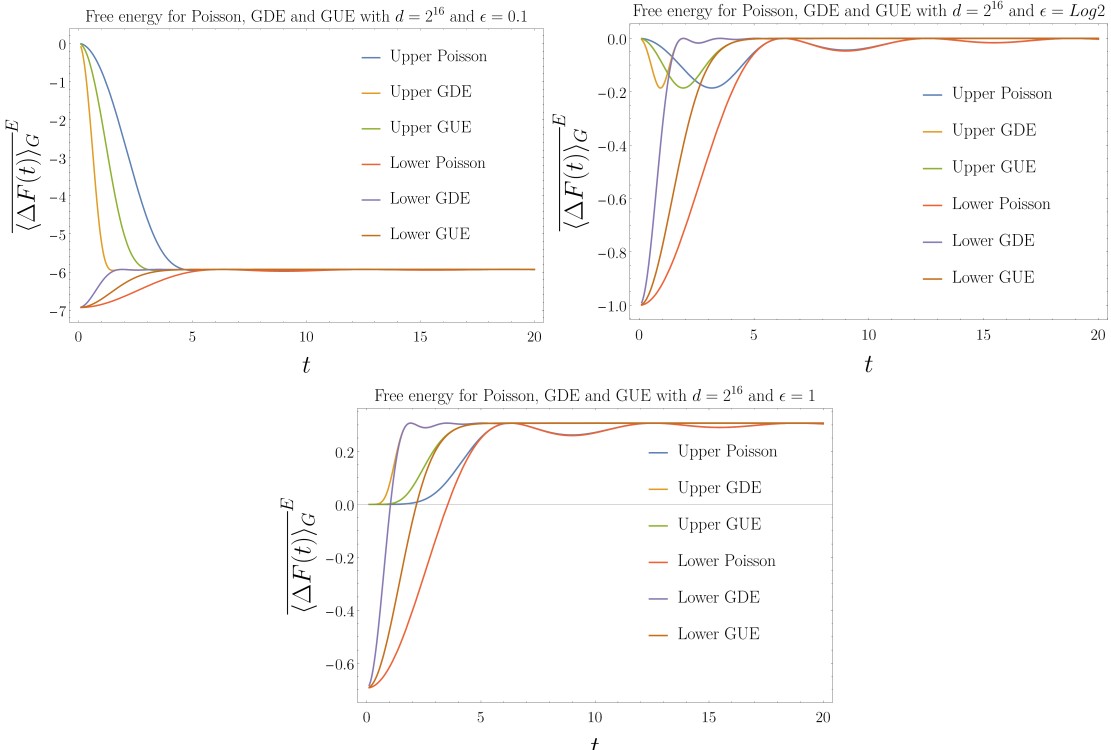

Figure 11: Upper and lower bounds for normalized extractable work $\overline{\langle \Delta \mathcal{F} \rangle_G}^{\mathrm{E}}$ for the Poisson, GUE and GDE ensemble for $d = 2^{16}$, $d_A = 2$ and $\epsilon = 0.1$ $(a)$, $\log 2$ $(b)$, $1$ $(c)$. There is a trade off for $\epsilon = \log d_A = \log 2$: the asymptotic extractable work becomes positive, see Eq. (84).

i.e. it is a linear combination of permutation operators weighted by coefficients depending on the spectral properties of the Kraus operators. The universality of the Isospectral twirling follows almost identically when it is applied to quantum maps. The 2-Isospectral twirling for quantum maps reads:

$$\hat{\mathcal{R}}^{(2)}(Q) = \frac{1}{d(d^2-1)}((d^2 - \operatorname{tr}\mathcal{K})T + d(\operatorname{tr}\mathcal{K} - 1), \tag{88}$$

see App. I for details. We see that it depends on the trace of $\mathcal{K}$ only; we will discuss the dephasing channel $D_B$ and explicitly compute the 2-Isospectral twirling for $D_B$, see Eq. (94). Let us first give two examples of applications of the above formalism: the Loschmidt-Echo of the first kind $\mathcal{L}_1$, defined in Sec. 3.2.1, and purity $\operatorname{Pur}(\psi) = \operatorname{tr}(\psi^2)$.

We define the Loschmidt-Echo of the first kind for quantum maps as:[2]

$$\mathcal{L}_1(Q) = \operatorname{tr}(\psi Q(\psi)) = \sum_\alpha \operatorname{tr}(\psi K_\alpha \psi K_\alpha^\dagger), \tag{89}$$

where $\psi \in \mathcal{B}(\mathcal{H})$ is a generic quantum state. Taking the Isospectral twirling we get:

$$\langle \mathcal{L}_1(Q) \rangle_G = \operatorname{tr}(T \hat{\mathcal{R}}^{(2)}(Q) \psi^{\otimes 2}), \tag{90}$$

see App. I; plugging the expression for the 2-Isospectral twirling Eq. (88) we have:

$$\langle \mathcal{L}_1(Q) \rangle_G = \frac{1}{d(d^2-1)}(d^2 - \operatorname{tr}\mathcal{K}) + d\operatorname{Pur}(\psi)(\operatorname{tr}\mathcal{K} - 1)). \tag{91}$$

---

[2]Note that if $Q \equiv U(\cdot)U^\dagger$, with $U \in \mathcal{U}(\mathcal{H})$, and $\psi$ is a pure state, the definition in Eq. (89) is identical to Eq. (34)

Now consider the purity a state $\rho = Q(\psi)$ with $\psi \in$ pure states:

$$\mathrm{Pur}(\rho) = \mathrm{tr}\,(Q(\psi)^2) = \sum_{\alpha,\beta} \mathrm{tr}\,(K_\alpha \psi K_\alpha^\dagger K_\beta \psi K_\beta^\dagger) \tag{92}$$

and taking the Isospectral twirling we get:

$$\langle \mathrm{Pur}(\rho)\rangle_G = \mathrm{tr}\,(T_{(12)(34)} \hat{\mathcal{R}}^{(4)}(Q)(\psi \otimes \mathbb{1})^{\otimes 2}), \tag{93}$$

for the derivation of the above formulas see App. I. Let us specialize a little bit our discussion and consider as quantum map the *completely dephasing channel*. Given $B = \{\Pi_i\}$ orthogonal projectors, the dephasing channel in its Kraus representation reads $D_B = \sum_i \Pi_i(\cdot)\Pi_i$, i.e. the Kraus operators are nothing but the orthogonal projectors $\Pi_i$s; recall the dephasing superoperator $\mathcal{D}_B$ has been used in App. 3.4. The Isospectral twirling of such a map randomizes the basis $B$ with respect to $D_B$ dephases. Noting that $\mathcal{D}_B = \sum_i \Pi_i^{\otimes 2}$, we have $\mathrm{tr}\,\mathcal{D}_B = d$; from Eq. (88) the 2-Isospectral twirling for $D_B$ is given by:

$$\hat{\mathcal{R}}^{(2)}(D_B) = \frac{(\mathbb{1}^{\otimes 2} + T)}{d+1}, \tag{94}$$

we see that it is proportional to the projector $\Pi^+ = (\mathbb{1}^{\otimes 2} + T)/2$ onto the symmetric subspace of $\mathcal{H}^{\otimes 2}$, defined in Eq. (108); one important remark is worth making here: the asymptotic limit $(t \to \infty)$ for $\hat{\mathcal{R}}^{(2)}(U)$ for a unitary channel $U$, given in Eq. (107), coincides with the Isospectral twirling applied on the dephasing channel, see Sec. 4. Plugging Eq. (94) in Eq. (90) the Isospectral twirling of the Loschmidt-Echo for the dephasing channel reads:

$$\langle \mathcal{L}_1(D_B)\rangle_G = \frac{1 + \mathrm{Pur}(\psi)}{d+1}, \tag{95}$$

the above expression makes sense: the Loschmidt-Echo is quantifying the probability of finding the state $D_B(\psi)$ in $\psi$; after the action of the dephasing $D_B$ on a random basis, the state becomes highly mixed; indeed, for the properties of the projectors of rank 1, we see that the purity in Eq. (92) for $Q \equiv D_B$:

$$\langle \mathrm{Pur}(D_B \psi)\rangle_G = \langle \mathcal{L}_1(D_B)\rangle_G = \mathrm{tr}\,(T \hat{\mathcal{R}}^{(2)}(D_B)\psi^{\otimes 2}) \tag{96}$$

is equivalent to the Loschmidt-Echo; $D_B$ requires the 2-Isospectral twirling instead of the 4-Isospectral twirling as the other quantum maps, see Eq. (92). We obtain the average coherence on a random basis $B$, see Sec. 3.4.1 for the definition:

$$\langle \mathcal{C}_B(\psi)\rangle = \mathrm{Pur}(\psi) - \frac{1 + \mathrm{Pur}(\psi)}{d+1}, \tag{97}$$

i.e. given a pure state $\psi$ the average coherence on a random basis is nearly maximal in the large $d$ limit. We remark that the formula Eq. (97) can be regarded as the average coherence of a random pure state $\psi$ on a fixed basis $B$, for this reason, an analog result can be found in [175, 180].

# 4 Isospectral twirling: techniques

This section is devoted to the description of all the techniques we used to derive the results in Sec. 3. It is organized as follows: Sec. 4.1 describes the Weingarten functions method to compute the Haar average, while Sec. 4.2 introduces the basics of the algebra of permutation

operators. In Sec. 4.3 and Sec. 4.4 we develop the random matrix theory techniques used in computing the spectral average for eigenvalue distributions and nearest level spacings distributions respectively. In Sec. 4.5 we prove the proposition which states that the asymptotic value of the spectral coefficients $c_\pi$s does not depend on the eigenvalue distribution. Finally, in Sec. 4.6 we make use of the Lévy lemma to infer about the typicality of the Isospectral twirling.

## 4.1 Haar average and Weingarten functions

The goal of this section is to calculate Eq. (2). The Haar average over the unitary channel can be performed via the following well known procedure [199–202], which we briefly review here for completeness. Let $A \in \mathcal{B}(\mathcal{H})$ a bounded operator on the Hilbert space $\mathcal{H}$ and $G \in \mathcal{U}(\mathcal{H})$ a unitary operator on $\mathcal{H}$ chosen uniformly at random. The Haar average of $A^{\otimes k}$ on the unitary group reads:

$$\left\langle A^{\otimes k} \right\rangle_G = \int dG \, G^{\dagger \otimes k} A^{\otimes k} G^{\otimes k}, \tag{98}$$

where $dG$ is the Haar measure over the unitary group; it has the following properties $\int dG = 1$ and $dG = d(GG') = d(G'G)$ for any $G' \in \mathcal{U}(\mathcal{H})$ unitary on $\mathcal{H}$; the latter is called left(right)-invariance of the Haar measure. The general formula to compute the Haar average is [32]:

$$\left\langle A^{\otimes k} \right\rangle_G = \sum_{\pi \sigma} W_g(\pi \sigma) \text{tr}\left(A^{\otimes k} T_\sigma\right) T_\pi, \tag{99}$$

where $\pi, \sigma$ are permutations of the permutation group $S_k$ while $W_g(\pi \sigma)$ are the Weingarten functions, and $\pi \sigma$ is the product of the permutation $\pi$ and $\sigma$, see Sec. 4.2 for details on permutation operators. The Weingarten functions [200] are defined as

$$W_g(\sigma) = \frac{1}{k!^2} \sum_\lambda \frac{\chi^\lambda(e)\chi^\lambda(\sigma)}{s_\lambda(1)}, \tag{100}$$

where the sum runs up to the number of conjugacy class of $S_k$. $\chi_\lambda$ is the irreducible character of $S_k$ associated with the conjugacy class $\lambda$ of $S_k$, $e$ is the identity element of the group. $s_\lambda(1) \equiv s_\lambda(1, \ldots, 1)$ is the Schur function evaluated on $d = \dim(\mathcal{H})$ elements. While the definition of the Weingarten functions is given above, we present an easy way to compute them originally introduced in [201]. Any permutation operator $T_\kappa \in S_k$ satisfies $[A^{\otimes k}, T_\kappa] = 0$, where $[\cdot, \cdot]$ is the commutator between operators of the algebra $\mathcal{B}(\mathcal{H}^{\otimes k})$ of the bounded operators on $\mathcal{H}^{\otimes k}$; therefore taking the average of the permutation operator $T_\kappa$, from Eq. (98) one has:

$$\left\langle T_\kappa \right\rangle_G = T_\kappa, \tag{101}$$

inserting the above condition in (99), we get:

$$T_\kappa = \sum_{\pi \sigma} W_g(\pi \sigma) \text{tr}\left(T_\kappa T_\sigma\right) T_\pi \iff \sum_\sigma W_g(\pi \sigma) \text{tr}\left(T_\kappa T_\sigma\right) = \delta_{\kappa \pi}, \tag{102}$$

therefore if we define $\Omega$ a $k! \times k!$ real symmetric matrix with components:

$$\Omega_{\pi \sigma} = \text{tr}\left(T_\pi T_\sigma\right). \tag{103}$$

We can simply express the Weingarten functions, treated as a matrix with components $W_g(\pi \sigma)$, as the inverse of $\Omega$, i.e:

$$W_g(\pi \sigma) = (\Omega^{-1})_{\pi \sigma}. \tag{104}$$

The above is a straightforward way to compute the Weingarten functions: we calculate traces of all the products $T_\pi T_\sigma$ and build the matrix $\Omega$, then we calculate the components of the inverse matrix $(\Omega^{-1})_{\pi \sigma}$.

### 4.1.1 Computation and properties of the Isospectral twirling

In this subsection, provided with tools and techniques of the Haar measure, we compute the Isospectral twirling in Eq. (2). Here we explicitly present the simplest case of Isospectral twirling, i.e $k = 1$, for $k = 2$ see App. A. For $k = 1$ we have only two permutation operators because the permutation group $S_2$ contains only two elements: the identity permutation $\mathbb{1}^{\otimes 2}$ and the swap operator $T$. The matrix $\Omega$, introduced in Eq. (103), and its inverse read:

$$\Omega = \begin{pmatrix} d^2 & d \\ d & d^2 \end{pmatrix}, \quad \Omega^{-1} = \frac{1}{d^2 - 1} \begin{pmatrix} 1 & d^{-1} \\ d^{-1} & 1 \end{pmatrix}, \tag{105}$$

since $T^2 = \mathbb{1}^{\otimes 2}$ the Weingarten functions read $W_g(e) = (d^2 - 1)^{-1}$ and $W_g(T) = d^{-1}(d^2 - 1)^{-1}$.

Table 3: Calculation of the spectral functions $c_\pi^{(2)}(U) = C_d (\operatorname{tr} U^\alpha)^\gamma (\operatorname{tr} U^{\dagger\beta})^\delta$, the corresponding powers $\alpha, \beta, \gamma, \delta$ and the constant $C_d$. See Eq. (8) for the short notation $c_2(t)$.

| $T_\pi^{(2)}$ | $c_\pi^{(2)}(U)$ | $\alpha$ | $\beta$ | $\gamma$ | $\delta$ | $C_d$ |
|---|---|---|---|---|---|---|
| $\mathbb{1}$ | $\operatorname{tr}(U)\operatorname{tr}(U^\dagger) \equiv c_2(t)$ | 1 | 1 | 1 | 1 | 1 |
| $T$ | $\operatorname{tr}(UU^\dagger) \equiv d$ | 1 | 1 | 0 | 0 | $d$ |

From Eq. (4) we obtain, see Table 3:

$$\hat{\mathcal{R}}^{(2)}(t) = \frac{c_2(t) - 1}{d^2 - 1} \mathbb{1}^{\otimes 2} + \frac{d^2 - c_2(t)}{d^2 - 1} \frac{T}{d} \tag{106}$$

we see that $\hat{\mathcal{R}}^{(2)}(t)$ depends only on $c_2(t)$, which we introduced in Eq. (8). $\hat{\mathcal{R}}^{(2)}(t)$ is a time-dependent hermitian operator; for $t = 0$ it equals the identity, while the asymptotic value $t \to \infty$ for the ensemble averages we consider reads (cfr. Table 2):

$$\lim_{t \to \infty} \overline{\hat{\mathcal{R}}^{(2)}(t)}^E = \frac{\mathbb{1}^{\otimes 2} + T}{d + 1}, \tag{107}$$

i.e. it is proportional to the projector $\Pi^+ = (\mathbb{1}^{\otimes 2} + T)/2$ onto the symmetric subspace $\mathcal{S}^+$ of $\mathcal{H}^{\otimes 2}$, defined as:

$$\mathcal{S}^+ := \operatorname{span}\left\{ \Pi^+ |\psi\rangle = |\psi\rangle \mid |\psi\rangle \in \mathcal{H} \right\}, \tag{108}$$

from Eq. (94) we can also write the following relation:

$$\lim_{t \to \infty} \overline{\hat{\mathcal{R}}^{(2)}(t)}^E = \hat{\mathcal{R}}^{(2)}(D_B), \tag{109}$$

where $\hat{\mathcal{R}}^{(2)}(D_B)$ is the Isospectral twirling applied to the dephasing superoperator, see Sec. 3.6 for the definition; the reason is twofold: first, it is easy to see that the infinite time average of $\mathbb{E}_T U^{\otimes 1,1} \propto D_B$ and we prove in Sec. 4.5 that the asymptotic value for the ensemble average of $\hat{\mathcal{R}}^{(2k)}$ coincides with its infinite time average: more precisely we prove that the infinite time average and the spectral average with Schwartzian probability distributions do commute. If one, instead of considering the Isospectral twirling of a unitary evolution $U(t)$ generated by an Hamiltonian $H$, compute the Haar average of the Isospectral twirling in Eq. (2) with respect to $U$, one obtains:

$$\left\langle \hat{\mathcal{R}}^{(2k)}(U) \right\rangle_U = \int dU \hat{\mathcal{R}}^{(2k)}(U) = \int dU U^{\otimes k} \otimes U^{\dagger \otimes k}, \tag{110}$$

where the last equality follows from the left-invariance of the Haar measure. The general result for Eq. (110) can be found in Corollary 3.21 of Ref. [203]:

$$\left\langle \hat{\mathcal{R}}^{(2k)}(U)\right\rangle_U = \sum_{\pi\sigma} W_g(\pi\sigma^{-1}) T_{\pi^{-1},\sigma}, \tag{111}$$

where $\pi,\sigma \in S_k$ and $T_{\pi^{-1},\sigma}\left|i_1,\ldots,i_k,j_1,\ldots,j_k\right\rangle = \left|j_{\sigma(1)},\ldots,j_{\sigma(k)},i_{\pi^{-1}(1)},\ldots,i_{\pi^{-1}(k)}\right\rangle$. For $k=1$, this expression reduces to:

$$\left\langle \hat{\mathcal{R}}^{(2)}(U)\right\rangle_U = \frac{T}{d}, \tag{112}$$

where $T$ is the swap operators. Looking at Eq. (106) we see that the closer $c_2(t)$ gets to 1, the closer the Isospectral twirling for $k=1$ is to its Haar average value.

## 4.2 Algebra of the permutation operators

For the sake of clarity, here we would like to briefly introduce the cycle representation of the permutation operators and their algebra used ubiquitously in the text.

First, the permutation operator $T_\pi$ corresponding to the permutation $\pi \in S_k$ reads:

$$T_\pi = \sum_{i_1,\ldots,i_k} \left|i_{\pi(1)},\ldots,i_{\pi(k)}\right\rangle\left\langle i_1,\ldots,i_k\right|. \tag{113}$$

These are generalization of the swap operator. Since the permutation operator acts on the right to left rather than left to right. In this paper, we are mostly concerned with permutation elements in the groups $S_2$ and $S_4$. Since we are using the cycle representation of the permutation group, if the element is (23) in the group $S_4$, clearly we are using the standard shorthand notation for (1)(23)(4), meaning that only the channels 2 and 3 are being swapped. The permutation operators thus follow the multiplication rules of the permutation group. For in the permutation group $S_4$, (12)(23) = (123), and thus $T_{(12)}T_{(23)} = T_{(123)}$. An intuitive and graphical representation of how the permutation operators multiply is shown in Fig. 12 (left). The permutation operator can be represented as a series of interchanging lines. For instance $T_{(12)}$ can be represented as the interchange of the first and second channel. Thus if we combine $T_{(12)}$ and $T_{(23)}$, their product results is an interchange between the first and third channel, and the second and the first and the third and the second.

This approach can also be used to calculate traces. Since we also have traces of permutation operators in $S_k$ combined with operators in $\mathcal{H}^{\otimes k}$, the traces can also be performed graphically. The graphical permutations of the channels are then combined with boxes, representing the action of operators on the channel. A trace is then represented as a closed sequence of lines that connects the final and initial channel in the same position. For instance $\text{tr}\left(T_{(324)}(A\otimes B\otimes C\otimes D)\right) = \text{tr}(A)\text{tr}(BCD)$, is shown in Fig. 12 (right). The global trace connects the $n$th line of the input to the $n$th line of the output. In the example above, one then has a continuous line in which one can read the line as a trace of $A$ multiplying the trace of $B$, $C$ and $D$. Two properties which we use a lot throughout the paper are the following; let $T_\pi, T_\sigma \in S_k$ two permutation operators with $\pi$ be a generic permutation and $\sigma = (1\,k\,k-1\,\ldots 2)$; let $A_i \in \mathcal{B}(\mathcal{H})$ with $i=1,\ldots,k$ be bounded operators on $\mathcal{H}$. Then:

$$T_\pi^\dagger\left(\bigotimes_i A_i\right)T_\pi = \bigotimes_{\pi(i)} A_i, \tag{114}$$

$$\text{tr}\left(T_\sigma \bigotimes_i A_i\right) = \text{tr}\left(\prod_i A_i\right), \tag{115}$$

the proofs are given in App. C.

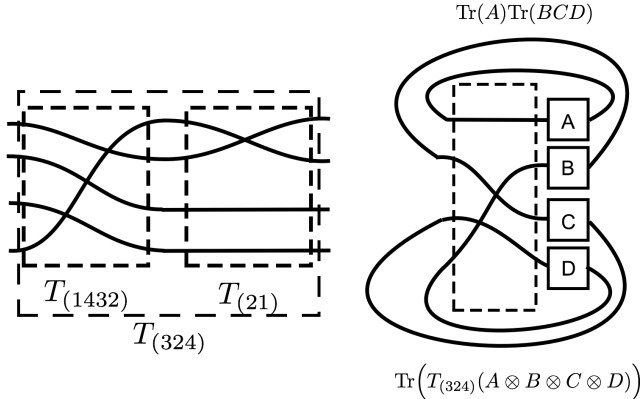

**Figure 12:** Graphical representation of the swap operator algebra and how the traces work. As a notational note, since the swap operator is defined as $T_\sigma = \sum_{ijkl} |\sigma(i)\sigma(j)\sigma(k)\sigma(l)\rangle\langle ijkl|$, the permutation group element in the cycle representation is intended to work from the right to the left.

### 4.3 Spectral average over eigenvalues distributions: GDE and GUE

In this section, we perform the spectral average of the spectral functions $c_a(t)$, $a = 2, 3, 4$, in the two ensemble of random matrices, GUE and GDE. Let us report Eq. (12) and Eq. (13) for convenience:

$$
\begin{aligned}
P_{\text{GUE}}(\{E_i\}) &\propto \exp\left\{-\frac{d}{2}\sum_i E_i^2\right\}\prod_{i<j}|E_i - E_j|^2, \qquad (116)\\
P_{\text{GDE}}(\{E_i\}) &\propto \exp\left\{-2\sum_i E_i^2\right\},
\end{aligned}
$$

the calculations for GUE were previously performed in [103, 204] and we discuss the technique here, see Sec. 4.3.1, for completeness. We compute the spectral average over the GDE distribution in Sec. 4.3.2; similar results can be found in [104].

#### 4.3.1 GUE

In this section, we present the calculation of the spectral average of $c_2(t)$ for GUE Hamiltonians, while the calculation for $c_a(t)$ for $a = 3, 4$ are reserved to App. B. By definition Eq. (10) the spectral average of the coefficient $c_2(t)$ reads:

$$
\begin{aligned}
\overline{c_2(t)}^{\text{GUE}} &= \sum_{i,j=1}^{d}\int dE_1\cdots dE_d\, e^{i(E_i - E_j)t} P_{\text{GUE}}(\{E_k\})\\
&= \sum_{i,j=1}^{d}\int dE_i\, dE_j e^{i(E_i - E_j)t}\int d\{E_{/ij}\} P_{\text{GUE}}(\{E_k\}) \qquad (117)\\
&= d + d(d-1)\int dE_1\, dE_2 e^{i(E_1 - E_2)t}\rho_{\text{GUE}}^{(2)}(E_1, E_2),
\end{aligned}
$$

where $d\{E_{/ij}\} \equiv dE_1\cdots dE_{i-1}\,dE_{i+1}\cdots dE_{j-1}\,dE_{j+1}\cdots dE_d$. First of all, note that since $E_i$, $E_j$ are dummy variables, the sum reduces to the counting $i \neq j$, while for $i = j$ the integral sum

to 1 thanks to the normalization of the probability distribution. In the third equality in Eq. (117) we have the 2-point marginal probability distribution. The $n$-point marginal probability distribution is defined as:

$$\rho_{\text{GUE}}^{(n)}(E_1,\dots,E_n) = \int dE_{n+1}\cdots dE_d \, P_{\text{GUE}}(\{E_k\}),\tag{118}$$

it can be expressed in terms of the determinant of the kernel $K$ [8–10]:

$$\rho_{\text{GUE}}^{(n)}(E_1,\dots,E_n) = \frac{(d-n)!}{d!}\det[K].\tag{119}$$

$K$ is a $n \times n$ matrix with the following matrix elements in the large $d$ limit:

$$K_{ij} = \delta_{ij}\frac{d}{2\pi}\sqrt{4-E_i^2} + (1-\delta_{ij})\frac{\sin\left(d\left(E_i-E_j\right)\right)}{\pi\left(E_i-E_j\right)},\tag{120}$$

the matrix elements $K_{ij}$ for $i = j$ are the well known Wigner semicircle law [3]. The calculation of $c_2(t)$ requires the 2-point correlation function:

$$\rho_{\text{GUE}}^{(2)}(E_1,E_2) = \frac{1}{d(d-1)}\left(\frac{d^2}{4\pi^2}\rho(E_1)\rho(E_2) - \frac{\sin^2[d(E_1-E_2)]}{\pi(E_1-E_2)^2}\right),\tag{121}$$

where $\rho(x) = (2\pi)^{-1}\sqrt{4-x^2}$. Substituting Eq. (121) in Eq. (117) one can finally complete the calculation, using the *box approximation* [103] to normalize the divergence and get the final result:

$$\overline{c_2(t)}^{\text{GUE}} = d + d^2\left(\frac{J_1(2t)}{t}\right)^2 + \theta(2d-t)\left(d-\frac{t}{2}\right),\tag{122}$$

where the $\theta$-function is defined as $\theta(x) = 1$ for $x \geq 0$, $\theta(x) = 0$ for $x < 0$ and $J_1(x)$ is the Bessel function of the first kind. A plot of $\overline{c_2(t)}^{\text{GUE}}$ is reported in Fig. 13(*Left*), while the scalings can be found in Table 2.

### 4.3.2 GDE

In this section we calculate the spectral average of $c_2(t)$ for GDE Hamiltonians. From Eq. (13) we immediately see that the probability distribution $P_{\text{GDE}}$ factorizes, i.e

$$P_{\text{GDE}}(\{E_k\}) = \prod_{i=1}^{d} p_{\text{GDE}}(E_i), \quad p_{\text{GDE}}(x) = \left(\frac{2}{\pi}\right)^{1/2}e^{-2x^2},\tag{123}$$

therefore the calculations for the spectral averages reduce to combination of one dimensional Fourier transforms of Gaussians; the marginal probability for the GDE distribution is trivial since the eigenvalues are uncorrelated from each other. Let us compute the spectral average for $c_2(t)$:

$$\begin{aligned}\overline{c_2(t)}^{\text{GDE}} &= \sum_{i,j=1}^{d}\int d\{E_k\}P_{\text{GDE}}(\{E_k\}) = d + (d-1)\left(\left(\frac{2}{\pi}\right)^{1/2}\int dx\, e^{-2x^2}\right)^2 \\ &= d + d(d-1)e^{-t^2/4}.\end{aligned}\tag{124}$$

The calculations for $c_a(t)$, $a = 3,4$, can be easily made following the same techniques, see App. B. A plot of $\overline{c_2(t)}^{\text{GDE}}$ can be found in Fig. 13(*Right*).

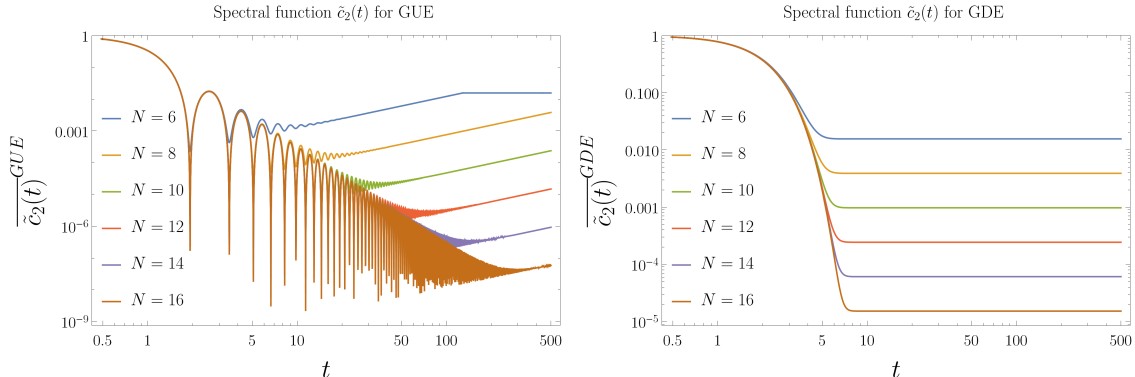

Figure 13: *Left:* Log Log plot of the rescaled spectral function $\tilde{c}_2(t) = c_2(t)/d^2$ averaged over the GDE ensemble for different system size $d = 2^N$ with $N = 6\dots16$. *Right:* Log Log plot of the spectral function $\tilde{c}_2(t) = c_2(t)/d^2$ averaged over the GDE ensemble for different system size $d = 2^N$ with $N = 6\dots16$.

## 4.4 Nearest level spacing distribution: technique

In Sec. 2.2 we claimed that the spectral functions $c_\pi^{(2k)}(U)$ are functions of the nearest level spacing distribution $P(s)$; in this section we illustrate the technique used to average over families E of Hamiltonians with a given nearest level spacings distribution $P_E(s)$. The nearest level spacings distribution we consider are listed in Eq. (15) and reported here for convenience:

$$
\begin{aligned}
P_P(s) &= e^{-s} \quad \text{Poisson,} \\
P_{\text{WD-GOE}}(s) &= \frac{\pi}{2}s\,e^{-\frac{\pi}{4}s^2} \quad \text{Wigner-Dyson from GOE,} \\
P_{\text{WD-GUE}}(s) &= \frac{32}{\pi^2}s^2\,e^{-\frac{4}{\pi}s^2} \quad \text{Wigner-Dyson from GUE.}
\end{aligned}
\tag{125}
$$

We use the same techniques of [108], i.e. we define the characteristic functions of the probability distributions $P_{\mathcal{E}}(s)$; a calculation for the spectral average of $c_2(t)$ for Poisson recently appeared in [205]. Here we compute $c_2(t)$, while the details of $c_a(t)$ for $a = 3, 4$ can be found App. B. The gaps between the energy levels are

$$
\Delta_{ij} = E_i - E_j, \quad i > j, \quad i, j \in [1, d]
\tag{126}
$$

there are $d(d-1)/2$ gaps, but only $d-1$ are independent from each other. Indeed, defining the nearest level spacings as:

$$
s_\alpha = E_{\alpha+1} - E_\alpha, \quad \alpha \in [1, d-1],
\tag{127}
$$

we can express the gaps $\Delta_{ij}$ Eq. (126) for any $i, j$ as a linear combination of $s_\alpha$, namely:

$$
\Delta_{ij} = \sum_{\alpha=j}^{i-1} s_\alpha.
\tag{128}
$$

To compute the spectral average of $c_2(t)$ with respect to an family of Hamiltonians E with a given nearest level spacings distribution, we first need to express $c_2(t)$ in terms of $s_\alpha$:

$$
c_2(t) = d + \sum_{i \neq j} e^{i(E_i - E_j)t} = d + \sum_{i \neq j} \cos[(E_i - E_j)t] + i \sin[(E_i - E_j)t] = d + 2\sum_{i > j} \cos[(E_i - E_j)t],
\tag{129}
$$

i.e for $c_2(t)$ only the real part contributes thanks to the symmetry $i \leftrightarrow j$. From Eq. (126) and Eq. (128):

$$c_2(t) = d + 2\sum_{i=2}^{d}\sum_{j=1}^{i-1}\cos(\Delta_{ij}t) = d + 2\sum_{i=2}^{d}\sum_{j=1}^{i-1}\cos\left(\sum_{\beta=j}^{i-1}s_\beta t\right). \tag{130}$$

Let us use the expansion of the cosine in terms of exponentials:

$$c_2(t) = d + 2\sum_{i=3}^{d}\sum_{j=1}^{i-2}\cos\left(\sum_{\beta=j}^{i-1}s_\beta t\right) = d + \sum_{i=3}^{d}\sum_{j=1}^{i-2}\left(\prod_{\beta=j}^{i-1}e^{is_\beta t} + \prod_{\gamma=j}^{i-1}e^{-is_\gamma t}\right). \tag{131}$$

The next step is to take the spectral average of $c_2(t)$. In order to make the average of $c_2(t)$ Eq. (131) tractable, we use the hypothesis of molecular chaos, that is, the *stosszahlansatz* [206], for which we consider $s_\alpha$'s to be independent identically distributed stochastic variables; let us define the operation formally: let $f(\{s_\alpha\})$ be a function of the nearest level spacings, we compute its spectral average as:

$$\overline{f(\{s_\alpha\})}^{\mathrm{E}} = \int ds_1 \cdots ds_{d-1}\, f\{s_\alpha\}P_{\mathrm{E}}(s_1)\cdots P_{\mathrm{E}}(s_{d-1}); \tag{132}$$

this hypothesis works well for Integrable Hamiltonians and therefore for $\mathrm{E} \equiv P$ [8,113]. Given a probability distribution on the nearest level spacings $P_{\mathrm{E}}(s)$ we define the characteristic function of the level spacing $s_\alpha$ [108]:

$$g_{\mathrm{E}}(t) = \int_0^\infty ds_\alpha\, e^{is_\alpha t}P_{\mathrm{E}}(s_\alpha), \tag{133}$$

which, thanks to the *stosszahlansatz*, does not depends on $\alpha$. Taking the spectral average of Eq. (131), we get:

$$\overline{c_2(t)}^{\mathrm{E}} = d + \sum_{i=3}^{d}\sum_{j=1}^{i-2}\left(\prod_{\beta=j}^{i-1}g_{\mathrm{E}}(t) + \prod_{\gamma=j}^{i-1}g_{\mathrm{E}}^*(t)\right). \tag{134}$$

Making the sum with the usual techniques, one has the final result:

$$\overline{c_2(t)}^{\mathrm{E}} = d + \frac{g_{\mathrm{E}}^{d+1}(t) - d g_{\mathrm{E}}^2(t) + (d-1)g_{\mathrm{E}}(t)}{(g_{\mathrm{E}}(t)-1)^2} + \mathrm{c.c.} \tag{135}$$

The characteristic functions for Poisson, Wigner-Dyson from GOE and GUE read:

$$\begin{aligned}
g_{\mathrm{P}}(t) &= \frac{i}{i+t} \\
g_{\mathrm{WDO}}(t) &= 1 - t\, e^{-t^2/\pi}(\mathrm{erfi}(t/\sqrt{\pi}) - i) \\
g_{\mathrm{WDU}}(t) &= \frac{i}{8}\left(4t - e^{-\pi t^2/16}(\pi t^2 - 8)(\mathrm{erfi}(\sqrt{\pi}t/4) - 1)\right),
\end{aligned} \tag{136}$$

where $\mathrm{erfi}(x)$ is the imaginary error function, defined as $\mathrm{erfi}(z) := -i\,\mathrm{erf}(iz)$ and $\mathrm{erf}(x)$ is the error function defined through the Gaussian. Plugging Eq. (136) in Eq. (135) we have the spectral average of $c_2(t)$ with respect to Poisson and Wigner-Dyson (GOE, GUE); these results are plotted in Fig. 14.

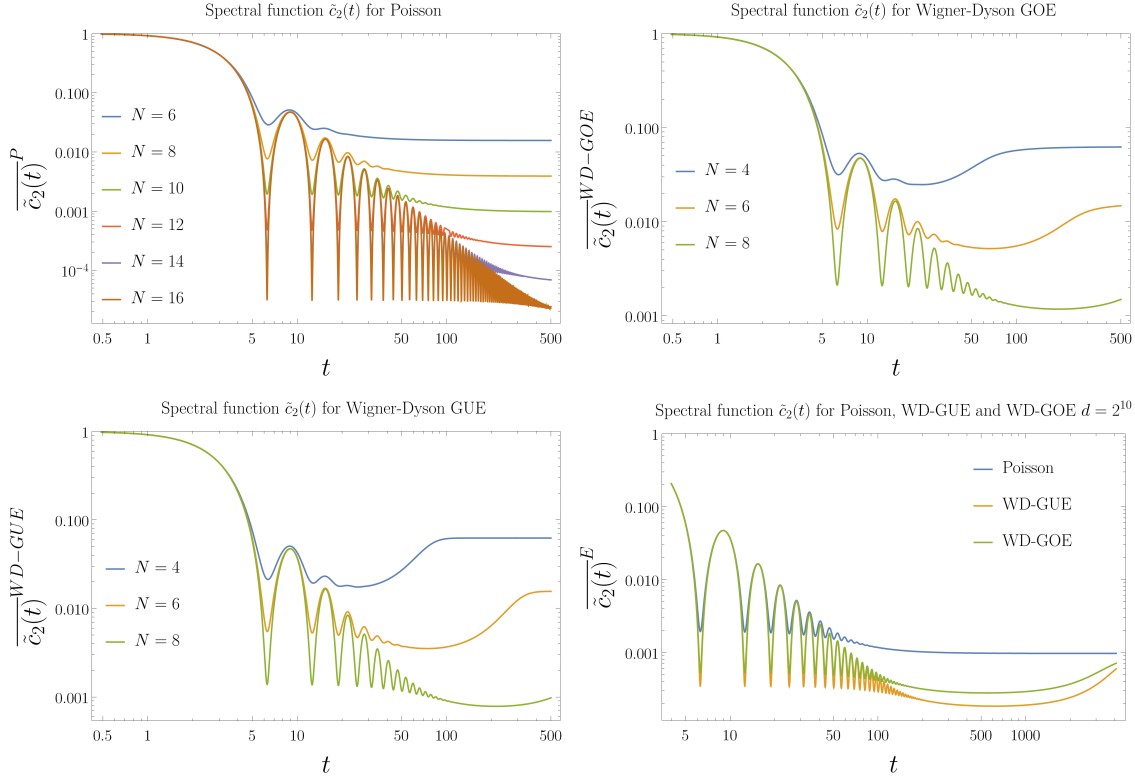

Figure 14: *Top-Left*: Log Log plot of $\tilde{c}_2(t)$ for Poisson and different system size $d = 2^N, N = 4, \ldots, 16$. Starting from 1, it reaches the asymptotic value $1/d$ in a time $O(d^{1/2})$. *Top-Right*: Log Log plot of Wigner-Dyson GOE for different system size $2^N$, $N = 4, 6, 8$. Starting from 1, it goes below the asymptotic value $1/d$ in a time $O(d^{1/2})$ and reaches the dip $O(d^{-2})$ in a time $O(d^{3/4})$ and the asymptotic value in a time $O(d)$. *Bottom-Right*: Log Log plot of Wigner-Dyson GUE for different system size $2^N$, $N = 4, 6, 8$. Starting from 1, it goes below the asymptotic value $1/d$ in a time $O(d^{1/2})$ and reaches the dip $O(d^{-2})$ in a time $O(d^{3/4})$ and the asymptotic value in a time $O(d)$. *Bottom-Left*: Comparison between Poisson, Wigner-Dyson from GOE and GUE for $d = 2^{10}$.

### 4.4.1 Normalization of the energy scale.

Throughout the paper we compare the spectral average of $c_a(t)$ for various ensembles of Hamiltonians. To this aim we need to compare Hamiltonians whose largest eigenvalues scale in the same way with the Hilbert space dimension $d$. In Sec. 2.2, we have defined GUE and GDE ensembles such that the largest eigenvalue is $O(1)$. We need to normalize the energy scale of the nearest level spacings distributions; indeed, we claim that, in our settings, the largest eigenvalues for the nearest level spacings distributions is $O(d)$; we give the following argument: compute the average maximum spacing $\Delta_{d1}$:

$$\overline{\Delta_{d1}}^{\mathrm{E}} = \sum_{i=1}^{d-1} \overline{s_i}^{\mathrm{E}} = d - 1, \tag{137}$$

where we used the fact that $\bar{s} = 1$; indeed the distributions in Eq. (15) are normalized such that $\int \mathrm{d}s P_{\mathrm{E}}(s) = 1$ and $\bar{s} = \int \mathrm{d}s\, P_{\mathrm{E}}(s)s = 1$. This implies the maximum eigenvalue to be $O(d)$: without loss of generality choose $E_1 = 0$:

$$\overline{E_d}^{\mathrm{E}} = d - 1 = O(d), \tag{138}$$

i.e. the nearest level spacings distribution for $H$ and $H - E_0 \mathbb{1}$ is the same. The normalization of the largest eigenvalue can be done by the replacement $t \longrightarrow t/d$ in the expression of $c_a(t)$.

### 4.4.2 Envelope curves

In this section present the envelope curves of the spectral average of $c_2(t)$ and $c_4(t)$, for Poisson and GUE ensembles, from which the scaling behavior in Table 2 can be computed. We evaluate the envelope curves using the method of [103], i.e. we calculate the asymptotic behavior of $c_2$ and $c_4$ for large $d$ and then we suppress the oscillating part. Using this technique, the envelope curves for $\overline{\tilde{c}_2(t)}^E$, for E $= P$, GUE are given by:

$$\overline{c_2(t)}^P \approx \frac{1}{d} + \frac{2}{t^2} \tag{139}$$

$$\overline{c_2(t)}^{GUE} \approx \frac{1}{d} + \frac{1}{\pi t^3} + r_2(t), \tag{140}$$

where we define:

$$r_2(t) \equiv \theta(t - 2d)\left(1 - \frac{t}{2d}\right). \tag{141}$$

The envelope curves for $\overline{\tilde{c}_4(t)}^E$ are then

$$\overline{\tilde{c}_4(t/d)}^P \approx \frac{2d-1}{d^3} + \frac{6}{t^4} + \frac{16d-15}{d^2 t^2} \tag{142}$$

$$\overline{\tilde{c}_4(t/d)}^{GUE} \approx \frac{1}{\pi^2 t^6} + \frac{1}{d^2}\left[2 - \frac{31}{8\pi t^3} - \frac{8r_2(2t) - 16r_2(t) + r_2(t)/\sqrt{2}}{\pi^{3/2} t^{5/2}} - 4r_2(t) + 2r_2^2(t) + \frac{r_2^2(t)}{\pi^2 t^2}\right]. \tag{143}$$

In order to study the timescales of GUE up to the dip time, as claimed in [103], it is sufficient to consider simpler expressions: $\overline{c_2}^{GUE} \approx 1/\pi t^3$ and $\overline{c_4}^{GUE} \approx 1/\pi^2 t^6$. Let us give some example of the calculations for the timescales presented in Table 2; let us compute the equilibrium time for Poisson, see Eq. (139). We define the equilibrium time $t_p$ as the time for which the function $2/t^2$ become comparable with the factor $1/d$ and therefore we need to solve the following equation for $t$:

$$\frac{2}{t^2} \approx \frac{1}{d} \implies t_p \approx \sqrt{2d} = O(\sqrt{d}). \tag{144}$$

The equilibrium time for GDE can be obtained with a similar argument from the exact expression of the spectral average, see Eq. (124):

$$\overline{c_2(t)}^{GDE} = d + d(d-1)e^{-t^2/4}, \tag{145}$$

then $\overline{c_2(t)}^{GDE} \approx d^{-1}$ for $t = O(\sqrt{\log d})$.

## 4.5 Asymptotic value collapse

In this section we give the mathematical motivation for which the asymptotic value of GUE and GDE are the same. The reason lies on the properties of the spectral distributions. Indeed here we prove that if the spectral probability distribution $P(\{E_i\})$ is Schwartz, namely it decays to zero faster than any polynomial of generic degree, then the asymptotic value of the spectral average does not depend on the particular choice of $P(\{E_i\})$.

**Proposition.** Let E to be an ensemble of isospectral Hamiltonians characterized by a Schwartz probability distribution $P(\mathbf{E})$ with $\mathbf{E} \equiv (E_1, \ldots, E_d)$, then the spectral average of the coefficients $c_\pi^{(2k)}(U)$ reads:

$$\lim_{t \to \infty} \overline{c_\pi^{(2k)}(U)}^{\mathrm{E}} = f_\pi(d), \tag{146}$$

where $f_\pi(d)$ depends on the permutation element $\pi$ and on the dimension of the Hilbert space $\mathcal{H}$, but on the particular choice E.

*Proof.* In general, the spectral function $c_\pi^{(2k)}(U)$ can be written as:

$$c_\pi^{(2k)}(U) = C_d (\mathrm{tr}(U^\alpha))^\gamma (\mathrm{tr}(U^{\dagger^\beta}))^\delta = C_d \sum_{\substack{i_1, \ldots, i_\gamma \\ j_1, \ldots, j_\delta}} e^{i\left(\alpha \sum_{\mu=1}^\gamma E_{i_\mu} t - \beta \sum_{\nu=1}^\delta E_{j_\nu}\right)t}, \tag{147}$$

where $\alpha, \beta, \gamma, \delta$ and $C_d$ are assigned to $\pi$ according to Tables 3 and 4 and the overall sum contains $d^{\gamma+\delta}$ elements. From decompositions in Eq. (172) and Eq. (212) is clear that the these coefficients split in two parts: one independent from $t$ (which is the asymptotic value $f_\pi(d)$) and one dependent from $t$. This decomposition does not depend on the particular choice of the spectral probability distribution, but on the powers $\alpha, \beta, \gamma, \delta$. Indeed, because of the linearity of the exponent in Eq. (147) with respect to the energies $E_i$, we can recast it in the form of a scalar product in $\mathbb{R}^d$:

$$\alpha \sum_{\mu=1}^\gamma E_{i_\mu} - \beta \sum_{\nu=1}^\delta E_{j_\nu} = \mathbf{C}_{i_1, \ldots, i_\gamma}^{j_1, \ldots, j_\delta} \cdot \mathbf{E}, \tag{148}$$

where we have defined vectors $\mathbf{C}_{i_1, \ldots, i_\gamma}^{j_1, \ldots, j_\delta} \in \mathbb{R}^d$, one for each element of the sum in Eq. (147); their components depend on $\alpha, \beta$ and take into account the coefficients of the linear combination of the $E_i$s. Defining the multi-index $a = (i_1, \ldots, i_\gamma, j_1, \ldots, j_\delta) \in [1, d^{\gamma+\delta}]$ to clean the notation, we can rewrite Eq. (147) as:

$$c_\pi^{(2k)}(U) = C_d \sum_{a=1}^{d^{\gamma+\delta}} e^{i\mathbf{C}_a \cdot \mathbf{E} t}, \tag{149}$$

let us split the sum in two parts: $\mathbf{C}_a \cdot \mathbf{E} = 0$ and $\mathbf{C}_a \cdot \mathbf{E} \neq 0$:

$$c_\pi^{(2k)}(U) = C_d \sum_{a=1}^{d^{\gamma+\delta}} \delta(\mathbf{C}_a \cdot \mathbf{E}) + C_d \sum_{\substack{a=1 \\ (\mathbf{C}_a \cdot \mathbf{E} \neq 0)}}^{d^{\gamma+\delta}} e^{i\mathbf{C}_a \cdot \mathbf{E} t} = f_\pi(d) + C_d \sum_{b=1}^D e^{i\mathbf{C}_b \cdot \mathbf{E} t}, \tag{150}$$

where we used $\delta(x) = 1$ iff $x = 0$ and we defined:

$$f_\pi(d) := C_d \sum_{a=1}^{d^{\gamma+\delta}} \delta(\mathbf{C}_a \cdot \mathbf{E}), \tag{151}$$

we re-labeled as $\mathbf{C}_b$ the first $D = d^{\gamma+\delta} - f_\pi(d)$ vectors satisfying $\mathbf{C}_b \cdot \mathbf{E} \neq 0$. Taking the spectral average (see Eq. (10)) of the r.h.s. of Eq. (150) we finally get:

$$\overline{c_\pi^{(2k)}(U)}^{\mathrm{E}} = f_\pi(d) + C_d \sum_{b=1}^D \int d\mathbf{E} P(\mathbf{E}) e^{i\mathbf{C}_b \cdot \mathbf{E} t} = f_\pi(d) + C_d \sum_{b=1}^D \hat{P}(\mathbf{C}_b t), \tag{152}$$

where $\hat{P}(\mathbf{x})$ is the Fourier transform of $P(\mathbf{E})$; taking the limit $t \to \infty$ of both sides of Eq. (152) one get the desired result. Indeed the Fourier transform of a Schwartz function is a Schwartz function and dies asymptotically.

We remark that the asymptotic value for the ensemble average, Eq. (146), coincides with the infinite time average, defined in Eq. (27), of $c_\pi^{(2k)}(U)$; indeed, from Eq. (149), we have:

$$\mathbb{E}_T\left[c_\pi^{(2k)}(U)\right] = \lim_{T\to\infty}\frac{1}{T}\int_0^T dt\, c_\pi^{(2k)}(U) = C_d\sum_{a=1}^{d^{\gamma+\delta}}\lim_{T\to\infty}\frac{1}{T}\int_0^T dt\, e^{i\mathbf{C}_a\cdot\mathbf{E}t}. \tag{153}$$

It is well known that the above integral is always null, but for $\mathbf{C}_a\cdot\mathbf{E}=0$; therefore:

$$\mathbb{E}_T\left[c_\pi^{(2k)}(U)\right] = C_d\sum_{a=1}^{d^{\gamma+\delta}}\delta(\mathbf{C}_a\cdot\mathbf{E}) = f_\pi(d), \tag{154}$$

i.e. we proved that the ensemble average for Schwartzian probability distributions and the infinite time average do commute.

## 4.6 Lévy lemma and typicality

In this section we apply the Lévy lemma [116] on $\mathcal{P}_\mathcal{O}(G) = \text{tr}(\tilde{T}^{(2k)}\mathcal{O}G^{\dagger\otimes 2k}U^{\otimes k,k}G^{\otimes 2k})$, where $G\in\mathcal{U}(\mathcal{H})$ and prove the bound Eq. (15). The Lèvi lemma states that if $f: G\longrightarrow\mathbb{C}$ is $\eta$-Lipschitz, then [139]:

$$\Pr\left(|f(G) - \langle f(G)\rangle_G|\geq\delta\right)\leq 4\exp\left\{-\frac{2d\delta^2}{9\eta^2\pi^3}\right\}, \tag{155}$$

where the average $\langle f(G)\rangle_G$ is taken with the Haar measure over the unitary group. This statement is extremely useful in the context of many body physics; the Hilbert space dimension $d$, appearing at the exponent in Eq. (155), grows exponentially with the system size, therefore, if the Lipschitz constant $\eta$ does not scale with $d$, the probability of finding a value $f(G)$ different from its average $\langle f(G)\rangle_G$ is double exponential small in the system size. This concentration bound is way stronger than the Chebyshev inequality. Let us prove the bound in Eq. (15).

**Proposition.** Let $\mathcal{P}_\mathcal{O}(G) = \text{tr}(\tilde{T}^{(2k)}\mathcal{O}G^{\dagger\otimes 2k}U^{\otimes k,k}G^{\otimes 2k})$, then:

$$|\mathcal{P}_\mathcal{O}(G) - \mathcal{P}_\mathcal{O}(G')|\leq\eta\|G-G'\|_2, \tag{156}$$

where $\eta = 4k\|\mathcal{O}\|_1\text{tr}^{-1}(T^{(2k)})$.

*Proof.* First of all note that $\mathcal{P}_\mathcal{O}(G) = \text{tr}(\tilde{T}^{(2k)}\mathcal{O}(G^\dagger UG)^{\otimes k,k})$, where $A^{\otimes k,k}\equiv A^{\otimes k}\otimes A^{\dagger\otimes k}$. Consider the following series of inequalities:

$$\begin{aligned}
|\mathcal{P}_\mathcal{O}(G) - \mathcal{P}_\mathcal{O}(G')| &= |\text{tr}\left(\tilde{T}^{(2k)}\mathcal{O}((G^\dagger UG)^{\otimes k,k} - (G'^\dagger UG')^{\otimes k,k})\right)| \\
&\leq \|T^{(2k)}\|_\infty\|\mathcal{O}\|_1\text{tr}^{-1}(T^{(2k)})\|(G^\dagger UG)^{k,k} - (G'^\dagger UG')^{k,k}\|_\infty \quad (157) \\
&\leq 2k\|\mathcal{O}\|_1\text{tr}^{-1}(T^{(2k)})\|G^\dagger UG - G'^\dagger UG'\|_\infty \\
&\leq 4k\|\mathcal{O}\|_1\text{tr}^{-1}(T^{(2k)})\|G-G'\|_\infty \\
&\leq 4k\|\mathcal{O}\|_1\text{tr}^{-1}(T^{(2k)})\|G-G'\|_2.
\end{aligned}$$

The first equality follows from the linearity of the trace. The first inequality follows from $|\text{tr}(AB)|\leq\|A\|_p\|B\|_q$, where $p^{-1}+q^{-1}=1$, $\|\cdot\|_p$ is the Schatten p-norm [116] and from $\|AB\|_p\leq\|A\|_p\|B\|_p$; the second inequality has been proven in [138], we include here the proof for completeness: consider $A,B,C,D$ to be unitary operators, then

$$\begin{aligned}
\|A\otimes B - C\otimes D\|_\infty &= \|A\otimes(B-D) + (A-C)\otimes D\|_\infty \tag{158} \\
&\leq \|A\|_\infty\|B-D\|_\infty + \|D\|_\infty\|A-C\|_\infty \\
&= \|B-D\|_\infty + \|A-C\|_\infty
\end{aligned}$$

indeed for any unitary operator $U$, $\|U\|_\infty = 1$. By iterative application of the latter inequalities one get:

$$\|(G^\dagger U G)^{k,k} - (G'^\dagger U G')^{k,k}\|_\infty \leq 2k \|G^\dagger U G - G'^\dagger U G'\|_\infty. \tag{159}$$

The third inequality in Eq. (158) is obtained by similar techniques:

$$
\begin{aligned}
\|G^\dagger U G - G'^\dagger U G'\|_\infty &= \|G^\dagger U (G - G') + (G^\dagger - G'^\dagger) U G'\|_\infty \\
&\leq \|G^\dagger U\|_\infty \|G - G'\|_\infty + \|U G'\|_\infty \|G^\dagger - G'^\dagger\|_\infty \\
&= 2\|G - G'\|_\infty.
\end{aligned}
\tag{160}
$$

The last inequality in Eq. (158) follows easily from the hierarchy of the Schatten $p$-norm $\|A\|_p \leq \|A\|_q$ if $p > q$. This concludes the proof.

## 5 Summary and Commentary of the Results

We now want to summarize and comment the main findings and results of this paper. In Sec.2 we define the general approach to the study of the isospectral twirling as way to unifying probes to quantum chaos. We define the $2k$- isospectral twirling of a unitary Channel $U$ as

$$\hat{\mathcal{R}}^{(2k)}(U) := \int dG\, G^{\dagger \otimes 2k} \left( U^{\otimes k} \otimes U^{\dagger \otimes k} \right) G^{\otimes 2k}. \tag{161}$$

This quantity arises naturally if one wants to take the average expectation values of polynomials of the time evolution $\mathcal{U}(X) = U^\dagger X U$ of some operator $X$ over all the evolutions with the same spectrum. The information about the spectrum is contained in the spectral form factors $\tilde{c}_a(t)$, see Eq.(5). Therefore, in Sec.2.2, we use random matrix theory to compute the average behavior of the $\tilde{c}_a(t)$ for relevant ensembles of random Hamiltonians like GUE, GDE and Poisson. The results, summarized in table 2, show that the spectral form factors characterize the different ensembles in the way the salient characteristics of the time evolution are reached and their values. We also show that these behaviors are typical. The first result of this paper is thus that, through spectral form factors, one can indeed distinguish ensembles of random Hamiltonians. Such quantities, though, are not of immediate accessibility.

In order to probe quantum chaos, one has to define some observable or correlation function and see how they behave for different quantum evolutions. OTOCs, entanglement, Loschmidt Echos, frame potentials, mutual information, scrambling and coherence have all been proposed as probes to quantum chaos with some notable heuristic arguments. The second result of this paper is that we show that, through the isospectral twirling, they all assume the form of the expectation value of the isospectral twirling with some (permuted) operator. Then one understands why all those probes are actual probes into quantum chaos: that is because they feature the spectral form factors in a way that is dominant with respect to the dimension $d$ of the Hilbert space. Then, through the spectral form factors, they efficiently distinguish between different ensembles of Hamiltonians characterized by chaotic or integrable spectra. The results are summarized in table 1.

Different scalings also show why some probes are more telling than others, for instance 4-point versus 2-point OTOCs or different kinds of Loschmidt-Echos. As a result, one has provided a unified framework to discuss the features of all these probes *and* show why they behave differently for integrable Hamiltonians, which was a piece of analysis largely still lacking. Once one has understood the intimate relationship between all these quantities, one has a systematic way to study other quantities. For instance, one is then interested in understanding whether the work performed by a quantum battery can reveal whether the driving Hamiltonian is integrable and chaotic. Indeed, one would like to see how such notions have an impact

in macroscopic situations like those of quantum thermodynamics. It turns out that the average work will not care much about the chaoticity of the driving Hamiltonian, but its fluctuations yes, thus establishing a novel connection between quantum thermodynamics and the study of quantum chaos and quantum integrability.

Finally, we applied the techniques of isospectral twirling to the case of more general quantum maps, showing their usefulness in order to compute quantities like the Loschmidt Echo, coherence or purity for a general quantum channel. These results are more technical, and somehow in search of an application. For instance, one could consider the spectrum of Lindblad superoperators (eigenvalues associated to their left and right eigenvectors) and classify by random matrix theory the typical behavior of the aforementioned probes in open quantum systems.

# 6   Conclusions

Quantum chaos is a property of quantum dynamics that should, on the one hand, feature a very diverse list of properties and high sensitivity to the details of interactions - the butterfly effect - complex growth of entanglement and coherence, information scrambling, and rapid decay of the out-of-time-order correlation functions, and, on the other hand, tell apart the dynamics generated by an integrable Hamiltonian, which should not be chaotic from that of non integrable Hamiltonians, that are supposed to be chaotic. These properties are usually understood in terms of the statistics of the gaps in the energy spectra, resulting in Poisson or universal distributions. In this paper we have unified in a single framework the description of the numerous probes to quantum chaos in a way that allows for a clear classification in terms also of the spectral properties. The (unitary) evolution operator belongs to a family of isospectral operators with different eigenvectors. By averaging over these eigenvectors, one obtains quantities that only depend on the spectrum. The isospectral twirling indeed consists in practice in averaging over the eigenvectors of a given quantum dynamics. By performing calculations in random matrix theory for the appropriate ensembles of integrable or non-integrable Hamiltonians, we showed how the unified probes to quantum chaos neatly separate chaotic from non chaotic behavior in terms of how their values and corresponding time scales scale with the dimension of the Hilbert space $d$. We also showed why the infinite time, large $d$ values do not depend on the type of Hamiltonian chosen and why they detach from the prediction from the Haar measure.

In perspective, a number of open questions can be explored using the several techniques used in this paper. We would like to understand how to incorporate the locality of interactions and what effects this has on quantum chaos. We showed how to extend the isospectral twirling to general completely positive maps. Moreover, it would be interesting to understand in what sense these channels or master equations described by Lindblad operators can be chaotic. Finally, a very interesting subject is the transition from an integrable dynamics to a chaotic one. In the context of quantum circuits, it has been understood that one can dope a random Clifford circuit and obtain higher order unitary $t$-designs or transitions to universal entanglement [140, 207]. In the context of quantum many-body systems, one could study integrable spin chains perturbed by an integrability-breaking term [153] and study the typical behavior in the sense of the isospectral twirling of the probes to quantum chaos. The insights gained in this way could open the way to formulate a quantum KAM theorem, that is, understand to what extent a chaotic system obtained by perturbing an integrable one [23] shows remnants of its lost integrability. Perhaps gazing enough into quantum chaos will have it gaze back at us.

**Acknowledgments** A.H. acknowledges partial support from NSF award number 2014000. F.C. acknowledges the support of NNSA for the U.S. DoE at LANL under Contract No. DE-AC52-06NA25396, also financed via DOE-LDRD ER grants and IMS 2020 Rapid Response.

The authors thank Bin Yan for useful discussions.

# Appendix

## A Computation of $\hat{\mathcal{R}}_4(U)$

In this section we are going to evaluate $\hat{\mathcal{R}}_4(U)$, from the definition Eq. (2) we have for $k = 2$

$$\hat{\mathcal{R}}^{(4)}(U) = \int dG \, G^{\dagger \otimes 4} \left(U^{\otimes 2,2}\right) G^{\otimes 4}, \tag{162}$$

where we recall that $U^{\otimes 2,2} = U^{\otimes 2} \otimes U^{\dagger \otimes 2}$. If we calculate the Haar average as showed in Sec. 4.1, we obtain:

$$\hat{\mathcal{R}}^{(4)}(U) = \sum_{\pi\sigma} (\Omega^{-1})_{\pi\sigma} \text{tr} \left(T_\pi U^{\dagger \otimes 2} \otimes U^{\otimes 2}\right) T_\sigma \equiv \sum_{\pi\sigma} (\Omega^{-1})_{\pi\sigma} c_\pi^{(4)}(U) T_\sigma, \tag{163}$$

where $c_\pi^{(4)}(U) := \text{tr}\left(T_\pi U^{\dagger \otimes 2} \otimes U^{\otimes 2}\right) = C_d (\text{tr} \, U^\alpha)^\gamma (\text{tr} \, U^{\dagger\beta})^\delta$ for the different permutations $T_\pi$. All the coefficients and the value of $\alpha, \beta, \gamma, \delta$ and $C_d$ are listed in Table 4.

Table 4: Calculation of the spectral functions $c_\pi^{(4)}(U) = C_d (\text{tr} \, U^\alpha)^\gamma (\text{tr} \, U^{\dagger\beta})^\delta$, the corresponding exponents $\alpha, \beta, \gamma, \delta$ and the constant $C_d$. The short notation for $c_2(t)$, $c_3(t)$ and $c_4(t)$ is defined in Eq. (8).

| $T_\pi^{(4)}$ | $c_\pi(U)$ | $\alpha$ | $\beta$ | $\gamma$ | $\delta$ | $C_d$ |
|---|---|---|---|---|---|---|
| $\mathbb{1}$ | $|\text{tr}(U)|^4 \equiv c_4(t)$ | 1 | 1 | 2 | 2 | 1 |
| $T_{(12)}$ | $\text{tr}(U^2)\text{tr}(U^\dagger)^2 \equiv c_3(t)$ | 2 | 1 | 1 | 2 | 1 |
| $T_{(13)}$ | $\text{tr}(UU^\dagger)|\text{tr}(U)|^2 \equiv dc_2(t)$ | 1 | 1 | 1 | 1 | $d$ |
| $T_{(14)}$ | $\text{tr}(UU^\dagger)|\text{tr}(U)|^2 \equiv dc_2(t)$ | 1 | 1 | 1 | 1 | $d$ |
| $T_{(23)}$ | $\text{tr}(UU^\dagger)|\text{tr}(U)|^2 \equiv dc_2(t)$ | 1 | 1 | 1 | 1 | $d$ |
| $T_{(24)}$ | $\text{tr}(UU^\dagger)|\text{tr}(U)|^2 \equiv dc_2(t)$ | 1 | 1 | 1 | 1 | $d$ |
| $T_{(34)}$ | $\text{tr}(U)^2\text{tr}(U^{\dagger 2}) \equiv c_3(t)$ | 1 | 2 | 2 | 1 | 1 |
| $T_{(12)(34)}$ | $\text{tr}(U^2)\text{tr}(U^{\dagger 2}) \equiv c_2(2t)$ | 2 | 2 | 1 | 1 | 1 |
| $T_{(13)(24)}$ | $\text{tr}(UU^\dagger)^2 \equiv d^2$ | 1 | 1 | 0 | 0 | $d^2$ |
| $T_{(14)(23)}$ | $\text{tr}(UU^\dagger)^2 \equiv d^2$ | 1 | 1 | 0 | 0 | $d^2$ |
| $T_{(ijk)}$ | $|\text{tr}(U)|^2 \equiv c_2(t)$ | 1 | 1 | 1 | 1 | 1 |
| $T_{(ijkl)}$ | $|\text{tr}(UU^\dagger UU^\dagger)| \equiv d$ | 1 | 1 | 0 | 0 | $d$ |

In order to write an explicitly the 4-Isospectral twirling $\hat{\mathcal{R}}^{(4)}(U)$, we define the following

elements:

$$
\begin{aligned}
Y_{(2)} &\equiv \left(T_{(13)} + T_{(14)} + T_{(23)} + T_{(24)}\right) \\
Y_{(3)}^{(+)} &\equiv \left(T_{(123)} + T_{(124)} + T_{(132)} + T_{(142)}\right) \\
Y_{(3)}^{(-)} &\equiv \left(T_{(234)} + T_{(134)} + T_{(243)} + T_{(143)}\right) \\
Y_{(4)}^{(1)} &\equiv \left(T_{(1234)} + T_{(1432)} + T_{(1243)} + T_{(1342)}\right) \\
Y_{(4)}^{(2)} &\equiv \left(T_{(1324)} + T_{(1423)}\right) \\
Y_{(2)(2)} &\equiv \left(T_{(14)(23)} + T_{(13)(24)}\right),
\end{aligned}
\tag{164}
$$

it is now possible to insert the $c_\pi^U$ in Eq. (163) and then the final formula for $\hat{\mathcal{R}}^{(4)}(U)$ reads:

$$
\begin{aligned}
\hat{\mathcal{R}}^{(4)}(U) = \frac{1}{d^2\left(d^6 - 14d^4 + 49d^2 - 36\right)} &\times \\
\times \Big[ & \left((c_4(t) - 4c_2(t))(d^4 - 8d^2 + 6) + c_2(2t)(d^2 + 6) + \operatorname{Re} c_3(t)(4d - d^3) + 2d^4 - 18d^2\right) \mathbb{1}^{\otimes 4} \\
+ & \left((4d - d^3)(-4c_2(t) + c_2(2t) + c_4(t)) + c_3(t)(d^4 - 8d^2 + 6) + c_3^*(t)(d^2 + 6)\right) T_{(12)} \\
+ & \left(c_4(t)(4d - d^3) + c_2(t)(d^5 - 7d^3 + 2d) - 5c_2(2t)d + (2d^2 - 3)\operatorname{Re} c_3(t) - d^5 + 9d^3\right) Y_{(2)} \\
+ & \left((c_4(t) + c_2(2t) - 4c_2(t))(4d - d^3) + c_3(t)(6 + d^2) + c_3^*(t)(d^4 - 8d^2 + 6)\right) T_{(34)} \\
+ & \left((c_4(t) + c_2(2t))(2d^2 - 3) - c_2(t)(d^4 - d^2 - 12) + c_3(t)(4d - d^3) - 5c_3^*(t)d + d^4 - 9d^2\right) Y_{(3)}^{(+)} \\
+ & \left((c_4(t) + c_2(2t))(2d^2 - 3) - c_2(t)(d^4 - d^2 - 12) + c_3^*(t)(4d - d^3) - 5c_3(t)d + d^4 - 9d^2\right) Y_{(3)}^{(-)} \\
+ & \left(c_2(t)(2d^3 + 2d) + c_2(2t)(4d - d^3) + (2d^2 - 3)\operatorname{Re} c_3(t) - 5c_4(t)d\right) Y_{(4)}^{(1)} \\
+ & \left(c_2(t)(4d^3 - 16d) + (d^2 + 6)\operatorname{Re} c_3(t) - 5d(c_2(2t) + c_4(t)) - d^5 + 9d^3\right) Y_{(4)}^{(2)} \\
+ & \left((6 + d^2)(c_4(t) - 4c_2(t)) - d(d^2 - 4)\operatorname{Re} c_3(t) + (d^4 - 8d^2 + 6)c_2(2t) - 2d^2(d - 9)\right) T_{(12)(34)} \\
+ & \left(c_2(t)(-2d^4 + 14d^2 - 24) + (d^2 + 6)(c_2(2t) + c_4(t)) - 5d \operatorname{Re} c_3(t) + d^6 - 11d^4 + 18d^2\right) Y_{(2)(2)} \Big].
\end{aligned}
\tag{165}
$$

$\hat{\mathcal{R}}^{(4)}(U)$ is a time-dependent operator as $\hat{\mathcal{R}}^{(2)}(U)$; for $t = 0$ it equals $\mathbb{1}^{\otimes 4}$, while the asymptotic limit $t \to \infty$ for the ensemble average we consider is (cfr. Table 2):

$$
\begin{aligned}
\lim_{t \to \infty} \overline{\hat{\mathcal{R}}^{(4)}(t)}^{\,\mathrm{E}} = \frac{1}{d(d+1)(d+2)(d+3)} &\times \Big( (2d^2 + 7d + 4)\mathbb{1}^{\otimes 4} - (2 + d)T_{(12)} \\
&- (2 + d)(T_{(12)} + T_{(34)}) + (d^2 + 3d + 1)Y_{(2)} - (2 + d)(Y_{(3)}^{(+)} + Y_{(3)}^{(-)}) \\
&- (2 + d)Y_{(4)}^{(2)} + Y_{(4)}^{(1)} + (4 + 4d + d^2)Y_{(2)(2)} + (4 + d)T_{(12)(34)} \Big).
\end{aligned}
\tag{166}
$$

In Sec. 4.5 we proved that the infinite time average and the ensemble averages commute; for this reason we can express Eq. (166) in terms of the isospectral twirling of the dephasing superoperator. First, consider the infinite time average of $U^{\otimes 2,2}$:

$$
\begin{aligned}
\mathbb{E}_T(U^{\otimes 2,2}) &= \lim_{T \to \infty} \frac{1}{T} \sum_{ijkl} \int_0^T e^{i(E_i + E_j - E_k - E_l)}(\Pi_i \otimes \Pi_j \otimes \Pi_k \otimes \Pi_l) \\
&= \sum_{ijkl} (\delta_{ik}\delta_{jl} + \delta_{il}\delta_{jk} - \delta_{ik}\delta_{jl}\delta_{il}\delta_{jk})(\Pi_i \otimes \Pi_j \otimes \Pi_k \otimes \Pi_l) \\
&= \sum_{ij} (\Pi_i \otimes \Pi_j \otimes \Pi_i \otimes \Pi_j + \Pi_i \otimes \Pi_j \otimes \Pi_j \otimes \Pi_i - \delta_{ij}\Pi_i \otimes \Pi_i \otimes \Pi_i \otimes \Pi_i).
\end{aligned}
\tag{167}
$$

Now, taking the Isospectral twirling in the sense of quantum maps, defined in Sec. 3.6, we get:

$$\lim_{t\to\infty}\overline{\hat{\mathcal{R}}^{(4)}(t)}^{E} = T_{(23)}\hat{\mathcal{R}}^{(4)}(D_B)T_{(23)} + T_{(24)}\hat{\mathcal{R}}^{(4)}(D_B)T_{(24)} - \frac{\Pi^{+}_{(4)}}{(d+1)(d+2)(d+3)}, \quad (168)$$

where:

$$\hat{\mathcal{R}}^{(4)}(D_B) = -\frac{1[-4(d+1)(d+3)\Pi^{+}_1\Pi^{+}_2 + 24\Pi^{+}_{(4)} - (d+3)\Pi^{+}_1(T_{(13)(24)} + T_{(14)(23)})]}{d(d+1)(d+2)(d+3)}, \quad (169)$$

where $\Pi^{+}_{1(2)}$ is the projector onto the symmetric subspace of the first(second) two copies of $\mathcal{H}$, while $\Pi^{+}_{(4)} = \sum_\pi T^{(4)}_\pi/24$ is the projector onto the symmetric subspace of $\mathcal{H}^{\otimes 4}$.

It is possible to compute the Haar average of the Isospectral twirling with respect to $U$ for as shown in Eq. (110) for $k = 2$, from Eq. (111) reads:

$$\left\langle\hat{\mathcal{R}}^{(4)}(U)\right\rangle_U = \frac{T_{(13)(24)} + T_{(14)(23)}}{d^2-1} - \frac{T_{(1423)} + T_{(1324)}}{d(d^2-1)}. \quad (170)$$

Looking at Eq. (166), it is possible to see that if $c_4(t) = c_2(2t) = 2$, $c_2(t) = 1$ and $\mathrm{Re}\,c_3(t) = 0$ we obtain the Haar value.

# B  Higher-order spectral functions

In this section, we compute spectral averages for the higher-order spectral functions $c_3(t)$ and $c_4(t)$ for different ensembles. This section is organized as follows: Sec. B.1 is devoted to the spectral average of $c_4(t)$, in particular in Sec. B.1.1 we compute the spectral average for GUE, in Sec. B.1.2 the spectral average for GDE and in Sec. B.1.3 the spectral average for nearest level spacings distributions: Poisson and Wigner-Dyson. Sec. B.2 we present the computation of the spectral average of $c_3(t)$; it follows immediately from $c_4(t)$.

## B.1  Spectral function $c_4(t)$

In this section, we develop the calculation of the spectral function $c_4(t)$ introduced in Eq. (8), before to move on the spectral average for each distribution, we give a decomposition of $c_4(t)$, that will help us in the calculation of the spectral average. First, we start recalling the definition of $c_4(t)$:

$$c_4(t) = \mathrm{tr}\,(U)^2\mathrm{tr}\,(U^\dagger)^2 = \sum_{ijkl}e^{i(E_i+E_j-E_j-E_l)t}, \quad (171)$$

it is possible to decompose the sum in Eq. (171) as:

$$\begin{aligned}
c_4(t) &= \sum_{i\neq j\neq k\neq l}e^{i(E_i+E_j-E_k-E_l)t} + \sum_{i\neq j\neq k}e^{i(2E_i-E_j-E_k)t} + 4(d-1)\sum_{i\neq j}e^{i(E_i-E_j)t}\\
&+ \sum_{i\neq j\neq k}e^{-i(2E_i-E_j-E_k)t} + \sum_{i\neq j}e^{2i(E_i-E_j)t} + 2d(d-1) + d,
\end{aligned} \quad (172)$$

that can also be written in terms of cosines as:

$$\begin{aligned}
c_4(t) &= \sum_{ijkl}e^{i(E_i+E_j-E_k-E_l)t} = d + 2d(d-1) + \sum_{i\neq j}\cos(2E_i-2E_j)t + 4(d-1)\sum_{i\neq j}\cos(E_i-E_j)t\\
&+ 2\sum_{i\neq j\neq k}\cos(2E_i-E_j-E_k)t + \sum_{i\neq j\neq k\neq l}\cos(E_i+E_j-E_k-E_l)t.
\end{aligned} \quad (173)$$

### B.1.1 Calculation of $\overline{c_4(t)}^{\text{GUE}}$

The coefficient $\overline{c_4(t)}^{\text{GUE}}$ can be defined trough $c_4(t)$ as:

$$\overline{c_4(t)}^{\text{GUE}} = \int dE_1\, dE_2\, dE_3\, dE_4\, P_{\text{GUE}}(\{E_m\}) c_4(t), \tag{174}$$

before to proceed with the calculations, we introduce the following functions:

$$r_1(t) \equiv \frac{J_1(2t)}{t}, \quad r_2(t) \equiv \theta(2d-t)\left(d-\frac{t}{2}\right), \quad r_3(t) \equiv \frac{\sin(\pi t/2)}{\pi t/2}. \tag{175}$$

The first two terms $r_1(t)$ and $r_2(t)$ appear also in $\overline{c_2(t)}^{\text{GUE}}$, while the third term comes from the convolution of the kernel Eq. (120) with itself when $i \neq j$, a general result for the convolution of kernels when $i \neq j$ is provided in Theorem 2.1 and Theorem 2.2 of [204]. We can decompose $c_4(t)$ as in Eq. (172) and the definition and making use of the definition of $\rho^{(n)}(E_1, \cdots, E_n)$ given in Eq. (118), we obtain:

$$
\begin{aligned}
\overline{c_4(t)}^{\text{GUE}} &= d(d-1)(d-2)(d-3)\int dE_1\, dE_2\, dE_3\, dE_4\, \rho^{(4)}_{\text{GUE}}(E_1, E_2, E_3, E_4) e^{i(E_1+E_2-E_3-E_4)t} \\
&+ 2d(d-1)(d-2)\,\text{Re}\int dE_1\, dE_2\, dE_3\, \rho^{(3)}_{\text{GUE}}(E_1, E_2, E_3) e^{i(2E_1-E_2-E_3)t} \\
&+ d(d-1)\int dE_1\, dE_2\, \rho^{(2)}_{\text{GUE}}(E_1, E_2) e^{i(2E_1-2E_2)t} \\
&+ 4d(d-1)^2\int dE_1\, dE_2\, \rho^{(2)}_{\text{GUE}}(E_1, E_2) e^{i(E_1-E_2)t} + 2d(d-1)+d \\
&= d(d-1)(d-2)(d-3)\int dE_1\, dE_2\, dE_3\, dE_4\, \rho^{(4)}_{\text{GUE}}(E_1, E_2, E_3, E_4) e^{i(E_1+E_2-E_3-E_4)t} \\
&+ 2d(d-1)(d-2)\,\text{Re}\int dE_1\, dE_2\, dE_3\, \rho^{(3)}_{\text{GUE}}(E_1, E_2, E_3) e^{i(2E_1-E_2-E_3)t} \\
&+ d^2\,|r_1(2t)|^2 - d r_2(2t) + 4(d-1)\left(d^2\,|r_1(t)|^2 - d r_2(t)\right) + 2d(d-1)+d.
\end{aligned}
\tag{176}
$$

The complete calculation for the two integral terms can be found in [204], here we show just the final result for $\overline{c_4(t)}^{\text{GUE}}$ and it is:

$$
\begin{aligned}
\overline{c_4(t)}^{\text{GUE}} &= d^4\,|r_1(t)|^4 - 2d^3\,\text{Re}\left(r_1^2(t)\right) r_2(l) r_3(2t) - 4d^3\,|r_1(t)|^2 r_2(t) + 2d^3\,\text{Re}\left(r_1(2t) r_1^{*2}(t)\right) \\
&+ 4d^3\,|r_1(t)|^2 + 2d^2 r_2^2(t) + d^2 r_2^2(t) r_3^2(2t) + 8d^2\,\text{Re}\left(r_1(t)\right) r_2(t) r_3(t) \\
&- 2d^2\,\text{Re}\left(r_1(2t)\right) r_3(t) r_2(t) + d^2\,|r_1(2t)|^2 - 4d^2\,\text{Re}\left(r_1^*(t)\right) r_3(t) r_2(2t) - 4d^2\,|r_1(t)|^2 \\
&- -4d^2 r_2(t) + 2d^2 - 7d r_2(2t) + 4d r_2(3t) + 4d r_2(t) - d.
\end{aligned}
$$

The 4-point spectral for factor is plotted in Fig. 15

### B.1.2 Calculation of $\overline{c_4(t)}^{\text{GDE}}$

The calculations for the spectral function $\overline{c_4(t)}^{\text{GDE}}$ repeat similarly to what done for the spectral function $\overline{c_2(t)}^{\text{GDE}}$, we decompose first as Eq. (173) and after we take the average over the GDE

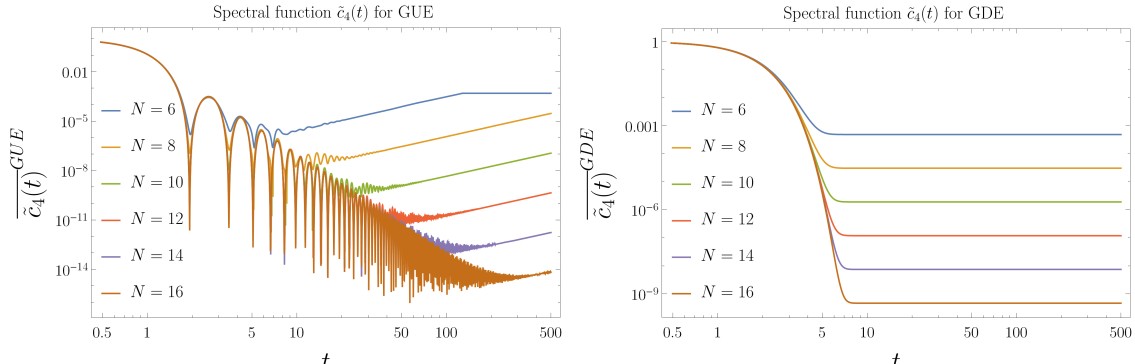

Figure 15: *Left*:Log Log plot of the normalized spectral function $\tilde{c}_4(t) = c_4(t)/d^4$ averaged over the GUE ensemble for different system size $d = 2^N$ with $N = 6, \ldots, 16$. *Right*: Log Log plot of the rescaled spectral function $\tilde{c}_4(t) = c_4(t)/d^4$ averaged over the GDE ensemble for different system size $d = 2^N$ with $N = 6, \ldots, 16$

ensemble, we obtain the following terms:

$$
\begin{aligned}
\overline{\sum_{i \neq j \neq k \neq l} e^{i(E_i+E_j-E_k-E_l)t}}^{\text{GDE}} &= d(d-1)(d-2)(d-3)e^{-t^2/2} \\
\text{Re} \overline{\sum_{i \neq k \neq l} e^{i(2E_i-E_k-E_l)t}}^{\text{GDE}} &= d(d-1)(d-2)e^{-3t^2/4} \\
\overline{\sum_{i \neq j} e^{2i(E_i-E_j)t}}^{\text{GDE}} &= d(d-1)e^{-t^2} \\
\overline{\sum_{i \neq j} e^{i(E_i-E_j)t}}^{\text{GDE}} &= d(d-1)e^{-t^2/4}.
\end{aligned}
\tag{177}
$$

The final expression for $\overline{c_4(t)}^{\text{GDE}}$ is:

$$
\overline{c_4(t)}^{\text{GDE}} = d(2d-1)+4d(d-1)(d-1)e^{-t^2/4}+2d(d-1)(d-2)e^{-3t^2/4}+d(d-1)(d-2)(d-3)e^{-t^2/2}.
\tag{178}
$$

The asymptotic value is:

$$
\lim_{t \to \infty} \overline{c_4(t)}^{\text{GDE}} = d(2d-1).
\tag{179}
$$

The time for which it achieves the equilibrium is the same as for $\overline{c_2(t)}^{\text{GDE}}$, that is $O(\sqrt{\log d})$. A plot for $\overline{c_4(t)}^{\text{GDE}}$ can be found in Fig. 15.

### B.1.3 Calculation of $\overline{c_4(t)}^{\text{E}}$ for nearest level spacing

In this section, we calculate the coefficient $\overline{c_4(t)}^{\text{E}}$ for the nearest level spacing distribution, and we present the plot of the averaged spectral function $c_4(t)$ for the Poisson distribution, Wigner-Dyson GUE distribution and Wigner-Dyson GOE distribution. We use the same technique presented in Sec. 4.4. The calculation for $\overline{c_4(t)}^{\text{E}}$ start decomposing $c_4(t)$ as in Eq. (173) and subsequently we take the average over the nearest level spacing distributions Taking the average over the ensembles E, we have four dynamics terms, two of them are already evaluated

in (135):

$$\overline{\sum_{i\neq j} e^{i(E_i - E_j)t}} = \frac{g_E^{d+1}(t) - d g_E^2(t) + (d-1)g_E(t)}{(g_E(t) - 1)^2} + \text{c.c}$$

$$\overline{\sum_{i\neq j} e^{i2(E_i - E_j)t}} = \frac{g_E^{d+1}(2t) - d g_E^2(2t) + (d-1)g_E(2t)}{(g_E(2t) - 1)^2} + \text{c.c.} \tag{180}$$

We first deal with the average of $\sum_{i\neq j\neq k\neq l} \cos(E_i + E_j - E_k - E_l)t$. It is possible to point out that there are 4! ways to order four items, but with the conditions, $i > j$ and $k > l$ they can be reduced to six, the following ones:

$$\begin{array}{llll} 1. & i > j > k > l, & 2. \quad k > l > i > j, & 3. \quad i > k > l > j, \\ 4. & i > k > j > l, & 5. \quad k > i > l > j, & 6. \quad k > i > j > l, \end{array} \tag{181}$$

noting that there is another symmetry, that is the transformation $i \to k$ and $j \to l$. Due to this symmetry, we have only three remaining different orders:

$$\begin{array}{lll} 1. \quad i > j > k > l, & 3. \quad i > k > l > j, & 4. \quad i > k > j > l. \end{array} \tag{182}$$

Therefore, we can split the sum into 3 pieces:

$$\sum_{i\neq j\neq k\neq l} \overline{\cos(E_i + E_j - E_k - E_l)}^E = 8\overline{(S_1(t) + S_2(t) + S_3(t))}^E. \tag{183}$$

The first term $S_1(t)$ given by the combination $(i > j > k > l)$ is:

$$\begin{aligned}
S_1(t) &= \sum_{l=1}^{k-1}\sum_{k=2}^{j-1}\sum_{j=3}^{i-1}\sum_{i=4}^{d} \cos(\Delta_{il} + \Delta_{jk}) \\
&= \sum_{l=1}^{k-1}\sum_{k=2}^{j-1}\sum_{j=3}^{i-1}\sum_{i=4}^{d} \cos(s_l + \cdots + s_{k-1} + 2s_k + \cdots + 2s_{j-1} + s_j + \cdots + s_{i-1}),
\end{aligned} \tag{184}$$

where $\Delta_{il}$ are the gaps between the energy levels. An equal result could have been obtained using $\cos(\Delta_{ik} + \Delta_{jl})$. The cosine reads:

$$\cos(s_l + \cdots + s_{k-1} + 2s_k + \cdots + 2s_{j-1} + s_j + \cdots + s_{i-1}) = \cos(\sum_{\beta=l}^{k-1} s_\beta + \sum_{\gamma=k}^{j-1} 2s_\gamma + \sum_{\beta=j}^{i-1} s_\beta). \tag{185}$$

Recalling the definition of $g_E(t)$ in Eq. (133) and making use of the hypothesis of molecular chaos discussed in Sec. 4.4 for which $s_i$ can be treated independently from each other let us take the ensemble average:

$$\begin{aligned}
\overline{\cos\left(\sum_{\beta=l}^{k-1} s_\beta + \sum_{\gamma=k}^{j-1} 2s_\gamma + \sum_{\beta=j}^{i-1} s_\beta\right)}^E &= 1/2 \prod_\beta \overline{e^{is_\beta}}^E \prod_\gamma \overline{e^{2is_\gamma}}^E \prod_\beta \overline{e^{is_\beta}}^E + \text{c.c} \\
&= 1/2 \prod_{\beta=l}^{k-1} g_E(t) \prod_{\gamma=k}^{j-1} g_E(2t) \prod_{\beta=j}^{i-1} g_E(t) + \text{c.c.}
\end{aligned} \tag{186}$$

The result is:

$$\overline{\cos\left(\sum_{\beta=l}^{k-1} s_\beta + \sum_{\gamma=k}^{j-1} 2s_\gamma + \sum_{\beta=j}^{i-1} s_\beta\right)}^E = \frac{1}{2} g_E(t)^{k+i-l-j} g_E(2t)^{j-k} + \text{c.c.} \tag{187}$$

The term $S_1(t)$ becomes:

$$\overline{S_1(t)}^E = 1/2 \sum_{l=1}^{k-1} \sum_{k=2}^{j-1} \sum_{j=3}^{i-1} \sum_{i=4}^{d} g(t)^{k+i-l-j} g(2t)^{j-k} + c.c.$$ (188)

The second term $S_2(t)$ corresponding to the case $(i > k > l > j)$ read as:

$$S_2(t) = \sum_{j=1}^{l-1} \sum_{l=2}^{k-1} \sum_{k=3}^{i-1} \sum_{i=4}^{d} \cos(\Delta_{ik} - \Delta_{lj}) = \sum_{l=1}^{k-1} \sum_{k=2}^{j-1} \sum_{j=3}^{i-1} \sum_{i=4}^{d} \cos(s_k + \cdots + s_{i-1} - s_j - \cdots - s_{l-1}).$$ (189)

Therefore the cosine can be rewritten as:

$$\cos(E_i + E_j - E_k - E_l) = \cos\left( \sum_{\gamma=k}^{i-1} s_\gamma - \sum_{\beta=j}^{l-1} s_\beta \right).$$ (190)

Following the same procedure as before we obtain:

$$\overline{S_2(t)}^E = \sum_{j=1}^{l-1} \sum_{l=2}^{k-1} \sum_{k=3}^{i-1} \sum_{i=4}^{d} g_E(t)^{(i-k)} g_E(t)^{* \, (l-j)} + c.c.$$ (191)

The third term $S_3(t)$ corresponding to the case $(i > k > j > l)$:

$$S_3(t) = \sum_{l=1}^{j-1} \sum_{j=2}^{k-1} \sum_{k=3}^{i-1} \sum_{i=4}^{d} \cos(\Delta_{ik} + \Delta_{jl}),$$ (192)

then the cosine can be written as:

$$\overline{\cos\left(\Delta_{ik} + \Delta_{jl}\right)}^E = \overline{\cos(s_k + \cdots + s_{i-1} + s_l + \ldots s_{j-1})} = \overline{\cos(\sum_{\beta=k}^{i-1} s_\beta + \sum_{\gamma=l}^{j-1} s_\gamma)}^E$$

$$= = \frac{1}{2} \prod_{\beta=k}^{i-1} g_E(t) \prod_{\gamma=l}^{j-1} g_E(t) + c.c,$$ (193)

that becomes:

$$\overline{\cos(\Delta_{ik} + \Delta_{jl})} = \frac{1}{2} g_E(t)^{i+j-k-l} + c.c.$$ (194)

This results allows us to write $\overline{S_3(t)}^E$:

$$\overline{S_3(t)}^E = \frac{1}{2} \sum_{l=1}^{j-1} \sum_{j=2}^{k-1} \sum_{k=3}^{i-1} \sum_{i=4}^{d} g_E(t)^{i+j-k-l} + c.c.$$ (195)

The final result is:

$$\sum_{i \neq j \neq k \neq l} \overline{\cos(E_i + E_j - E_k - E_l)}^E = 4 \sum_{l=1}^{k-1} \sum_{k=2}^{j-1} \sum_{j=3}^{i-1} \sum_{i=4}^{d} g_E(t)^{k+i-l-j} g_E(2t)^{j-k}$$ (196)

$$+ \quad 4 \sum_{j=1}^{l-1} \sum_{l=2}^{k-1} \sum_{k=3}^{i-1} \sum_{i=4}^{d} g_E(t)^{(i-k)} g_E(t)^{* \, (l-j)}$$

$$+ \quad 4 \sum_{l=1}^{j-1} \sum_{j=2}^{k-1} \sum_{k=3}^{i-1} \sum_{i=4}^{d} g_E(t)^{i+j-k-l} + c.c.$$ (197)

The last term that we have to study is $2\sum_{i\neq j>k}\cos[(2E_i - E_j - E_k)t]$, as done before for $\sum_{i\neq j\neq k\neq l}\cos(E_i + E_j - E_k - E_l)$, we can divide this sum into three parts: $i > j > k$, $j > k > i$ and $j > i > k$:

$$2\left(\sum_{i>j>k} + \sum_{j>k>i} + \sum_{j>i>k}\right)\cos[(2E_i - E_j - E_k)t] = 2(v_1(t) + v_2(t) + v_3(t)). \tag{198}$$

It can be exploited and we obtain that $v_1(t)$ is:

$$v_1(t) \equiv \sum_{i>j>k}\cos[(2E_i - E_j - E_k)t] = \sum_{i=3}^{d}\sum_{j=2}^{i-1}\sum_{k=1}^{j-1}\cos\left(s_k + \ldots + s_{j-1} + 2s_j + 2s_{i-1}\right). \tag{199}$$

The cosine reads:

$$\cos\left(s_k + \ldots + s_{j-1} + 2s_j + 2s_{i-1}\right) = \cos\left(\sum_{\beta=k}^{j-1}s_\beta + \sum_{\gamma=j}^{i-1}2s_\gamma\right). \tag{200}$$

Since, all the terms in the cosine are independent from each other we have

$$\overline{\cos\left(\sum_{\beta=k}^{j-1}s_\beta + \sum_{\gamma=j}^{i-1}2s_\gamma\right)}^{E} = \frac{1}{2}\prod_{\beta}\overline{e^{is_\beta}}^{E}\prod_{\gamma}\overline{e^{2is_\gamma}}^{E} + c.c = \frac{1}{2}\prod_{\beta}g(t)\prod_{\gamma}g(2t). \tag{201}$$

The result is:

$$\overline{\cos\left(\sum_{\beta=k}^{j-1}s_\beta + \sum_{\gamma=j}^{i-1}2s_\gamma\right)}^{E} = \frac{1}{2}g_E(t)^{j-k}g_E(2t)^{i-j} + c.c, \tag{202}$$

this means that

$$\overline{v_1(t)}^{E} = \sum_{i=3}^{d}\sum_{j=2}^{i-1}\sum_{k=1}^{j-1}\frac{1}{2}g_E(t)^{j-k}g_E(2t)^{i-j} + c.c. \tag{203}$$

For the case $j > k > i$:

$$v_2(t) = \sum_{j>k>i}\cos[(2E_i - E_j - E_k)t] = \sum_{j=3}^{d}\sum_{k=2}^{j-1}\sum_{i=1}^{k-1}\cos[(2E_i - E_j - E_k)t]. \tag{204}$$

Making use of the cosine property we have that

$$v_2(t) = \sum_{j=3}^{d}\sum_{k=2}^{j-1}\sum_{i=1}^{k-1}\cos[(E_j + E_k - 2E_i)t] = \sum_{j=3}^{d}\sum_{k=2}^{j-1}\sum_{i=1}^{k-1}\cos(s_k + \ldots s_{j-1} + 2s_i + \ldots 2s_{k-1}). \tag{205}$$

We can proceed as before and we obtain

$$\overline{v_2(t)}^{E} = \frac{1}{2}\sum_{j=3}^{d}\sum_{k=2}^{j-1}\sum_{i=1}^{k-1}g_E(t)^{j-k}g_E(2t)^{k-i} + c.c. \tag{206}$$

For the case $j > i > k$:

$$v_3(t) = \sum_{j>i>k}\cos[(2E_i - E_j - E_k)t] = \sum_{j=3}^{d}\sum_{i=2}^{j-1}\sum_{k=1}^{i-1}\cos[(E_j - 2E_i + E_k)t]. \tag{207}$$

The cosine can be rewritten as

$$\cos[(E_j - 2E_i + E_k)t] = \cos\left(\sum_{\beta=i}^{j-1} s_\beta - \sum_{\gamma=k}^{i-1} s_\gamma\right). \tag{208}$$

Just as before we obtain:

$$\overline{v_3(t)}^{\mathrm{E}} = \frac{1}{2}\sum_{j=3}^{d}\sum_{i=2}^{j-1}\sum_{k=1}^{i-1} g_E(t)^{j-i} g_E(t)^{* \, i-k} + c.c. \tag{209}$$

The final result reads:

$$
\begin{aligned}
2\sum_{i\neq j>k} \overline{\cos[(2E_i - E_j - E_k)t]}^{\mathrm{E}} &= \sum_{i=3}^{d}\sum_{j=2}^{i-1}\sum_{k=1}^{j-1} g_E(t)^{j-k} g_E(2t)^{i-j} + \sum_{j=3}^{d}\sum_{k=2}^{j-1}\sum_{i=1}^{k-1} g_E(t)^{j-k} g_E(2t)^{k-i} \\
&+ \sum_{j=3}^{d}\sum_{i=2}^{j-1}\sum_{k=1}^{i-1} g_E(t)^{j-i} g_E(t)^{* \, i-k} + c.c.
\end{aligned} \tag{210}
$$

It is possible to combine all the obtained result and obtain the spectral function $c_4(t)$ for a nearest level distribution. We give a plot of the results in Fig. 16.

## B.2 Real part spectral function $c_3(t)$

In this section, we are going to calculate the real part of the spectral function $c_3(t)$, since in the calculations of our probes of chaos only the real part enters. The calculations are similar to the ones made for $c_4(t)$, we start recalling the definition of $c_3(t)$ introduced in Eq. (8).

$$\mathrm{Re}\, c_3(t) = \mathrm{Re}\sum_{ijk} e^{i(2E_i - E_j - E_k)t}, \tag{211}$$

that can be decomposed as:

$$\mathrm{Re}\, c_3(t) = d + \mathrm{Re}\sum_{i\neq j}\left(e^{2i(E_i-E_j)t} + 2e^{i(E_i-E_j)}\right) + \mathrm{Re}\sum_{i\neq j\neq k} e^{i(2E_i-E_j-E_k)t}. \tag{212}$$

### B.2.1 Calculation of $\overline{c_3(t)}^{\mathrm{GUE}}$

The coefficient $\overline{c_3(t)}^{\mathrm{GUE}}$ is defined as the contraction of $\overline{c_4(t)}^{\mathrm{GUE}}$, for this reason in the literature [103,204] the 3-point coefficient can appear labeled by 4, 1, where the 1 labels the contraction. The average of the real part of the coefficient $c_3(t)$ can be written as:

$$\mathrm{Re}\,\overline{c_3(t)}^{\mathrm{GUE}} = \int dE_1\, dE_2\, dE_3\, dE_4\, P_{\mathrm{GUE}}(\{E_m\})\,\mathrm{Re}\, c_3(t). \tag{213}$$

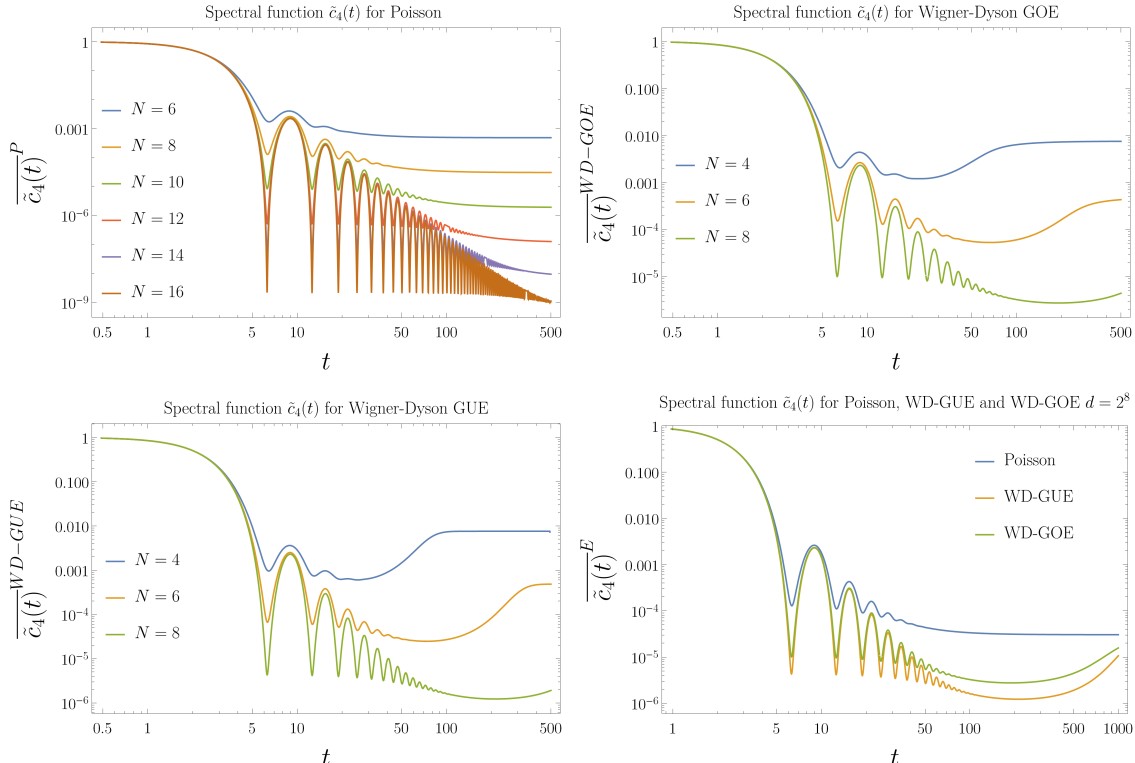

Figure 16: *Top-Left*: Log Log plot of $\tilde{c}_4(t)$ for Poisson and different system size $d = 2^N$, $N = 6, \ldots, 16$. Starting from 1, it reaches the asymptotic value $(2d-1)/d^3$ in a time $O(d^{1/2})$. *Top-Right*: Log Log plot of Wigner-Dyson GOE for different system size $2^N$, $N = 4, 6, 8$. Starting from 1, it goes below the asymptotic value $(2d-1)/d^3$ in a time $O(d^{1/2})$ and reaches the dip $O(d^{-2})$ in a time $O(d^{3/4})$ and the asymptotic value in a time $O(d)$. *Bottom-Right*: Log Log plot of Wigner-Dyson GUE for different system size $2^N$, $N = 4, 6, 8$. Starting from 1, it goes below the asymptotic value $(2d-1)/d^3$ in a time $O(d^{1/2})$ and reaches the dip $O(d^{-2})$ in a time $O(d^{3/4})$ and the asymptotic value in a time $O(d)$. *Bottom-Left*: Comparison between Poisson, Wigner-Dyson from GOE and GUE for $d = 2^8$.

If we decompose $c_3(t)$ as done in Eq. (212) and remember the definition of $\rho^{(n)}(E_1, \ldots, E_n)$ given in Eq. (118)

$$
\begin{aligned}
\overline{\mathrm{Re}\, c_3(t)}^{\mathrm{GUE}} &= d(d-1)(d-2)\, \mathrm{Re} \int \mathrm{d}E_1\, \mathrm{d}E_2\, \mathrm{d}E_3\, \rho^{(4)}_{\mathrm{GUE}}(E_1, E_2, E_3) e^{i(2E_1 - E_2 - E_3)t} \qquad (214) \\
&+ d(d-1) \int \mathrm{d}E_1\, \mathrm{d}E_2\, \rho^{(2)}_{\mathrm{GUE}}(E_1, E_2) e^{i(2E_1 - 2E_2)t} \\
&+ 2d(d-1) \int \mathrm{d}E_1\, \mathrm{d}E_2\, \rho^{(2)}_{\mathrm{GUE}}(E_1, E_2) e^{i(E_1 - E_2)t} + d \qquad (215) \\
&= 2d(d-1)(d-2)\, \mathrm{Re} \int\!\!\int \mathrm{d}E_1\, \mathrm{d}E_2\, \mathrm{d}E_3\, \rho^{(4)}_{\mathrm{GUE}}(E_1, E_2, E_3) e^{i(2E_1 - E_2 - E_3)t} \\
&+ d^2 |r_1(2t)|^2 - d r_2(2t) + 2(d-1)\big(d |r_1(t)|^2 - r_2(t)\big) + d,
\end{aligned}
$$

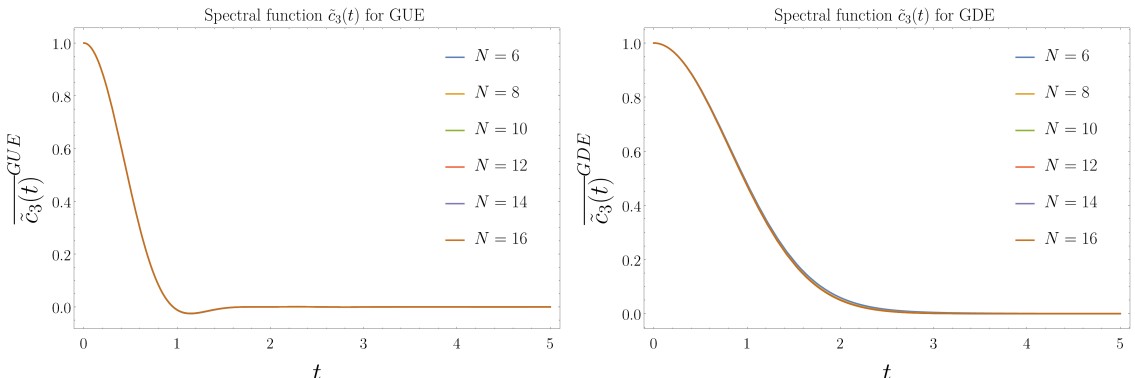

Figure 17: *Left*: Plot of the normalized $\mathrm{Re}\, c_3(t)^{\mathrm{GUE}} = \mathrm{Re}\, c_3(t)/d^3$ averaged over the GUE ensemble for different system size $d = 2^N$ with $N = 6,\ldots,16$. *Right*: Plot of the rescaled spectral function $\tilde{c}_3(t) = c_3(t)/d^3$ averaged over the GDE ensemble for different system size $d = 2^N$ with $N = 6,\ldots,16$.

the integrals are the same of the $\overline{c_4(t)}^{\mathrm{GUE}}$. The final result is:

$$
\begin{aligned}
\overline{\mathrm{Re}\, c_3(t)}^{\mathrm{GUE}} =\ & d^3 r_1(2t) r_1^2(t) - d^2 r_1(2t) r_2(t) r_3(2t) - 2d^2 r_1(t) r_2(2t) r_3(t) \\
& + d^2 r_1^2(2t) + 2d^2 r_1^2(t) + 2d r_2(3t) - d r_2(2t) - 2d r_2(t) + d.
\end{aligned}
\tag{216}
$$

A plot of the $\mathrm{Re}\, c_3(t)^{\mathrm{GUE}}$ for the GUE ensemble can be found in Fig. 17(*Left*).

### B.2.2 Calculation of $\overline{c_3(t)}^{\mathrm{GDE}}$

In order to calculate $\overline{c_3(t)}^{\mathrm{GDE}}$, we first decompose as in Eq. (212), after we take the average over the GDE distribution Eq. (13) and making use of the calculations made in Eq. (177), we obtain:

$$
\overline{\mathrm{Re}\, c_3(t)}^{\mathrm{GDE}} = d + d(d-1)e^{-t^2} + 2d(d-1)e^{-t^2/4} + d(d-1)(d-2)e^{-3t^2/4}.
\tag{217}
$$

The asymptotic value is:

$$
\lim_{t\to\infty} \overline{\mathrm{Re}\, c_3(t)}^{\mathrm{GDE}} = d.
\tag{218}
$$

A plot of the spectral function $\overline{c_3(t)}^{\mathrm{GDE}}$ can be found in Fig. 17(*Right*).

### B.2.3 Calculation of $\overline{c_3(t)}^{\mathrm{E}}$

In this section we calculate $\overline{c_3(t)}^{\mathrm{E}}$. As done before we decompose with Eq. 212 and the average of $\mathrm{Re}\, c_3(t)$ is given by:

$$
\overline{\mathrm{Re}\, c_3(t)}^{\mathrm{E}} = d + \overline{\mathrm{Re}\sum_{i\neq j} e^{2i(E_i - E_j)t}}^{\mathrm{E}} + \overline{\mathrm{Re}\sum_{i\neq j} 2e^{i(E_i - E_j)}}^{\mathrm{E}} + \overline{\mathrm{Re}\sum_{i\neq j\neq k} e^{i(2E_i - E_j - E_k)t}}^{\mathrm{E}}.
\tag{219}
$$

It is possible to recognize that the first two dynamic terms are the ones obtained in and are estimated in Eq. (180), while the last term is calculated in Eq. (210). The results are plotted in Fig. 18: The equilibrium value of $\overline{\mathrm{Re}\, c_3(t)}^{\mathrm{E}}$ is:

$$
\lim_{t\to\infty} \overline{\mathrm{Re}\, c_3(t)}^{\mathrm{E}} = d.
\tag{220}
$$

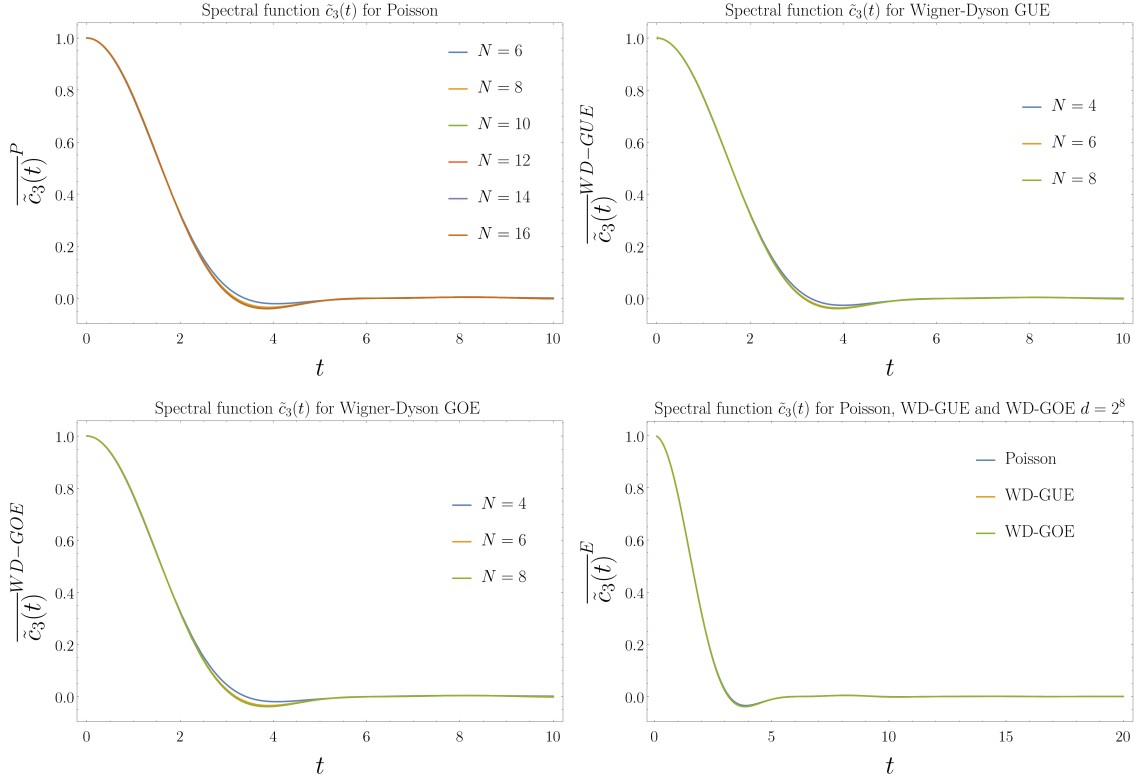

Figure 18: *Top-Left*: Log Log plot of $\tilde{c}_3(t)$ for Poisson and different system size $d = 2^N, N = 4, \ldots, 16$. Starting from 1, it reaches the asymptotic value $1/d$ in a time $O(1)$, after a negative dip reached in a time $O(1)$. *Top-Right*: Log Log plot of Wigner-Dyson GOE for different system size $2^N$, $N = 4, 6, 8$. Starting from 1, it goes below the asymptotic value $1/d$ in a time $O(1)$ and reaches the asymptotic value in a time $O(1)$. *Bottom-Right*: Log Log plot of Wigner-Dyson GUE for different system size $2^N$, $N = 4, 6, 8$. Starting from 1, it reaches the asymptotic value $1/d$ in a time $O(1)$, after a negative dip reached in a time $O(1)$. *Bottom-Left*: Comparison between Poisson, Wigner-Dyson from GOE and GUE for $d = 2^8$.

## C  Auxiliary results for algebra of permutations

The aim of this section is to derive the property introduced in Eq. (114) and (115)

### C.1  Permutations of operators

In order to prove the property of Eq. (114) we have two first to prove the result for a swap operator and then generalize it.

For this simple case, we aim to prove that $T_{(12)}(A \otimes B)T_{(12)} = (B \otimes A)$, where $A, B \in \mathcal{B}(\mathcal{H}^{\otimes 2})$. We have that

$$T_{(12)}(A \otimes B)T_{(12)} = \sum_{ijkl} A_{ij} B_{kl} T_{(12)} |ik\rangle \langle jl| T_{(12)}. \tag{221}$$

The action of $T_{(12)}$ is

$$T_{(12)} |ij\rangle = |ji\rangle. \tag{222}$$

This means that we can rewrite Eq. (221) as:

$$T_{(12)}(A \otimes B)T_{(12)} = \sum_{ijkl} A_{ij} B_{kl} |ki\rangle \langle lj| = B \otimes A. \tag{223}$$

This concludes our proof. This proof can be generalized for a generic permutation $T_\pi \in \mathcal{B}(\mathcal{H}^{\otimes k})$, regarding $T_\pi$ as a product of swaps, between $\mathcal{H}^i \otimes \mathcal{H}^j$, with $i, j \in 1, \dots, k$.

## C.2 Traces of order k

In this section we want to give a proof of Eq. (115). We have to proof the following statement:

$$\operatorname{tr}(A_1 \dots A_k) = \operatorname{tr}\left(T_{(1\ k\ (k-1)\dots 2)}(A_1 \otimes \dots \otimes A_k)\right), \tag{224}$$

where $A_1, \dots, A_k \in \mathcal{B}(\mathcal{H})$. The first term can be written as

$$\operatorname{tr}(A_1 \dots A_k) = \sum_{i_1, \dots, i_k} A_{i_1 i_2} A_{i_2 i_3} \dots A_{i_{k-1} i_k} A_{i_k i_1}. \tag{225}$$

Analyzing the second term, we have

$$\operatorname{tr}\left(T_{(1\ k\ (k-1)\dots 2)}(A_1 \otimes \dots \otimes A_k)\right) = \sum_{\substack{i_1, \dots, i_k \\ , j_1 \dots j_k}} A_{i_1 j_1} \dots A_{i_k j_k} \operatorname{tr}\left(T_{(1\ k\ (k-1)\dots 2)} |i_1 \dots i_k\rangle \langle j_1 \dots j_k|\right). \tag{226}$$

The action of $T_{(1\ k\ (k-1)\dots 2)}$ on the product state is

$$T_{(1\ k\ (k-1)\dots 2)} |i_1 \dots i_k\rangle = |i_2 \dots i_k i_1\rangle. \tag{227}$$

It is possible to rewrite Eq. (226) as

$$
\begin{aligned}
\operatorname{tr}\left(T_{(1\ k\ (k-1)\dots 2)}(A_1 \otimes \dots \otimes A_k)\right) &= \sum_{i_1, \dots, i_k, j_1 \dots j_k} A_{i_1 j_1} \dots A_{i_k j_k} \langle j_1 \dots j_k | i_2 \dots i_k i_1\rangle \\
&= \sum_{\substack{i_1, \dots, i_k, \\ j_1 \dots j_k}} A_{i_1 j_1} \dots A_{i_k j_k} \delta_{j_1 i_2} \dots \delta_{j_{k-1} i_k} \delta_{j_k i_1} \qquad (228) \\
&= \sum_{i_1, \dots, i_k} A_{i_1 i_2} A_{i_2 i_3} \dots A_{i_{k-1} i_k} A_{i_k i_1}. \qquad (229)
\end{aligned}
$$

This concludes our proof.

# D Frame potential

## D.1 Proof of Eq. (25)

The definition of the frame potential is [32, 136]:

$$\mathcal{F}^{(k)}_{\mathcal{E}_H} = \int dG_1 \, dG_2 \left| \operatorname{tr}\left(G_1^\dagger U^\dagger G_1 G_2^\dagger U G_2\right)\right|^{2k}. \tag{230}$$

The frame potential can be rewritten as:

$$\mathcal{F}^{(k)}_{\mathcal{E}_H} = \int dG_1 \, dG_2 \operatorname{tr}\left(G_1^\dagger U G_1 G_2^\dagger U^\dagger G_2\right)^k \operatorname{tr}\left(G_1^\dagger U^\dagger G_1 G_2^\dagger U G_2\right)^k. \tag{231}$$

It is possible to employ the equivalence $(\operatorname{tr} A)^k = \operatorname{tr}\left(A^{\otimes k}\right)$, the frame potential becomes:

$$\mathcal{F}^{(k)}_{\mathcal{E}_H} = \int dG_1 \, dG_2 \operatorname{tr}\left(G_1^{\dagger \otimes 2k} U^{\dagger \otimes k, k} G_1^{\otimes 2k} G_2^{\dagger \otimes 2k} U^{\otimes k, k} G_2^{\otimes 2k}\right), \tag{232}$$

where we have defined $U^{\otimes k, k} \equiv U^{\otimes k} \otimes U^{\dagger \otimes k}$. The trace operator and the integral commute, and employing the definition Eq. (2) we have:

$$\mathcal{F}^{(k)}_{\mathcal{E}_H} = \operatorname{tr}\left(\hat{\mathcal{R}}^{(2k)\dagger}(U)\hat{\mathcal{R}}^{(2k)}(U)\right) = \|\hat{\mathcal{R}}^{(2k)}(U)\|_2^2. \tag{233}$$

## D.2 Proof of the bound Eq. (26)

A proof for the bound Eq. (26) we make use the following bound [116]:

$$\|A\|_p \leq \text{rank}(A)^{\frac{1}{p}-\frac{1}{q}} \|A\|_q.$$

The 2-norm of $2k$-fold Haar channel of a unitary operator can be bounded as:

$$\|\hat{\mathcal{R}}^{(2k)}(U)\|^2 \geq \text{rank}(\hat{\mathcal{R}}^{(2k)}(U))^{(-\frac{1}{2})} \|\hat{\mathcal{R}}^{(2k)}(U)\|_1. \tag{234}$$

The property $\text{rank}(A) = \text{rank}A^\dagger A$ allows us to write $\text{rank}(\hat{\mathcal{R}}^{(2k)}(U)) = d^{2k}$, this means:

$$\|\hat{\mathcal{R}}^{(2k)}(U)\|_2 \geq \frac{\|\hat{\mathcal{R}}^{(2k)}(U)\|_1}{d^k}. \tag{235}$$

In order to proceed we employ the following bound [116]:

$$|\text{tr}(AB)| \leq \|A\|_p \|B\|_q, \tag{236}$$

where $p^{-1} + q^{-1} = 1$, if we set $B \equiv \mathbb{1}$, $p = 1$ and $q = \infty$, we obtain $|\text{tr}A| \leq \|A\|_1$, from which follows:

$$\|\hat{\mathcal{R}}^{(2k)}(U)\|_2 \geq \frac{\left|\text{tr}(\hat{\mathcal{R}}^{(2k)}(U))\right|}{d^k}. \tag{237}$$

The obtained bound, for the frame potential becomes:

$$\mathcal{F}_{\mathcal{E}_H}^{(k)} \geq \frac{\left|\text{tr}(\hat{\mathcal{R}}^{(2k)}(U))\right|^2}{d^{2k}} = \frac{|\text{tr}(U)|^{4k}}{d^{2k}}. \tag{238}$$

This concludes the proof.

## D.3 Proof of Eq. (30)

From the bound Eq. (26) we have that:

$$\mathcal{F}_{\mathcal{E}_H}^{(k)} \geq \frac{|\text{tr}(U)|^{4k}}{d^{2k}}, \tag{239}$$

using the decomposition Eq. (149) proven for the Proposition in Sec. 4.5, we can write the above expression as:

$$\mathcal{F}_{\mathcal{E}_H}^{(k)} \geq f_\pi(U) + C_d \sum_b e^{i\mathbf{C}_b \cdot \mathbf{E}t}, \tag{240}$$

for some $\mathbf{C}_a$s. Taking the spectral average of the above expression with respect to the GDE ensemble $P(\mathbf{E})$, one has, see Sec. 4.5:

$$\overline{\mathcal{F}_{\mathcal{E}_H}^{(k)}}^{\text{GDE}} \geq f_\pi(U) + C_d \hat{P}(\mathbf{C}t), \tag{241}$$

the spectral distribution for the GDE ensemble factorizes in Gaussians (cfr. Eq. (13)); the Fourier transform of a Gaussian is a Gaussian in the conjugate space and therefore we have $\hat{P}(\mathbf{C}t) \geq 0 \; \forall t \geq 0$. This concludes the proof.

# E  OTOC and Loschmidt-Echo

## E.1  4-point OTOC

The definition 4-point OTOC is:

$$\text{OTOC}_4(t) = d^{-1}\text{tr}\left(A^\dagger(t)B^\dagger A(t)B\right), \tag{242}$$

with $A, B \in \mathcal{B}(\mathcal{H})$ two local operators. The introduced definition can be written as:

$$\text{OTOC}_4(t) = d^{-1}\text{tr}\left(U^\dagger A^\dagger U B^\dagger U^\dagger A U B\right) = d^{-1}\text{tr}\left(A^\dagger U B^\dagger U^\dagger A U B U^\dagger\right). \tag{243}$$

The product can be linearized thanks to the replica trick C.2 as:

$$\text{OTOC}_4(t) = d^{-1}\text{tr}\left(T_{(1432)}\left(A^\dagger \otimes B^\dagger \otimes A \otimes B\right)\left(U \otimes U^\dagger\right)^{\otimes 2}\right). \tag{244}$$

This can be rewritten as:

$$\text{OTOC}_4(t) = d^{-1}\text{tr}\left(T_{(1432)}T_{(23)}T_{(23)}\left(A^\dagger \otimes B^\dagger \otimes A \otimes B\right)T_{(23)}T_{(23)}\left(U \otimes U^\dagger\right)^{\otimes 2}T_{(23)}T_{(23)}\right), \tag{245}$$

where we used that $T_{23}T_{23} = \mathbb{1}$. Thanks to the property Eq. (114) it is possible to write:

$$\text{OTOC}_4(t) = d^{-1}\text{tr}\left(T_{(23)}T_{(1432)}T_{(23)}\left(A^\dagger \otimes A \otimes B^\dagger \otimes B\right)U^{\otimes 2,2}\right). \tag{246}$$

The adjoint action of $T_{(23)}$ on $T_{(1432)}$ is equal to

$$T_{(23)}T_{(1432)}T_{(23)} = T_{(1423)}, \tag{247}$$

substituting and making the Isospectral twirling we obtain:

$$\left\langle\text{OTOC}_4(t)\right\rangle_G = d^{-1}\text{tr}(T_{(1423)}\mathscr{A} \otimes \mathscr{B}\hat{\mathcal{R}}^{(4)}(U)), \tag{248}$$

where $\mathscr{A} = A^\dagger \otimes A$, and $\mathscr{B} = B^\dagger \otimes B$.

## E.2  Loschmidt-Echo of second kind

The definition of Loschmidt-Echo of the second kind is:

$$\mathcal{L}(t) = d^{-2}|\text{tr}\left(e^{iHt}e^{-i(H+\delta H)t}\right)|^2, \tag{249}$$

where we introduced a small perturbation $\delta H$ of the Hamiltonian $H$. If $\text{Sp}(H) = \text{Sp}(H+\delta H)$, we can write:

$$H + \delta H = A^\dagger H A, \quad A \in \mathcal{U}(\mathcal{H}), \tag{250}$$

with $A$ a local unitary operator close to the identity. Eq. (250) and the unitarity of $A$ help us to rewrite the LE as:

$$\mathcal{L}(t) = d^{-2}|\text{tr}\left(e^{iHt}e^{-iA^\dagger HAt}\right)|^2 = d^{-2}|\text{tr}\left(e^{iHt}A^\dagger e^{-iHt}A\right)|^2. \tag{251}$$

For the Loschmidt-Echo writing $U_G := G^\dagger U G$:

$$\mathcal{L}(t) = d^{-2}|\text{tr}\left(U_G^\dagger A^\dagger U_G A\right)|^2 = d^{-2}\text{tr}\left(U_G A U_G^\dagger A^\dagger\right)\text{tr}\left(U_G A^\dagger U_G^\dagger A\right) = d^{-2}\text{tr}\left(U_G^{\otimes 2}(A \otimes A^\dagger)U_G^{\dagger \otimes 2}(A^\dagger \otimes A)\right), \tag{252}$$

where we used the cyclic property of the trace and that $\text{tr}(A)\text{tr}(B) = \text{tr}(A \otimes B)$. Now, we use the fact that $\text{tr}(T_{(13)(24)}A^{\otimes 2} \otimes B^{\otimes 2}) = \text{tr}(A^{\otimes 2}B^{\otimes 2})$ in order to write it as:

$$\mathcal{L}(t) = d^{-2}\text{tr}\left(T_{(13)(24)}(U_G^{\otimes 2,2})A \otimes A^\dagger \otimes A^\dagger \otimes A\right). \tag{253}$$

In order to get $\mathscr{A}\otimes\mathscr{A} = A^\dagger\otimes A\otimes A^\dagger\otimes A$, we act adjointly with $T_{(12)}$, noting that $[T_{(12)}, (U_G^{\otimes 2,2})] = 0$:

$$\mathcal{L}(t) = d^{-2}\mathrm{tr}\,(T_{(12)}T_{(13)(24)}T_{(12)}(U_G^{\otimes 2,2})A^\dagger\otimes A\otimes A^\dagger\otimes A) = d^{-2}\mathrm{tr}\,(T_{(14)(23)}(U_G^{\otimes 2,2})\mathscr{A}^{\otimes 2}). \quad (254)$$

Taking the average over $G$

$$\langle\mathcal{L}(t)\rangle_G = d^{-2}\mathrm{tr}\,(T_{(14)(23)}\hat{\mathcal{R}}^{(4)}(U)\mathscr{A}^{\otimes 2}), \quad (255)$$

we obtained the desired result.

### E.3  Estimations for 4-point OTOC and second kind of Loschmidt-Echo

Let us now consider $A, B$ to be non-overlapping Pauli operators on qubits and let us make this calculation explicit. For Pauli operators on qubits, the OTOC and the LE read:

$$\langle\mathrm{OTOC}_4(t)\rangle_G = d^{-1}\mathrm{tr}\,(T_{(1423)}\hat{\mathcal{R}}^{(4)}(U)B^{\otimes 2}\otimes A^{\otimes 2}) \quad (256)$$

$$\langle\mathcal{L}(t)\rangle_G = d^{-2}\mathrm{tr}\,(T_{(14)(23)}(\hat{\mathcal{R}}^{(4)}(U))A^{\otimes 4}). \quad (257)$$

In this settings, we have $\mathrm{tr}\,(A) = \mathrm{tr}\,(B) = \mathrm{tr}\,(AB) = 0$, $\mathrm{tr}\,(ABAB) = d$ and $\mathrm{tr}\,(A^2) = \mathrm{tr}\,(B^2) = d$. Computing the traces for both the quantities the results read:

$$\langle\mathrm{OTOC}_4(t)\rangle_G = \frac{d(c_4(t) - 4c_2(t) + c_2(2t) - d^2 + 9) - 6\,\mathrm{Re}\,c_3(t)}{d(d^4 - 10d^2 + 9)}, \quad (258)$$

$$\langle\mathcal{L}(t)\rangle_G = \frac{(d^2 - 6)(c_4(t) - 4c_2(t) + c_2(2t)) - 2d\,\mathrm{Re}\,c_3(t) + d^4 - 9d^2}{(d^2(d^4 - 10d^2 + 9))}, \quad (259)$$

from the equation above, the large $d$ expressions: Eq. (32) and Eq. (39) follow.

## F  Entanglement and Correlations

### F.1  Entanglement

Consider the unitary time evolution of a state $\psi\in\mathcal{B}(\mathcal{H}_A\otimes\mathcal{H}_B)$ by $\psi\mapsto\psi_t\equiv U\psi U^\dagger$. The entanglement of $\psi_t$ in the given bipartition computed by the 2-Rényi entropy is given by

$$S_2 = -\log\mathrm{tr}\,(\psi_A(t)^2) = -\log\mathrm{tr}\,\left(T_A\otimes\mathbb{1}_B^{\otimes 2}U^{\otimes 2}\psi^{\otimes 2}U^{\dagger\otimes 2}\right), \quad (260)$$

where we used the identity $\mathrm{tr}\,(A^2) = \mathrm{tr}\,(TA\otimes A)$. In order to simplify the notation we introduced the the operator $T_{(A)}$ defined as $T_{(A)}\equiv T_A\otimes\mathbb{1}_B^{\otimes 2}$. This means that the entanglement can be written as

$$S_2 = -\log\mathrm{tr}\,\left(U^{\otimes 2}\psi^{\otimes 2}U^{\dagger\otimes 2}T_{(A)}\right). \quad (261)$$

As usual, we want to linearize the product in the trace, in order to do this we set $a = U^{\otimes 2}\psi^{\otimes 2}$, while $b = U^{\dagger\otimes 2}T_{(A)}$ and we use $\mathrm{tr}\,(ab) = \mathrm{tr}\,\left(T_{(13)(24)}a\otimes b\right)$ and we obtain

$$S_2 = -\log\mathrm{tr}\,\left(T_{(13)(24)}U^{\otimes 2,2}\left(\psi^{\otimes 2}\otimes T_{(A)}\right)\right). \quad (262)$$

We can map $U\mapsto G^\dagger UG$ and average over $G$. The result is

$$\begin{aligned}\langle S_2\rangle_G &= \langle -\log\mathrm{tr}\,\left(T_{(13)(24)}U^{\otimes 2,2}\left(\psi^{\otimes 2}\otimes T_{(A)}\right)\right)\rangle_G \\ &\geq -\log\mathrm{tr}\,\left(T_{(13)(24)}\hat{\mathcal{R}}^{(4)}(U)\left(\psi^{\otimes 2}\otimes T_{(A)}\right)\right),\end{aligned} \quad (263)$$

where The lower bound in Eq. (263) is obtained thanks to the Jensen inequality and we used the equivalence $\left\langle U^{\otimes 2,2}\right\rangle_G = \hat{\mathcal{R}}^4(U)$. Using the explicit form of $\hat{\mathcal{R}}^{(4)}$, see Eq.( 165), we find

$$
\begin{aligned}
\langle S_2 \rangle_G \geq & -\log\Big\{ \frac{1}{d^2(d^6 - 14d^4 + 49d^2 - 36)} \times \\
& \times \big[(d^2-4)\big(\text{tr}(\psi_A^2)\big((c_4(t)-4c_2(t)+c_2(2t))(d^2-3)-4\,\text{Re}\,c_3(t)d\big) \\
& + \text{tr}(\psi_B^2)\big(2\big(4c_2(t)-c_2(2t)-c_4(t)\big)+2\,\text{Re}\,c_3(t)\big(d^2-3\big)\big)\big) \\
& + \text{tr}(\psi^2)\big((c_4(t)+c_2(2t)-4c_2(t))(d_A d - 3d_B) \\
& + 2\,\text{Re}\,c_3(t)(dd_B - 3d_A) - d_A d^5 + d^4 d_B + 9(d_A d^3 - d^2 d_B)\big) \\
& + \big((c_4(t)+c_2(2t)-4c_2(t))(d_B d - 3d_A) \\
& + 2\,\text{Re}\,c_3(t)(dd_A - 3d_B) - d^5 d_B + d_A d^4 + 9d^3 d_B - 9d_A d^2\big)\big]\Big\}.
\end{aligned}
\tag{264}
$$

If our state $\psi$ is pure we obtain

$$
\begin{aligned}
\langle S_2 \rangle_G \geq & -\log\Big\{ \frac{1}{d^2(d+3)(d+1)(d-1)} \times \\
& \times \big[(d+1)\text{tr}(\psi_A^2)\big(c_4(t)-4c_2(t)+c_2(2t)+2\,\text{Re}\,c_3(t)\big) \\
& + (4c_2(t)-c_2(2t)-2\,\text{Re}\,c_3(t)-3d^2+2d^3+d^4-c_4(t))(d_A+d_B)\big]\Big\}.
\end{aligned}
\tag{265}
$$

A plot of the *r.h.s* of Eq. (265) can be found in Fig. 6. The large $d$ expressions are:

$$
\langle S_2 \rangle_G \geq -\log\left[\frac{1}{d_A} + \tilde{c}_4(t)\left(\text{tr}(\psi_A^2) - \frac{1}{d_A}\right)\right] + O(1/d), \quad d_A = O(1), d_B = O(d) \tag{266}
$$

$$
\langle S_2 \rangle_G \geq -\log\left[\frac{2}{\sqrt{d}} + \tilde{c}_4(t)\left(\text{tr}(\psi_A^2) - \frac{2}{\sqrt{d}}\right)\right] + O(1/d), \quad d_A = d_B = \sqrt{d}. \tag{267}
$$

The equilibrium value for the case $d_A \ll d_B$ is

$$
\lim_{t\to\infty} \overline{\langle S_2 \rangle_G}^{\text{E}}(t \to \infty) = \log d_A + O(1/d), \tag{268}
$$

while in the case $d_A = d_B = \sqrt{d}$ is

$$
\lim_{t\to\infty} \overline{\langle S_2 \rangle_G}^{\text{E}} = \frac{1}{2}\log d - \log 2 + O(1/d). \tag{269}
$$

## F.2 Tripartite mutual Information

In this section we are going to review the concept of Tripartite mutual information. In Sec. F.2.1 we introduce the Choi state, an isomorphism between $\mathcal{H}$ and $\mathcal{H}^{\otimes 2}$. In Sec. F.2.2 we introduce the tripartite mutual information [72, 96]. In Sec. F.2.3 we review the 2-Renyi tripartite mutual information and we report some proposition already presented in [1] for completeness.

### F.2.1 Definition of the Choi state

Introduced a unitary operator $U \in \mathcal{U}(\mathcal{H})$, it can be decomposed in a basis $\{|i\rangle\}$, so we can write:

$$
U = \sum_{i,j} u_{ij} |i\rangle \langle j|. \tag{270}
$$

It is possible to map isomorphically an operator $\mathcal{O} \in \mathcal{B}(\mathcal{H})$ into a state $|\mathcal{O}\rangle \in \mathcal{H}^{\otimes 2}$, the isomorphism between an operator and a state is called *Choi isomorphism*. If we apply the map to the unitary operator $U$ introduced above, we obtain the normalized state:

$$|U\rangle = \frac{1}{\sqrt{d}} \sum_{ij} u_{ji} |i\rangle_1 \otimes |j\rangle_2, \tag{271}$$

where we labeled the states with 1(2) the two copies $\mathcal{H}^{\otimes 2}$. The state $|U\rangle$ can be rewritten as:

$$|U\rangle = (\mathbb{1}_1 \otimes U_2)|I\rangle, \tag{272}$$

where $|I\rangle$ is the Bell-state on $\mathcal{H}^{\otimes 2}$ defined as

$$|I\rangle = \frac{1}{\sqrt{d}} \sum_i |i\rangle_1 \otimes |i\rangle_2. \tag{273}$$

In what follows we use the density matrix associated with this state:

$$\rho = |U\rangle\langle U| = (\mathbb{1}_1 \otimes U_2)|I\rangle\langle I|(\mathbb{1}_1 \otimes U_2^\dagger). \tag{274}$$

It is interesting to point out a property of the Choi state if we trace the input or the
    An important property is that if one traces out the input or the output, the resulting state be maximally mixed.

$$\rho_1 = \mathrm{tr}_2(\rho_U) \propto \mathbb{1}_1, \quad \rho_2 = \mathrm{tr}_1(\rho_U) \propto \mathbb{1}_2. \tag{275}$$

The above property can be generalized for any trace preserving unital $CP$-map $f$ acting on the output $\rho_f = \mathbb{1} \otimes f(|I\rangle\langle I|)$, one can prove that:

$$\begin{aligned} \mathrm{tr}_1(\rho_f) &\propto \mathbb{1}_2, \\ \mathrm{tr}_2(\rho_f) &\propto \mathbb{1}_1, \end{aligned} \tag{276}$$

where the labels *in* and *out* labels the two copies of $\mathcal{H}$. The proof is first developed for input. The *r.h.s* of Eq. (276), can be written explicitly written as:

$$\mathrm{tr}_1(\rho_f) = d^{-1} \sum_{ij} \mathrm{tr}(|i\rangle\langle j|_1) \otimes f(|i\rangle\langle j|_2) = d^{-1} \sum_i f(|i\rangle\langle i|_2) = d^{-1} f(\mathbb{1}_2), \tag{277}$$

since $f$ is unital the proof for the input is concluded. We have now to prove the second statement, the one involving the trace of the output:

$$\mathrm{tr}_2(\rho_f) = d^{-1} \sum_{ij} |i\rangle\langle j|_1 \otimes \mathrm{tr}(f(|i\rangle\langle j|_2)) = d^{-1} \sum_{ij} |i\rangle\langle j|_1 \otimes \mathrm{tr}(|i\rangle\langle j|_2) = \frac{\mathbb{1}}{d}, \tag{278}$$

where the second equality follows from the fact that $f$ is trace preserving. The last equation concludes the proof.

### F.2.2 Tripartite Mutual Information

This section aims to provide a brief introduction on the tripartite mutual information, as pointed out in [72] and then in [135], the tripartite information can be considered a measure of scrambling. The simplest model available is to consider a unitary time evolution $U_{AB \to CD}$ defined on two fixed bipartitions $A, B$ the input bipartition and $C, D$ the output bipartition, with

the constraints that $d_A d_B = d_C d_D$. In this set-up, the definition of tripartite mutual information is:

$$I_3(A:C:D) := I(A:C) + I(A:D) - I(A:CD), \tag{279}$$

in terms of Von Neumann entropy the TMI reads, [72,96]:

$$I_3 = S(C) + S(D) - S(AC) - S(AD). \tag{280}$$

It is possible to use the property for the sum of entropies to obtain

$$I_3 = S(C \otimes D) - S(AC) - S(AD). \tag{281}$$

It is possible to prove from the above definition and properties that $S(C \otimes D)$ is equal to $\log d$ for the Choi state and therefore:

$$I_3 = \log d - S(AC) - S(AD). \tag{282}$$

This quantity given is a constant minus a sum of two entropies. All these entropies are calculated, from the Choi state $\rho_U$ associated to $U$. Now, we condensate all the results proven in [135] that we need in the following. First, the tripartite mutual information has the following bounds:

$$-2\log d_A \leq I_3 \leq 0, \tag{283}$$

so it's a negative-definite quantity. The more negative is, the more scrambled the information will be. The upper bound is achieved, by all and only the unitaries made as:

$$U_{A_L \to C_L} \otimes U_{A_R \to D_L} \otimes U_{B_L \to C_R} \otimes U_{B_R \to D_R}, \tag{284}$$

where $A_L, A_R, \ldots, D_L, D_R$ are sub-partition of $A, \ldots, D$ respectively, see [135].

### F.2.3 2-Renyi TMI

Since it is not easy to work with the Von Neumann entropy, we focus our attention on the 2-Renyi entropy, as previously done in [72]:

$$S^{(2)}(\rho) = -\log \operatorname{tr} \rho^2. \tag{285}$$

The choice of the two Renyi entropy as a substitute of the Von-Neumann entropy gives us an upper bound for the mutual information. Let's avoid the subscript $\rho_U$ and write $I_{3_{(2)}}$ as

$$I_{3_{(2)}} = \log d + \log \operatorname{tr} \rho_{AC}^2 + \log \operatorname{tr} \rho_{AD}^2, \tag{286}$$

where $\rho_{AC} = \operatorname{tr}_{BD} \rho$, while $\rho_{AD} = \operatorname{tr}_{BC} \rho$, we have:

$$I_3 \leq I_{3_{(2)}}. \tag{287}$$

From one hand, indeed, the quantity we are going to deal with is not exactly the TMI, but from the other hand, if one is interested in unitary evolutions that scramble the information, the 2-Renyi TMI (2RTMI) is sufficient.
Moreover, it's easy to convince ourselves that the TMI and the 2RTMI share the same bounds:

$$-2\log d_A \leq I_{3_{(2)}} \leq 0. \tag{288}$$

**Proposition** The unitaries of the type $U_C \otimes U_D$, where $U_C \in \mathcal{U}(\mathcal{H}_C)$, $U_D \in \mathcal{U}(\mathcal{H}_D)$ achieve the upper bound of the 2RTMI and therefore, according to this measure of scrambling, do not scramble the information.

**Proof** Firstly, rewrite Eq. (286) as

$$I_{3_{(2)}} = \log d + \log\left(\operatorname{tr}\rho^{\otimes 2}T_{(A)}T_{(C)}\right) + \log\left(\operatorname{tr}\rho^{\otimes 2}T_{(A)}T_{(D)}\right), \tag{289}$$

where $T_{(A)} \equiv T_A \otimes \mathbb{1}_{BCD}$ and $T_A$ a swap operator with support on $\mathcal{H}_A$ similarly for $T_{(C)}$ and $T_{(D)}$. Now, we calculate the scrambling give by:

$$\rho = (\mathbb{1} \otimes U_C \otimes U_D)\,|I\rangle\,\langle I|\,(\mathbb{1} \otimes U_C \otimes U_D)^\dagger \tag{290}$$

and insert it in Eq. (289) and we obtain

$$
\begin{aligned}
I_{3_{(2)}} = \log d \quad &+ \quad \log\!\Big(\operatorname{tr}\big((\mathbb{1}_{AB} \otimes U_C \otimes U_D)\,|I\rangle\,\langle I|\,(\mathbb{1}_{AB} \otimes U_C^\dagger \otimes U_D^\dagger)\big)^{\otimes 2}\,T_{(A)}T_{(C)}\Big) \\
&+ \quad \log\!\Big(\operatorname{tr}\big((\mathbb{1}_{AB} \otimes U_C \otimes U_D)\,|I\rangle\,\langle I|\,(\mathbb{1}_{AB} \otimes U_C^\dagger \otimes U_D^\dagger)\big)^{\otimes 2}\,T_{(A)}T_{(D)}\Big).
\end{aligned}
$$

Now making use of the cyclic property of the trace

$$
\begin{aligned}
I_{3_{(2)}} = \log d \quad &+ \quad \log\!\big(\operatorname{tr}\,|I\rangle\,\langle I|\,T_{(A)}(U_C^\dagger \otimes U_D^\dagger)^{\otimes 2}T_{(C)}(U_C \otimes U_D)^{\otimes 2}\big) \\
&+ \quad \log\!\big(\operatorname{tr}\,|I\rangle\,\langle I|\,T_{(A)}(U_C^\dagger \otimes U_D^\dagger)^{\otimes 2}T_{(D)}(U_C \otimes U_D)^{\otimes 2}\big). \tag{291}
\end{aligned}
$$

It is possible to observe that for the second term $U_D^{\otimes 2}$ does not act on $T_{(C)}$, so it cancels with $U_D^{\dagger\,\otimes 2}$, similarly for the third term but with $U_C^{\otimes 2}$

$$I_{3_{(2)}} = \log d + \log\!\big(\operatorname{tr}\,|I\rangle\,\langle I|\,T_{(A)}(U_C^\dagger)^{\otimes 2}T_{(C)}(U_C)^{\otimes 2}\big) + \log\!\big(\operatorname{tr}\,|I\rangle\,\langle I|\,T_{(A)}(U_D^\dagger)^{\otimes 2}T_{(D)}(U_D)^{\otimes 2}\big). \tag{292}$$

Since $\left[U_C^{\otimes 2}, T_{(C)}\right] = 0$, and $\left[U_D^{\otimes 2}, T_{(D)}\right] = 0$, we obtain

$$I_{3_{(2)}} = \log d + \log\!\big(\operatorname{tr}\,|I\rangle\,\langle I|^{\otimes 2}\,T_{(A)}T_{(C)}\big) + \log\!\big(\operatorname{tr}\,|I\rangle\,\langle I|^{\otimes 2}\,T_{(A)}T_{(D)}\big), \tag{293}$$

at this point a straightforward calculation shows that $I_{3_{(2)}}(U_C \otimes U_D) = 0$. This is not true if we employ the Von Neumann entropy as proved in [135], as explained in the first proposition.

**Proof of Eq.** (50)    Starting from Eq. (286) we can rewrite it as Eq. (289):

$$I_{3_{(2)}} = \log d + \log\operatorname{tr}\left(\rho^{\otimes 2}T_{(A)}T_{(C)}\right) + \log\operatorname{tr}\left(\rho^{\otimes 2}T_{(A)}T_{(D)}\right), \tag{294}$$

where $\rho = (\mathbb{1}_{AB} \otimes U_{CD})\,|I\rangle\,\langle I|\,(\mathbb{1}_{AB} \otimes U_{CD}^\dagger)$. Let's focus on the second term alone, the third one is similar:

$$\log\operatorname{tr}\left(\rho^{\otimes 2}T_{(A)}T_{(C)}\right) = \log\operatorname{tr}\left((\mathbb{1}_{AB} \otimes U_{CD})^{\otimes 2}\,|I\rangle\,\langle I|^{\otimes 2}\,(\mathbb{1}_{AB} \otimes U_{CD}^\dagger)^{\otimes 2}T_{(A)}T_{(C)}\right). \tag{295}$$

In order to proceed with the proof we need two facts. First of all note that $|I\rangle\,\langle I| = \psi_{AC} \otimes \psi_{BD}$ where $\psi_{AC}, \psi_{BD}$ are Bell states. Then, note that the following identity holds: let $T$ be the swap operator and let $\psi \in \mathcal{H}$ a pure state:

$$T\psi^{\otimes 2} = \psi^{\otimes 2}. \tag{296}$$

Now turn again to Eq. (295): first we use the above identity

$$T_{AC}^{(2)}\psi_{AC}^{\otimes 2} = \psi_{AC}^{\otimes 2} \implies T_{(A)}\psi_{AC}^{\otimes 2} = T_{(C)}\psi_{AC}^{\otimes 2}, \tag{297}$$

so we obtain:

$$\log\operatorname{tr}\left((\mathbb{1}_{AB} \otimes U_{CD})^{\otimes 2}\psi_{AC}^{\otimes 2} \otimes \psi_{BD}^{\otimes 2}T_{(C)}(\mathbb{1}_{AB} \otimes U_{CD}^\dagger)^{\otimes 2}T_{(C)}\right), \tag{298}$$

then we can trace out $\mathcal{H}_A \otimes \mathcal{H}_B$ and Eq. (295) becomes:

$$\log \text{tr}\left(\rho^{\otimes 2} T_{(A)} T_{(C)}\right) = \frac{1}{d^2} \log \text{tr}\left(U_{CD}^{\otimes 2} T_{(C)} U_{CD}^{\dagger \otimes 2} T_{(C)}\right), \tag{299}$$

where the factor $d^{-2}$ comes from the trace on the Bell state:

$$\text{tr}_A \psi_{AC}^{\otimes 2} = \frac{\mathbb{1}_C^{\otimes 2}}{d_C^2}, \quad \text{tr}_B \psi_{BD}^{\otimes 2} = \frac{\mathbb{1}_C^{\otimes 2}}{d_D^2}, \tag{300}$$

the proof repeats identically for the second term Eq. (294). This concludes the proof.

**Isospectral twirling of 2RTMI** If we write the complete form of $\hat{\mathcal{R}}^{(4)}(t)$ in Eq. (51), the complete expression of $\left\langle I_{3_{(2)}} \right\rangle_G$ reads:

$$
\begin{aligned}
\left\langle I_{3_{(2)}} \right\rangle_G &= \log_2(d) - 2\log_2\left(d^3\left(d^4 - 10d^2 + 9\right)\right) + \log_2(d\,c_4(t) + 6\,\text{Re}(c_3)(t) \tag{301} \\
&+ d(c_2(2t) - 4c_2(t)))\left(d^2 - d_C^2 - d_D^2 + 1\right) + d^3\left((d_C^2 + d_D^2 - 2)\right)\left(d^2 - 9\right) \\
&+ \log_2\left((12c_2(t) - 3c_2(2t) + 2d\,\text{Re}(c_3)(t) - 3c_4(t)) + d^2\left(2 - d_C^2 + d_D^2\right)\left(d^2 - 9\right)\right).
\end{aligned}
$$

# G  Coherence and Wigner-Yanase-Dyson skew information

## G.1  Coherence

Define $U_G = G^\dagger U G$. The coherence in $B$ of a state evolved by $U_G$ can be written as

$$
\begin{aligned}
\langle \mathcal{C}(\psi_U) \rangle_G &= 1 - \text{tr}\left[(D_B \psi_{U_G})^2\right] = 1 - \text{tr}\left[T_2 (D_B \psi_{U_G})^{\otimes 2}\right] \\
&= \sum_{ij} \text{tr}\left[T_2 \Pi_i \otimes \Pi_j \left(U_G \psi U_G^\dagger\right)^{\otimes 2} \Pi_i \otimes \Pi_j\right] \\
&= 1 - \sum_{ij} \text{tr}\left[\Pi_i \otimes \Pi_j T_2 \Pi_i \otimes \Pi_j \left(U_G \psi U_G^\dagger\right)^{\otimes 2}\right] \\
&= 1 - \sum_i \text{tr}\left[\Pi_i \otimes \Pi_i \left(U_G \psi U_G^\dagger\right)^{\otimes 2}\right] \\
&= 1 - \sum_i \text{tr}\left[T_{(13)(24)} \Pi_i \otimes \Pi_i U_G^{\otimes 2} \otimes \psi^{\otimes 2} U_G^{\dagger \otimes 2}\right] \\
&= 1 - \sum_i \text{tr}\left[T_{(13)(24)} (\Pi_i \otimes \Pi_i \otimes \psi^{\otimes 2})(U_G^{\otimes 2} \otimes U_G^{\dagger \otimes 2})\right], \tag{302}
\end{aligned}
$$

where we used the trick $\text{tr}\,ab = \text{tr}\,(Ta \otimes b)$ calling $a = \Pi_i \otimes \Pi_i U_G^{\otimes 2}, b = \psi^{\otimes 2} U_G^{\dagger \otimes 2}$. Now, taking the average over $G$ by twirling we obtain

$$\langle \mathcal{C}_B(\psi_U) \rangle_G = 1 - \overline{\text{tr}\left[(D_B \psi_{U_G})^2\right]}^G = 1 - \sum_i \text{tr}\left[T_{(13)(24)} (\Pi_i \otimes \Pi_i \otimes \psi^{\otimes 2})\hat{\mathcal{R}}^{(4)}\right]. \tag{303}$$

Using the explicit form of $\hat{\mathcal{R}}^{(4)}$, see Eq.( 165), we find

$$
\begin{aligned}
\langle \mathcal{C}_B(\psi_U) \rangle_G &= \frac{1}{d^2(d^2 - 1)(d + 3)}\left[d^2(3 + d)(d - 1)^2 + 2\left(c_4(t) + 2\,\text{Re}\,c_3(t) + c_2(2t) - 8c_2(t)\right)\right. \\
&+ \left.(d + 1)\text{tr}\,(D_B \psi)^2\left(4c_2(t) - c_2(2t) - 2\,\text{Re}\,c_3(t) + c_4(t)\right)\right]. \tag{304}
\end{aligned}
$$

The expression of $\langle \mathcal{C}_B(\psi_U) \rangle_G$ for large $d$ becomes:

$$\langle \mathcal{C}_{B(\psi_U)} \rangle_G = 1 - \tilde{c}_4(t) \mathrm{tr}\left[(D_B \psi)^2)\right] - \frac{2}{d}\left(1 + (\tilde{c}_3(t) - \tilde{c}_4(t)) \mathrm{tr}\left[(D_B \psi)^2)\right]\right) + O(1/d^2). \quad (305)$$

It is interesting to observe that $\mathrm{tr}\left[(D_B \psi)^2)\right]$ is the purity of the dephased initial state and for $t = 0$ we have the coherence in the basis $B$ of the initial state. The equilibrium value is in the large $d$ limit:

$$\langle \mathcal{C}_B(\psi_U) \rangle_G = 1 - \frac{2}{d} + O(1/d^2). \quad (306)$$

Since we are interested to see how the evolution affects the coherence, we choose $\psi$ as one of the rank-one projector of $B$, Eq. (304) becomes

$$\langle \mathcal{C}_B(\psi_U) \rangle_G = \frac{d^2(d+3)(d-1) + 4c_2(t) - c_2(2t) - 2\,\mathrm{Re}\,c_3(t) - c_4(t)}{d^2(d+1)(d+3)}. \quad (307)$$

The result is plotted in Fig. 8 in the main text.

## G.2 Wigner-Yanase-Dyson skew information

We use the expression

$$\rho_t^\beta = (U\rho U^\dagger)^\beta = e^{\beta \log(U\rho U^\dagger)} = e^{\beta U \log(\rho) U^\dagger} = U e^{\beta \log(\rho)} U^\dagger = U\rho^\beta U^\dagger, \quad (308)$$

in the formula for the Wigner-Yanase-Dyson skew information. We have then

$$\mathcal{I}_\eta(\rho, X) = \mathrm{tr}(X^2 U\rho U^\dagger) - \mathrm{tr}(XU\rho^{1-\eta}U^\dagger X U\rho^\eta U^\dagger). \quad (309)$$

We now write

$$\begin{aligned}
\mathcal{I}_\eta(\rho, X) &= \mathrm{tr}(X^2 U\rho U^\dagger) - \mathrm{tr}(XU\rho^{1-\eta}U^\dagger X U\rho^\eta U^\dagger) &(310)\\
&= \mathrm{tr}\left(T_{(12)}(X^2 \otimes \rho)(U^{\otimes 1,1})\right) - \mathrm{tr}\left(T_{(1423)}(X^{\otimes 2} \otimes \rho^{1-\eta} \otimes \rho^\eta)(U^{\otimes 2,2})\right).
\end{aligned}$$

Taking the Isospectral twirling one gets:

$$\left\langle \mathcal{I}_\eta(\rho, X) \right\rangle_G = \mathrm{tr}\left(T_{(12)}(X^2 \otimes \rho)\hat{\mathcal{R}}^{(2)}(U)\right) - \mathrm{tr}\left(T_{(1423)}(X^{\otimes 2} \otimes \rho^{1-\eta} \otimes \rho^\eta)\hat{\mathcal{R}}^{(4)}(U)\right). \quad (311)$$

We now replace the expressions for $\hat{\mathcal{R}}^{(2)}(U)$ and $\hat{\mathcal{R}}^{(4)}(U)$, writing $\left\langle \mathcal{I}_\eta(\rho, X) \right\rangle_G = (\left\langle \mathcal{I}_\eta(\rho, X) \right\rangle_G)_a - (\left\langle \mathcal{I}_\eta(\rho, X) \right\rangle_G)_b$.

The first term in Eq. (311) is given by

$$(\left\langle \mathcal{I}_\eta(\rho, X) \right\rangle_G)_a = \frac{c_2(t) - 1}{d^2 - 1}\mathrm{tr}(\rho X^2) + \frac{d^2 - c_2(t)}{(d^2 - 1)d}\mathrm{tr}(X^2). \quad (312)$$

Let us call $Z = \rho^{1-\eta} \otimes \rho^{\eta} \otimes X \otimes X$. Then, following the swap operator algebra, we obtain

$$
(\mathcal{I}_\eta(\rho, X))_b = \frac{1}{d^2(d^6 - 14d^4 + 49d^2 - 36)} \times
$$
$$
\times \Big( \operatorname{tr} X^2 \big( (d^6 - 11d^4 + 18d^2 + (c_4(t) + c_2(2t))(d^2 + 6) - 10d \operatorname{Re} c_3(t) )
$$
$$
+ \operatorname{tr} \rho^{1-\eta} \operatorname{tr} \rho^{\eta} \big( 4d c_2(t)(d^2 - 4) - 5(c_2(2t) + c_4(t)) + \operatorname{Re} c_3(t)(2d^2 + 12) - d^3(d^2 - 9) \big) \big)
$$
$$
+ \operatorname{tr} X \operatorname{tr} X \rho \big( c_2(t) 2d(d^2 - 1)(d^2 - 4) - 2d(d^2 + 1)(c_4(t) + c_2(2t)) + \operatorname{Re} c_3(t) - 2d^3(d^2 - 9) \big)
$$
$$
+ \big( \operatorname{tr} X \operatorname{tr} \rho^{1-\eta} \operatorname{tr} X \rho^{\eta} + \operatorname{tr} X \operatorname{tr} \rho^{\eta} \operatorname{tr} X \rho^{1-\eta} \big) \big( 2(c_2(2t) + c_4(t))(2d^2 - 3)
$$
$$
- 2c_2(t)(d^2 - 4)(d^2 - 3) - 2d \operatorname{Re} c_3(t)(d^2 - 1) + 2d^2(d^2 - 9) \big) \tag{313}
$$
$$
+ \operatorname{tr} X \rho^{\eta} X \rho^{1-\eta} \big( c_4(t) + c_2(2t))(d^4 - 7d^2 + 12) - 4d \operatorname{Re} c_3(t)(d^2 - 4)
$$
$$
- 4c_2(t)(d^2 - 4)(d^2 - 3) + d^2(d^2 - 9) \big)
$$
$$
+ \operatorname{tr} X \rho^{\eta} \operatorname{tr} X \rho^{1-\eta} \big( 8c_2(t)d(d^2 - 4) - 2d(c_4(t) + c_2(2t))(d^2 - 4) + 2 \operatorname{Re} c_3(t)(2d^4 - 14d^2 + 24) \big)
$$
$$
+ \big( \operatorname{tr} X^2 \big( 4d c_2(t)(d^2 - 4) - 5(c_2(2t) + c_4(t)) + 2 \operatorname{Re} c_3(t)(d^2 + 6) - d^3(d^2 + 9) \big)
$$
$$
+ \operatorname{tr} \rho^{\eta} \operatorname{tr} \rho^{1-\eta} \big( (c_4(t) + c_2(2t))(d^2 + 6) - 2c_2(t)(d^2 - 4)(d^2 - 3) + d^2(d^4 - 11d^2 + 18) \big) \big)
$$
$$
+ \operatorname{tr} X^2 \rho \big( 4(c_4(t) + c_2(2t))(2d^2 - 3) - 4c_2(t)(d^2 - 4)(d^2 - 3) - 4d \operatorname{Re} c_3(t)(d^2 - 1) + 4d^2(d^2 - 9) \big)
$$
$$
+ \big( \operatorname{tr} X^2 \rho^{\eta} \operatorname{tr} \rho^{1-\eta} + \operatorname{tr} X^2 \rho^{1-\eta} \operatorname{tr} \rho^{\eta} \big) \big( c_2(t)d(d^2 - 4)(d^2 - 1) - (c_2(2t)
$$
$$
+ c_4(t))d(d^2 + 1) + 4 \operatorname{Re} c_3(t)(d^2 - 3) - d^3(d^2 - 9) \big) \Big),
$$

where now we can replace the values of $c_2, c_3$ and $c_4$.

### G.3 Wigner-Yanase-Dyson skew information as a measure of quantum coherence

As it has been claimed in the main text the Wigner-Yanase-Dyson skew information can be set to be a measure of quantum coherence, as in Eq. (60). The Isospectral twirling can be calculated easily from the previous formula. Setting $X = \Pi_i$, we have $\operatorname{tr}(\Pi_i) = 1, \Pi_i^2 = \Pi_i$. Then taking the sum over $i$, we exploit the decomposition of the identity, namely $\sum_i \Pi_i = \mathbb{1}$.

$$
= 1 - \frac{1}{d^2(d^2-1)(d+2)(d+3)} \Big( \big( d^2(d+3)(d-1) + 4c_2(t) - c_2(2t) - 2 \operatorname{Re} c_3(t) - c_4(t) \big)
$$
$$
\times (1 + \operatorname{tr}(\sqrt{\rho})^2) + \operatorname{tr}(D_B(\sqrt{\rho})^2) \big( (d+1) \big( c_4(t) + c_2(2t) - 4c_2(t) + 2 \operatorname{Re} c_3(t) \big) \big) \Big) \tag{314}
$$

The asymptotic behavior:

$$
\langle C(\rho) \rangle_G = 1 - \tilde{c}_4(t) \operatorname{tr}(D_B(\sqrt{\rho})^2) + \frac{(1 - \tilde{c}_4(t))(1 + \operatorname{tr}(\sqrt{\rho})^2) + 2 \big( \operatorname{Re} \tilde{c}_3 - \tilde{c}_4(t) \big) \operatorname{tr}(D_B(\sqrt{\rho})^2)}{d}. \tag{315}
$$

If we make the hypothesis that $\rho$ is pure we obtain

$$
\langle C(\rho) \rangle_G = 1 - \tilde{c}_4(t) \operatorname{tr} \big[ (D_B \psi)^2 \big] - \frac{2}{d} \big( 1 + (\tilde{c}_3(t) - \tilde{c}_4(t)) \operatorname{tr} \big[ (D_B \psi)^2 \big] \big) + O(1/d^2). \tag{316}
$$

It is possible to point out that for $\rho$ pure the asymptotic behavior of $\langle C(\rho) \rangle_G$ is equal to the asymptotic expansion for $\langle C_{B(\psi_U)} \rangle_G$ given in Eq. (57).

## H  Quantum Thermodynamics

### H.1  Quantum Batteries in Closed system

In this section we develop the calculations relative to the work in Sec. H.1.1 and to the fluctuations of the work in Sec. H.1.2.

### H.1.1 Work Average

The Isospectral twirling of the work operator can be obtained considering the average over the spectrum $K$ of the work operator introduced in Eq. (71). It is possible to use the identity $\text{tr}(TA \otimes B) = \text{tr}(AB)$ where $T$ is the swap operator, and the work can be rewritten as:

$$W(t) = \text{tr}(\psi H_{0)} - \text{tr}(\psi K H_0 K^\dagger) = \text{tr}(\psi H_0) - \text{tr}(T(\psi \otimes H_0)K^\dagger \otimes K), \tag{317}$$

to get the correct order of Eq. (2) we insert $\mathbb{1} = T^2$, and we use the swap property Eq. (114), the result is:

$$W(t) = \text{tr}(\psi H_0) - \text{tr}(T K \otimes K^\dagger (\psi \otimes H_0)). \tag{318}$$

We can take the Haar average average $\hat{\mathcal{R}}^{(2)}(K)$ (see Eq. (106)) and obtain:

$$\langle W(t) \rangle_G = \text{tr}(\psi H_0) - \text{tr}[T(\psi \otimes H_0)\hat{\mathcal{R}}^{(2)}(K)], \tag{319}$$

where we exploit the fact that $\left[\hat{\mathcal{R}}^{(2)}, T\right] = 0$. It is possible to replace $\hat{\mathcal{R}}^{(2)}(K) = \frac{c_2(t)-1}{d^2-1}\mathbb{1} + \frac{d^2-c_2(t)}{d(d^2-1)}T$ and one obtains

$$\langle W(t) \rangle_G = \text{tr}(\psi H_0) - \text{tr}\left[T(\psi \otimes H_0)(\frac{c_2(t)-1}{d^2-1}\mathbb{1} + \frac{d^2-c_2(t)}{d(d^2-1)}T)\right] \tag{320}$$

$$= (1 - \frac{c_2(t)-1}{d^2-1})\text{tr}(\psi H_0) - \frac{d^2-c_2(t)}{(d^2-1)}\frac{\text{tr}\,H_0}{d}. \tag{321}$$

Let us define as $E_0 \equiv \text{tr}(\psi H_0)$ the expectation value of $H_0$ on $\psi$ and $E_{HT} \equiv \frac{\text{tr}(H_0)}{d}$ the expectation value of $H_0$ on the infinite temperature state $\frac{\mathbb{1}}{d}$. Then

$$\langle W(t) \rangle_G = \frac{d^2}{d^2-1}(E_0 - E_{HT}) - \frac{c_2(t)}{d^2-1}(E_0 - E_{HT})$$

$$= \frac{d^2-c_2(t)}{d^2-1}(E_0 - E_{HT}) = \frac{d^2-c_2(t)}{d^2-1}\delta W_0, \tag{322}$$

where we define $E_0 - E_{HT} \equiv \delta W_0$. Now since $c_2(t) = d^2$ at $t = 0$. We have

$$\langle W(t=0) \rangle_G = 0. \tag{323}$$

First, we discuss the envelopes for the case of Poisson, GUE and GDE. The normalized work $\frac{W(t)}{\delta W_0}$ is given in Fig. 10

We are interested to know how it is possible to approximate the work for the different ensembles. For Poisson, we use the asymptotic expansion Eq. (139) so, it results:

$$W(t) \approx \delta W_0(1 - \frac{2}{d^2 t^2}). \tag{324}$$

In the case of GDE we have

$$\overline{c_2(t)}^{\text{GDE}} = d + d(d-1)e^{-t^2/4}. \tag{325}$$

In this case, we have

$$\langle W(t) \rangle_G = \delta W_0(\frac{d}{d+1} - \frac{d(d+1)}{d^2-1}e^{-\frac{t^4}{4}}) = d\delta W_0(\frac{1}{d+1} - \frac{e^{-\frac{t^4}{4}}}{d-1}), \tag{326}$$

which for $d \gg 1$ we can write again as

$$\langle W(t) \rangle_G = \delta W_0(1 - e^{-\frac{t^4}{4}}). \tag{327}$$

In the case of GUE, we recall Eq. (122) from that we can obtain that for $d \gg 1$ we have

$$W(t) = \delta W_0 \begin{cases} 1 - \frac{J_2(2t)}{t} - \frac{1}{d} & t > 2d \\ \frac{1}{d}(\frac{t}{2} - 1) - \frac{J_2(2t)}{t} & t < 2d \end{cases}. \tag{328}$$

### H.1.2 Work fluctuations

In this section, we aim to develop the calculations for the work fluctuations. We start from the definition of fluctuations

$$\left\langle \Delta W^2 \right\rangle_G := \left\langle W^2(t) \right\rangle_G - \left\langle W(t) \right\rangle_G^2. \tag{329}$$

If we analyze the term $W(t)^2$ and make use of the property Eq. (115) and we can write:

$$W(t)^2 = \mathrm{tr}\,(H_0\psi)^2 - 2\mathrm{tr}\,(H_0\psi)\mathrm{tr}\,(T\,(K\otimes K^\dagger)(\psi\otimes H_0)) + \mathrm{tr}\,(T_{(12)(34)}(K\otimes K^\dagger)^{\otimes 2}.(\psi\otimes H_0)^{\otimes 2}). \tag{330}$$

To see the correct order, let us insert $T_{(23)}^2 = \mathbb{1}$ and make use of the property shown in Eq. (114):

$$W(t)^2 = E_0^2 - 2E_0\mathrm{tr}\,(T_2(K\otimes K^\dagger)(\psi\otimes H_0)) + \mathrm{tr}\,(T_{(13)(24)}(K^{\otimes 2}\otimes K^{\dagger\,\otimes 2})(\psi^{\otimes 2}\otimes H_0^{\otimes 2})). \tag{331}$$

Now we take the Isospectral twirling of $(K^{\otimes 2}\otimes K^{\dagger\,\otimes 2})$:

$$\langle W\rangle_G^2 = E_0^2 - 2E_0\mathrm{tr}\,(T_2\hat{\mathcal{R}}^2(K)(\psi\otimes H_0)) + \mathrm{tr}\,(T_{(13)(24)}\hat{\mathcal{R}}^4(K)((\psi^{\otimes 2}\otimes H_0^{\otimes 2})). \tag{332}$$

The fluctuations of the work are given by

$$\Delta W^2(t) = \left\langle W^2(t) \right\rangle_G - \left\langle W(t) \right\rangle_G^2. \tag{333}$$

Making use of Eq. (319) we obtain

$$\left\langle \Delta W^2(t) \right\rangle_G = \mathrm{tr}\,\big(T_{(13)(24)}\hat{\mathcal{R}}^4(K)(\psi^{\otimes 2}\otimes H_0^{\otimes 2})\big) - \mathrm{tr}\,\big(T_{(12)(34)}(\hat{\mathcal{R}}^2(K))^{\otimes 2}(\psi\otimes H_0)^{\otimes 2}\big). \tag{334}$$

Inserting $\hat{\mathcal{R}}^4(K)$ given in Eq. (166), we obtain:

$$\begin{aligned}
\left\langle \Delta W^2(t) \right\rangle_G =&\ \frac{1}{d^2(d^2-9)(d^2-4)(d^2-1)^2}\times\\
&\times\big(d^2(d^2-9)\big((d^2-1)\big(4\mathrm{tr}\,H_0^2\psi + 4\mathrm{tr}\,H_0\mathrm{tr}\,H_0\psi^2 - d\,(\mathrm{tr}\,H_0^2 + (\mathrm{tr}\,H_0)^2\,\mathrm{tr}\,\psi^2) + 2\mathrm{tr}\,H_0^2\psi^2\big)\\
&+ E_0^2(4-d^2) + E_0\mathrm{tr}\,H_0(10d-4d^3) + (2+d^2)(\mathrm{tr}\,H_0)^2 + (2-3d^2+d^4)\mathrm{tr}\,H_0^2\mathrm{tr}\,\psi^2\big)\\
&+2(d^2-1)\mathrm{Re}\,c_3(t)\big[(E_0\mathrm{tr}\,H_0 + \mathrm{tr}\,(H_0^2\psi^2))(8d^2-12) - 2dE_0^2(d^2-4)\\
&+\mathrm{tr}\,H_0\psi H_0\psi(d^4-7d^2+12) + (6+d^2)(\mathrm{tr}\,\psi^2(\mathrm{tr}\,H_0)^2 + \mathrm{tr}\,H_0^2)\\
&-2d(1+d^2)(\mathrm{tr}\,H_0^2\psi + \mathrm{tr}\,H_0\psi^2\mathrm{tr}\,H_0) - 5d((\mathrm{tr}\,H_0)^2 + \mathrm{tr}\,H_0^2\mathrm{tr}\,\psi^2)\,\big]\\
&+c_4(t)\big[E_0^2(d^2-4)(5d^2+3) - E_0\mathrm{tr}\,H_0(26d^3-74d) - 2d(d^4-5d^2+4)\mathrm{tr}\,H_0\psi H_0\psi\\
&-2d(d^4-1)\mathrm{tr}\,(H_0^2\psi^2) + (8d^4-20d^2+12)(\mathrm{tr}\,H_0^2\psi + \mathrm{tr}\,H_0\psi^2\mathrm{tr}\,H_0)\\
&+(18d^2-42)(\mathrm{tr}\,H_0)^2 + (d^4+5d^2-6)\mathrm{tr}\,H_0^2\mathrm{tr}\,\psi^2 - 5d(d^2-1)(\mathrm{tr}\,\psi^2\mathrm{tr}\,H_0^2 + \mathrm{tr}\,(H_0)^2)\big]\\
&+(d^2-1)c_2(2t)\big[E_0^2(d^4-7d^2+12) - 2d\mathrm{tr}\,(H_0\psi H_0\psi)(d^2-4)\\
&-2d(1+d^2)(E_0\mathrm{tr}\,H_0 + \mathrm{tr}\,H_0^2\psi^2) + (8d^2-12)(\mathrm{tr}\,H_0^2\psi\\
&+\mathrm{tr}\,H_0\psi^2\mathrm{tr}\,H_0) - 5d(\mathrm{tr}\,\psi^2\mathrm{tr}\,H_0^2 + \mathrm{tr}\,(H_0)^2) + (6+d^2)(\mathrm{tr}\,H_0^2 + \mathrm{tr}\,H_0^2\mathrm{tr}\,\psi^2)\,\big]\\
&+c_2(t)\big[2(d^2-4)(4d\mathrm{tr}\,(H_0\psi H_0\psi)(d^2-1) + E_0^2(d^4-d^2-6) + 2dE_0\mathrm{tr}\,H_0(d^4-6d^2+5)\\
&+(d-2d^3+d^5)\mathrm{tr}\,(H_0^2\psi^2) - (2d^4+4d^2-6)(\mathrm{tr}\,H_0^2\psi + \mathrm{tr}\,H_0\psi^2\mathrm{tr}\,H_0)\\
&-2d(1+d^2)(\mathrm{tr}\,\psi^2(\mathrm{tr}\,H_0)^2 + \mathrm{tr}\,H_0^2) - (5d^2+3)(\mathrm{tr}\,H_0)^2 - (d^4-4d^2+3)\mathrm{tr}\,H_0^2\mathrm{tr}\,\psi^2)\,\big]\big).
\end{aligned} \tag{335}$$

Making the hypothesis that $\psi$ is a pure state and that it is an eigenstate of $H_0$, the work fluctuations becomes:

$$
\begin{aligned}
\left\langle \Delta W^2(t) \right\rangle_G &= \frac{1}{d^2(d^2-9)(d+2)(d^2-1)^2} \times \\
&\times \ \left( d^2(d^2-9)((d^3+d^2-d-1)\mathrm{tr}(H_0^2) - E_0\mathrm{tr}\,H_0(4d^2+4d-2) \right. \\
&- \ E_0^2(d+2) - 2(d^2-1)\mathrm{tr}\,H_0^2\psi - (d^2+d+1)(\mathrm{tr}(H_0))^2) \\
&+ \ c_4(t)(E_0^2(5d^2+3)(d+2) + E_0\mathrm{tr}\,H_0(8d^2+14d+2)(d-3) \\
&- \ (4d^4-10d^2+6)\mathrm{tr}\,H_0^2\psi - (5d^2-8d-21)(\mathrm{tr}(H_0))^2 \\
&+ \ (d^3-3d^2-d+3)\mathrm{tr}(H_0^2)) + c_2(2t)(d^2-1)(E_0^2(d^3+2d^2-3d-6) \\
&- \ E_0\mathrm{tr}\,H_0(2d^2-4d-6) - (4d^2-6)\mathrm{tr}\,H_0^2\psi + (\mathrm{tr}(H_0)^2+\mathrm{tr}(H_0^2))) \\
&+ \ 2c_3(t)(d^2-1)((d-3)(\mathrm{tr}(H_0)^2+\mathrm{tr}(H_0^2)) - 2E_0^2d(d+2) \\
&- \ E_0\mathrm{tr}\,H_0(2d^2-4d-6) + (d^3+d)\mathrm{tr}\,H_0^2\psi) \\
&+ \ 2c_2(t)(d+2)(E_0^2(d^4+d^2+6) - 2E_0\mathrm{tr}\,H_0(d-3)(d-1)(d+1)^3 \\
&- \ (d-2)(d^2-1)(d^2+3)\mathrm{tr}\,H_0^2\psi - (d-3)(2d^3+d+1)(\mathrm{tr}\,H_0)^2 \\
&+ \ 2(d^2-1)(d+1)(d-3)\mathrm{tr}\,H_0^2)).
\end{aligned}
\tag{336}
$$

For large $d$ the fluctuations are given by

$$
\left\langle \Delta W^2(t) \right\rangle_G = 4E_{HT}E_0\tilde{c}_2(t) + \frac{1}{d}\left( \frac{\mathrm{tr}\,H_0^2}{d} - E_{HT}^2 \right) + O(d^{-2}).
\tag{337}
$$

It is interesting to point out that for $t \longrightarrow \infty$, the ensemble averages of the Isospectral twirling of the work fluctuations for the studied distributions converge to the same value that is

$$
\overline{\left\langle \Delta W^2(t \to \infty) \right\rangle}_G^{\mathrm{E}} = \frac{1}{d}\left( 4E_{HT}E_0 + \frac{\mathrm{tr}\,H_0^2}{d} - E_{HT}^2 \right) + O(d^{-2}).
\tag{338}
$$

Let us define the dimensionless control parameter $h = 4E_{HT}E_0/(\mathrm{tr}(H_0^2)/d - E_{HT}^2)$ and study:

$$
\left\langle \Delta \tilde{W}^2(t) \right\rangle_G = h\tilde{c}_2(t) + \frac{1}{d} + O(d^{-2}),
\tag{339}
$$

where $\left\langle \Delta \tilde{W}^2(t) \right\rangle_G = \left\langle \Delta W^2(t) \right\rangle_G / (\mathrm{tr}(H_0^2)/d - E_{HT}^2)$. In large $d$ limit Eq. (339):

$$
\lim_{t \to \infty} \Delta \tilde{W}^2(t) = \frac{h+1}{d} + O(d^{-2}),
\tag{340}
$$

the results are plotted in Fig. 10 in the main text.

## H.2 Quantum batteries in Open systems

This section will expand the calculations made in Sec. 3.5.3, we begin recalling the definition of free energy operator is Eq. (77)

$$
\hat{\mathcal{F}} := H_A + \beta^{-1}\log\psi_{U,A},
\tag{341}
$$

defined on a Hilbert space $\mathcal{H}_A \otimes \mathcal{H}_B$. The $A$ subsystem represents the system from which one extracts work and the $B$ is the bath of the system with temperature $T = 1/(k_B\beta)$. The reduced evolution on $A$ is not unitary and reads $\psi_{U,A} = \mathrm{tr}_B(U_G\psi U_G^\dagger)$ and is induced by a Hamiltonian of the form $H = H_A + H_B + H_{AB}$. The extractable work is defined as

$$
\begin{aligned}
\mathcal{F} &:= \mathrm{tr}(\hat{\mathcal{F}}\psi_{U,A}) = \mathrm{tr}\left( (\hat{\mathcal{F}} \otimes \mathbb{1}_B)U\psi U^\dagger \right) \tag{342} \\
&= \mathrm{tr}\left( [(H_A + \beta^{-1}\log\psi_{U,A}) \otimes \mathbb{1}_B]U\psi U^\dagger \right), \tag{343}
\end{aligned}
$$

where we have defined $\psi_{U,A} = \text{tr}_B U\psi U^\dagger = \text{tr}_B \psi_U$. Let us manipulate this quantity:

$$\mathcal{F} = \text{tr}\left((H_A \otimes \mathbb{1}_B)U\psi U^\dagger\right) + \beta^{-1}\text{tr}\left[[\log \psi_{U,A} \otimes \mathbb{1}_B]\psi_U\right]. \tag{344}$$

If we use this identity $\text{tr}\left(G(A \otimes 1)\right) = \text{tr}_A(\text{tr}_B(G)A)$ we obtain:

$$\mathcal{F} = \text{tr}\left((H_A \otimes \mathbb{1}_B)U\psi U^\dagger\right) + \beta^{-1}\text{tr}\left(\psi_{U,A}\log \psi_{U,A}\right). \tag{345}$$

It's possible to recognize the Von Neumann entropy in the second term of the *r.h.s.* We can regard it as a Renyi entropy $S_\alpha$, defined as:

$$S_\alpha(\psi) = \frac{1}{1-\alpha}\log \text{tr}\left(\psi^\alpha\right), \tag{346}$$

with $\alpha = 1$ is:

$$\mathcal{F} = \text{tr}\left((H_A \otimes \mathbb{1}_B)U\psi U^\dagger\right) - \beta^{-1}S_1(\psi_{U,A}) \tag{347}$$

and use the hierarchy of Renyi entropies:

$$S_\alpha(\psi) \geq S_{\alpha'}(\psi), \quad \alpha < \alpha' \in \mathbb{N}_0, \tag{348}$$

to bound it:

$$S_2(\psi) \leq S_1(\psi) \leq \log \text{rank}(\psi), \tag{349}$$

therefore the extractable work is:

$$\text{tr}\left((H_A \otimes \mathbb{1}_B)U\psi U^\dagger\right) - \beta^{-1}\log d_A \leq \mathcal{F} \leq \text{tr}\left((H_A \otimes \mathbb{1}_B)U\psi U^\dagger\right) - \beta^{-1}S_2(\psi_{U,A}). \tag{350}$$

Now we can take the Isospectral twirling of this quantity $U \to U_G = G^\dagger UG$, defined in Eq. (2):

$$\text{tr}\left((H_A \otimes \mathbb{1}_B)U\psi U^\dagger\right)_G - \beta^{-1}\log d_A \leq \langle \mathcal{F}\rangle_G \leq \text{tr}\left((H_A \otimes \mathbb{1}_B)U\psi U^\dagger\right)_G - \beta^{-1}S_2(\psi_{U,A})_G. \tag{351}$$

Now taking a look at the result for the entanglement Eq. (44) and following a procedure identical to the one used for the work H.1.2, we find:

$$\text{tr}\left((H_A \otimes \psi)\hat{\mathcal{R}}^{(2)}(U)\right) - \beta^{-1}\log d_A \leq \langle \mathcal{F}\rangle_G \tag{352}$$

$$\langle \mathcal{F}\rangle_G \leq \text{tr}\left((H_A \otimes \psi)\hat{\mathcal{R}}^{(2)}(U)\right) + \beta^{-1}\log \text{tr}\left(T_{(13)(24)}\hat{\mathcal{R}}^{(4)}(U)\psi^{\otimes 2} \otimes T_A \otimes \mathbb{1}_B^{\otimes 2}\right).$$

Recall Eq. (265), Eq. (266) and Eq. (73) and consider the difference between the extractable work for $t = 0$ and in $t$:

$$\frac{d^2 - c_2(t)}{d^2 - 1}\delta W_0 - \beta^{-1}\log d_A \leq \langle \mathcal{F}\rangle_G - \text{tr}(H_0\psi) \tag{353}$$

$$\langle \mathcal{F}\rangle_G - \text{tr}(H_0\psi) \leq \frac{d^2 - c_2(t)}{d^2 - 1}\delta W_0 + \beta^{-1}\log\left[1 + \tilde{c}_4(t)(d_A - 1)\right] - \beta^{-1}\log d_A + O(1/d),$$

where we defined $\delta W_0 \equiv E_0 - \text{tr}(H_0)/d$, that is the difference between the expectations value of the Hamiltonian evaluated on the state $\psi$ and on the infinite temperature state $\mathbb{1}/d$. The bounds for the extractable work in the large $d$ limit can be written as:

$$\frac{d^2 - c_2(t)}{d^2 - 1}\epsilon^{-1}\log d_A \leq \Delta\langle \mathcal{F}\rangle_G \leq \frac{d^2 - c_2(t)}{d^2 - 1} + \epsilon^{-1}\log\left[1 + \tilde{c}_4(t)(d_A - 1)\right] - \epsilon^{-1}\log d_A + O(1/d), \tag{354}$$

where we introduced a dimensionless parameter $\epsilon \equiv \Delta W_0\beta$, and the quantity $\Delta\langle \mathcal{F}\rangle_G \equiv (\langle \mathcal{F}\rangle_G(t) - \langle \mathcal{F}\rangle_G(0))/\delta W_0$. In the large $d$ limit we have:

$$\lim_{t \to \infty}\overline{\Delta\langle \mathcal{F}(t)\rangle_G}^{\text{E}} = 1 - \epsilon^{-1}\log d_A. \tag{355}$$

# I  Isospectral twirling and CP maps

Recall the definition of Isospectral twirling for CP-maps $Q(\cdot) = \sum_\alpha K_\alpha(\cdot)K_\alpha^\dagger$:

$$\hat{\mathcal{R}}^{(2k)}(Q) = \int dG G^{\dagger\otimes 2k} \mathcal{K}^{\otimes k} G^{\otimes 2k}. \tag{356}$$

Let us first give the expression for $k = 1$. From Sec. 4.1, the computation follows straightforwardly:

$$\hat{\mathcal{R}}^{(2)}(Q) = \frac{1}{d(d^2-1)}((d^2 - \operatorname{tr}\mathcal{K})T + d(\operatorname{tr}\mathcal{K} - 1). \tag{357}$$

Now let us prove the expression Eq. (90). Starting from the definition:

$$\mathcal{L}_1(Q) = \operatorname{tr}(\psi Q(\psi)) = \sum_\alpha \operatorname{tr}(\psi K_\alpha \psi K_\alpha^\dagger), \tag{358}$$

and using the trick of the swap operator $T \in \mathcal{B}(\mathcal{H}^{\otimes 2})$ we can write it as:

$$\mathcal{L}_1(Q) = \sum_\alpha \operatorname{tr}(T(K_\alpha \otimes K_\alpha^\dagger)\psi^{\otimes 2}), \tag{359}$$

then, twirling the Kraus operator as $K_\alpha \to G^\dagger K_\alpha G$, taking the Haar average, recalling the definition $\mathcal{K} = \sum_\alpha K_\alpha \otimes K_\alpha^\dagger$:

$$\langle \mathcal{L}_1(Q) \rangle_G = \int d \operatorname{tr}(T G G^{\dagger\otimes 2} \mathcal{K} G^{\otimes 2} \psi^{\otimes 2}) = \operatorname{tr}(T\hat{\mathcal{R}}(Q)\psi^{\otimes 2}). \tag{360}$$

Now, let us prove the expression for the purity, Eq. (93). Starting from the definition:

$$\operatorname{Pur}(Q\psi) = \operatorname{tr}(Q(\psi)^2) = \sum_{\alpha,\beta} \operatorname{tr}(K_\alpha \psi K_\alpha^\dagger K_\beta \psi K_\beta^\dagger) = \sum_{\alpha\beta} \operatorname{tr}(T(K_\alpha \otimes K_\beta)\psi^{\otimes 2}(K_\alpha^\dagger \otimes K_\beta^\dagger)), \tag{361}$$

setting $\psi \in$ pure states, use the property $T\psi^{\otimes 2} = \psi^{\otimes 2}$ and $T^2 = \mathbb{1}$ to write the above expression as:

$$\operatorname{Pur}(Q\psi) = \sum_{\alpha\beta} \operatorname{tr}((K_\beta \otimes K_\alpha)\psi^{\otimes 2}(K_\alpha^\dagger \otimes K_\beta^\dagger)), \tag{362}$$

recall that $\operatorname{tr}((A \otimes B)(C \otimes D)) = \operatorname{tr}(T_{(13)(24)}A \otimes B \otimes C \otimes D)$ and write:

$$\operatorname{Pur}(Q\psi) = \sum_{\alpha\beta} \operatorname{tr}(T_{(13)(24)}(K_\beta \otimes K_\alpha \otimes K_\alpha^\dagger \otimes K_\beta^\dagger)(\psi^{\otimes 2} \otimes \mathbb{1}^{\otimes 2})). \tag{363}$$

In order to have the correct expression in the definition Eq. (85), we need to manipulate it: act adjointly with $T_{(234)}$ inserting $T_{(234)}T_{(243)} = \mathbb{1}$ twice and get:

$$\operatorname{Pur}(Q\psi) = \sum_{\alpha\beta} \operatorname{tr}(T_{(12)(34)}(K_\alpha \otimes K_\alpha^\dagger \otimes K_\beta \otimes K_\beta^\dagger)(\psi \otimes \mathbb{1})^{\otimes 2}). \tag{364}$$

Taking the Isospectral twirling we finally have:

$$\operatorname{Pur}(Q\psi) = \operatorname{tr}(T_{(12)(34)}\hat{\mathcal{R}}^{(4)}(Q)(\psi \otimes \mathbb{1})^{\otimes 2}). \tag{365}$$

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
