# Peer review of "Random Matrix Theory of the Isospectral twirling"

_SciPost Physics, doi:SciPost Phys. 10, 076 (2021)_

## Round 2 · Referee Report · Karol Życzkowski (Referee 1) · 2021-1-11

Strengths

several new results on out-of-time-order correlators, Loschmidt-Echos
and several measures of entanglement
for various assumptions concerning the ensembles of random matrices applied

Weaknesses

Long paper with so many results that it is difficult to appreciate which of them are the most important...

Report

The paper by Oliviero et al presents an analysis if the twirling operation from the point of view of the theory of random matrices. Isospectral twirling is often used in quantum information processing
as it describes the Haar average over multiple usage of a channel.
The paper presents a detailed analysis of
out-of-time-order correlators, Loschmidt-Echos
and several measures of entanglement
for various assumptions concerning the
ensembles of random matrices applied.

The paper is well written and its results are presented clearly.
I do appreciate the Tables 1 and 2 which summarize well results obtained.
As they seem to be correct and new, so my opinion this work can
become suitable for the publication in SciPost.

In general the paper contains a lot of new and interesting results,
but their number seems to be even too large: thus the paper is really
long, difficult to read and appreciate.
To be honest I was not able to check
all the formulae and derivations presented...

Requested changes

My detailed remarks to the paper include:

a) p.1 The authors write:

quantum chaos has traditionally been associated to salient properties of those systems which were quantized from classical > chaotic systems. Such systems were found to possess statistics of energy > levels spacings corresponding to the predictions of Random Matrix Theory (RMT) [3–14]. This is correct, but we know also a lot concerning the statistical properties of eigenvectors of chaotic systems, what could be mentioned here with appropriate references. Another well-developed path of research concerned quantum Lyapunov exponent and quantum dynamical entropies.

b) p. 4, U= exp - iHt - brackets are missing

c) p 9. Figure 1 (with known figure) is somehow misleading - would you consider adding both axes, which cross the origin (0,0) d) this figure (and all othee figures !) - larger Figure labels and axes captions are welcome! So small fonts of the labels make the data presented difficult to understand (and some labels are hardly legible thus the figure becomes not readable)

e) p. 9 The authors discuss nearest level distributions for Poisson, GOE and GUE and recall eq. (13). the sentence

The nearest level distributions for these distributions are given by... sugests that Eq (13) presents exact results for large systems size which is not true - these are only results obtained for d=2 which can serve as reasonably precise approximations for the asymptotic distributions, d--> infty, see the book of Mehta [8]. This remark concerns also eq (123), p. 37.

f) p. 10) >>With the use of Levy lemma... a good reference to the leve lemma would be helpful

g) p.12 Fig.2 presents the spectral functions c_2 and c_4 for Poisson ensemble, GDE and GUE. After looking at Fig 1 a reader might ask, how it looks like for GOE? Are the differencies between GOE and GUE relevant here (and in further quantities discussed later in the paper? The same question concerns also Figs. 3-11).

h) p.13 - the formulation

A t-design is therefore ‘random enough’ up to the t-th moment. seems to me misleading, as the actual design is 'not random' - the points are carefully chosen to be placed 'uniformly' all around the space of all unitary matrices, such that the moments indeed agree with the integral over the Haar measure... Take the Gauss-Legendre approximate integration rule in interval [-1,1] approximated by the value of the function in two points $x_{\pm}=\pm 1/sqrt{3}$ -- see https://en.wikipedia.org/wiki/Gauss%E2%80%93Legendre_quadrature Would you call these numbers 'random'? The reasoning goes in other direction as a set of random numbers (or quasi-random - produced by a chaotic dynamics!) can approximate a $t$-design up to a certain accuracy.

i) p. 20, after eq (44) we learn that

Eq. (44) converges to the equilibrium value in a time O(d^{1/12}) and O(d^{1/8}) for GUE and Poisson, respectively. is it simple to explain the reader where these two exponents came from? (this remark concerns also the caption to Fig 6 and 9.)

j) I am not sure I understand Eq.(59) correctly - perhaps the last exponent should go after the last bracket to have tr (D_B(\sqrt{rho}))^2 as square is taken after the dephasing operator D_B... is it so? k) what is the physical meaning of the operator {\cal K} in eq (84). Is it somehow related to the superoperator (e.g. by partial transpose?) So it is also linked by another change of entries of the matrix to the Choi matrix D of the operation Q? Just after Eq. (84) a statement appears: >> taking T as the swap operator, then tr (TK) = d. is this just a consequence of the trance preserving condition which implies the standard normalization of the Choi matrix, Tr D=d ? If this is so, such a remakr could be helpful for a reader to see where this property comes from...

l) The sentence in p. 62 above Eq. (275) reads

The action of a unital map can be written as f(rho) = \sum_k U_k rho U_k^ (275) Any map in the above form of a mixed unitary channel is unital but the reverse statement is true for d=2 only. For d=3 there exist the channel of Landau-Streater which is unital but is not mixed unitary as (275). Thus it looks like this proof works for d=2 only and for higher dimensions in requires improvements...

m) well known papers of Wigner and Dyson have several parts, which are labeled by Roman numbers say paper I and paper II (capital letters - ref [5] and [7])

n) the authors might consider quoting in Ref. [9] the more recent edition of the book by Haake - revised and considerably extended

o) Ref [22] has title Hidden correlations in the hawking radiation possibly capital letter: Hawking, also Neumann's in [76] and Loschmidt in ref [86]

  • validity: high
  • significance: high
  • originality: high
  • clarity: good
  • formatting: good
  • grammar: excellent

Author:  Salvatore Francesco Emanuele Oliviero  on 2021-03-09  [id 1290]

(in reply to Report 1 by Karol Życzkowski on 2021-01-11)

We are grateful to the referee for his attentive readings and clarification requests. We have implemented some modifications to the text of our manuscript to take them into account.

To improve the clarity of the paper we introduced in the revised manuscript a new section called "Summary and Commentary of the Results" in order to present to the reader which results are the most relevant. In the following, we will answer the remarks of the referee.

a) We thank the referee for the suggestion, we modified the manuscript as follows:
"Such systems were found to possess statistics of energy levels spacings corresponding to the predictions of Random Matrix Theory (RMT)[3-14], as well as statistics of eigenvectors[15-20]."
We have also added these references [26,27,28] to the following sentence:
"After all, we still would not know what quantum chaos means [21–28] "

b) We have modified $\exp{-i H t}$ as $\exp{(-i H t)}$ in the revised manuscript.

c) As suggested by the referee, for clearness we modified the axes in Fig.1 in the revised manuscript.

d) We increased the font size of the figure and axes label in the revised manuscript.

e) As suggested by the referee we edited the sentence mentioned in point e), which now reads:
"Although $P_{\text{WD-GOE}}$ and $P_{\text{WD-GUE}}$ are calculated for $d=2$, they are seen to be a good approximation for $d\rightarrow\infty$[8]."

f) We added the reference [116] in the revised manuscript when we introduce the Levy lemma.

g) We calculated the spectral functions $c_{a}(t)$ for two eigenvalue distributions, GUE and GDE, see Sec. 4.3, and three nearest level spacings distributions: Poisson, Wigner-Dyson from GUE and from GOE, see Sec. 4.4. Then, in the discussion of the various probes to chaos, Sec. 3, we opted to use the eigenvalues distribution (GUE and GDE) and just the Poisson nearest level spacings distribution; the reason why we did not include there Wigner-Dyson from GUE and from GOE is twofold: first, in the calculation for $c_{a}(t)$ we exploited the hypothesis of independent spacings. This leads to an exact result in the case of Poisson distribution while it is an approximation for GOE; second, the behavior of $c_{a}(t)$, for the GOE case is not qualitatively different from the GUE eigenvalue distribution.

h) We removed the sentence "A $t$-design is therefore ‘random enough’ up to the $t$-th moment" because as pointed out by the referee, it comes from a clumsy jargon that can result misleading.

i) All the timescales - even those that not reported in Table 1 - are all calculated from the envelope curves as shown in the relative section 4.4.2. For clearness we edited the sentence which now reads:
" indeed the r.h.s. of Eq. 44 converges to the equilibrium value in a time $O(d^{1/12})$ and $O(d^{1/8})$ for GUE and Poisson respectively; these timescales can be explicitly calculated from the envelope curves of GUE and Poisson for $t=O(d^{1/2})$, see Sec. 4.4.2."

j) We have edited the expression $Tr(D_B\sqrt{\rho}^{2})$, which now reads $Tr((D_B\sqrt{\rho})^{2})$.

k) The operator $\mathcal{K}$ is a generalization for non-unitary quantum maps of the operator $U\otimes U^{\dagger}$; the latter naturally arises when one wants to linearize expectation values of polynomials of $U^{\dagger}X U$. As pointed out by the referee $U\otimes U^{\dagger}$ is related to the Choi isomorphism via the partial transposition on the second copy of $\mathcal{H}$. We have introduced the following two modifications in the manuscript to explain this concept, the first one on page 5 relative to $U^{\otimes k,k}$ reads:
"Above, $\text{d} \, G$ represents the Haar measure over the unitary group $\mathcal U (d)$ and we recall that

$
U^{\otimes k,k}\equiv U^{\otimes k}\otimes U^{\dagger\otimes k}
$

which naturally arises from the linearization of the expectation values of polynomials of $U^{\dagger}XU$."

The second one on page 28 relative to $\mathcal{K}$ reads:
"where we defined

$
\mathcal{K}\equiv\sum_{\alpha}K_{\alpha}\otimes K^{\dagger}_{\alpha}
$

it is a non hermitian bounded operator on $\mathcal{H}^{\otimes 2}$; it is a generalization of $U\otimes U^{\dagger}$ for non-unitary quantum maps."

l) We thank the referee for having pointed out the problem in our proof. The requirement of the trace preserving property is sufficient to conclude the proof. While for tracing the input we need $f$ to be unital, for tracing the output we just use the trace preserving property of $f$. The proof has been modified has follow:
" We have now to prove the second statement, the one involving the trace of the output:

$
\text{tr}_{2}(\rho_f)=d^{-1}\sum_{ij}|i\rangle \langle j|_{1}\otimes \text{tr}(f(|i\rangle \langle j|_{2}))=d^{-1}\sum_{ij}|i\rangle \langle j|_{1}\otimes \text{tr}(|i\rangle \langle j|_{2})=\frac{\mathbb{I}}{d}
$

where the second equality follows from the fact that $f$ is trace preserving. The last equation concludes the proof. "

m) We have added the capital letters in the references [5] and [7] in the revised manuscript.

n) We thank the referee for the suggestion. We have changed the version cited in Ref.[9] in the revised manuscript.

o) We have corrected the references [31],[84],[94] in the revised manuscript.

List of other changes:
We added to Table 2 the average values of the spectral functions $c_{a}(t)$ for a unitary $U$ drawn from the Haar measure.

New References:
$\left[15\right]$ F. Izrailev, "Chaotic structure of eigenfunctions in systems with maximal quantum chaos", Physics Letters A125(5), 250 (1987), doi:10.1016/0375-9601(87)90203-9.
$\left[16\right]$ M. Kus, J. Mostowski and F. Haake, "Universality of eigenvector statistics of kicked tops of different symmetries", Journal of Physics A: Mathematical and General21(22), L1073(1988), doi:10.1088/0305-4470/21/22/006.
$\left[17\right]$ F. Haake and K. Zyczkowski, "Random-matrix theory and eigenmodes of dynamical systems", Physical Review A42(2), 1013 (1990), doi:10.1103/physreva.42.1013.
$\left[18\right]$ K. Zyczkowski, "Indicators of quantum chaos based on eigenvector statistics", Journal of Physics A: Mathematical and General23(20), 4427 (1990), doi:10.1088/0305-4470/23/20/005.
$\left[19\right]$ K. Zyczkowski and G. Lenz, "Eigenvector statistics for the transitions from the orthogonal to the unitary ensemble", Zeitschrift für Physik B Condensed Matter82(2), 299 (1991), doi:10.1007/bf01324340.
$\left[20\right]$ J. Emerson, Y. S. Weinstein, S. Lloyd and D. G. Cory, "Fidelity decay as an efficient indicator of quantum chaos", Phys. Rev. Lett.89, 284102 (2002), doi:10.1103/PhysRevLett.89.284102.
$\left[26\right]$ F. Haake, H. Wiedemann and K. Zyczkowski, "Lyapunov exponents from quantum dynamics", Annalen der Physik504(7), 531 (1992), doi:10.1002/andp.19925040706.
$\left[27\right]$ V. Man’ko and R. Vilela Mendes, "Lyapunov exponent in quantum mechanics. A phase-space approach", Physica D: Nonlinear Phenomena145(3-4), 330–348 (2000),doi:10.1016/s0167-2789(00)00117-2.
$\left[28\right]$ Y. S. Weinstein, S. Lloyd and C. Tsallis, " Border between regular and chaotic quantum dynamics", Phys. Rev. Lett.89, 214101 (2002), doi:10.1103/PhysRevLett.89.214101.
$\left[116\right]$ J. Watrous, "The Theory of Quantum Information", Cambridge University Press, doi:10.1017/9781316848142 (2018).

---

## Round 2 · Referee Report · Ion Nechita (Referee 2) · 2021-3-8

Strengths

1- an approach to unify several "probes" of quantum chaos 2- very thorough analysis and comparison 3- rigorous computations using standard techniques in random matrix theory 4- results presented in a very visual and nice manner, easy to understand

Weaknesses

1- rather standard techniques, no theoretical insight

Report

The paper under review considers the problem of characterizing quantum chaos with the help of a newly introduced quantity, the isospectral twirling (IT) [leone2020isospectral].

The main theme of the paper is the realization that many of the ``probes'' of quantum chaos used in the literature (frame potential, OTOCs, Loschmidt echos, coherence, mutual information, thermodynamical quantities, etc) can be expressed as simple functions of the isospectral twirling operator of a given unitary dynamics. More precisely, given a fixed Hamiltonian $H$, consider the time evolution $U_t = \exp(-\mathrm{i}tH)$ and its isospectral twirling
$$\hat{\mathcal R}^{2k}(U):= \int_G (G^*)^{\otimes 2k} (U^{\otimes k} \otimes {\bar U}^{\otimes k}) G^{\otimes 2k} \mathrm{d}G,$$
where $G$ is a $d \times d$ Haar-distributed random unitary operator. Mainly, the $k=1,2$ values are analyzed, and the computations are performed with the help of the Weingarten calculus. Then, the values of the different quantum chaos probes are computed for random matrices $U$ coming from different ensembles (Poisson, GUE, Gaussian diagonal) and the values are nicely presented in Table 2.

I find that the work contains interesting results pertaining to the characterization of quantum chaos. This is a very active research area, and the efforts towards the unification of the different ``probes'' of chaos presented here are important and deserve publication. However, on the technical side, there is no profound insight, except maybe that most of the probes are polynomials in the matrix entries, which can be linearized by taking tensor powers of the unitary operator. The main computations are standard in Random Matrix Theory. However, the work is very thorough, and the proofs are complete (though the previous, unpublished paper [leone2020isospectral] is relied upon numerous times). My recommendation is: weak accept.

Requested changes

1- There is a large overlap between the results presented in this paper and those of [leone2020isospectral]; maybe the authors could add a paragraph explaining precisely what is new here 2- page 32, definition of the Weingarten function: the fact that the Weingarten function can be computed starting from the trace of the product of two tensor permutation matrices was already present in [Benoı̂t Collins and Piotr Śniady. Integration with respect to the haar measure on unitary, orthogonal and symplectic group. Communications in Mathematical Physics, 264(3):773–795, 2006]. There, the matrix having entries $d^{#(\pi^{-1}\sigma)}$ played an important role. 3- The authors should mention the pioneering work of Werner [Reinhard F Werner. Quantum states with einstein-podolsky-rosen correlations admitting a hidden-variable model. Physical Review A, 40(8):4277, 1989], who introduced the concept of twirling'' to quantum information theory 4- page 20, 2 lines above sec 3.3.3:respectively.,'' 5- page 21, top: ``[64]defined'' 6- page 21, below eq (48): the notation $I(A;B)$ is used, whereas above, in eq (48), the notation $I(A:C)$ is preferred 7- page 21, above eq (49): the notation $\rho_{AC(AD)}$ is confusing, the authors should write separately the two equations 8- page 40, below eq (141): the value $1/\pi^2t^6$ should correspond to the average of $c_4$, not $c_2$ 9- page 40, eq (142): $2/t^2$ instead of $2/d^2$ at the beginning

  • validity: high
  • significance: good
  • originality: ok
  • clarity: high
  • formatting: excellent
  • grammar: excellent

Author:  Salvatore Francesco Emanuele Oliviero  on 2021-03-09  [id 1291]

(in reply to Report 2 by Ion Nechita on 2021-03-08)

We thank the referee for the careful and positive report. Although referee is correct that the random matrix techniques we used are standard, we want to highlight that several results are new, as, to our knowledge, using Poisson spacing to compute the various quantities of this paper, e.g., the spectral coefficients $\overline{c_4(t)}^E$ of section B.1.3 has not been done before, and these results are new. We realize that we should have highlighted the novelties of this paper with greater care, especially with respect to [1], as the referee also suggested. We have implemented all the requested changes and modified some relevant passages in the text.

1) The introduction has been modified in order to highlight the novelties of this paper and its difference with the previous paper of reference [1]. We added the two following passages: "The framework for unification has been introduced in [1], where it was shown that several probes of quantum chaos can be cast in the form of the $2k$-Isospectral twirling (IT). In this paper, we extend these results to obtain a more complete classification." and "While the isospectral twirling has been introduced in [1], here we provide the calculations and techniques to obtain the random matrix theory results for several ensembles of spectra. Moreover, here we provide a concentration bound for the Isospectral twirling, generalize the notion of IT to general quantum channel, show the behavior of quantum coherence in quantum chaotic evolutions, include applications to quantum thermodynamics, and finally prove a general theorem regarding Schwartzian spectra distributions and universality at late times."

2) We added the reference $[201]$, in the revised manuscript.

3) We thank the referee for the suggestion, we added in the introduction the reference $[99]$.

4) , 5) , 6) , 8) ,9) We fixed the typos in the revised manuscript as per referee's suggestion.

7) We thank the referee for the suggestion, the text now reads "More specifically, $\rho_{AC}=tr_{BD}(\rho_U) $ and $\rho_{AD}=tr_{BC}(\rho_U)$ "

New Reference $[99]$ R. F. Werner, "Quantum state with Einstein-Podolsky-Rosen correlations admitting a hidden-variable model", Physical Review A 40,4277 (1989), doi:10.1103/PhysRevA.40.4277

Ion Nechita  on 2021-03-10  [id 1293]

(in reply to Salvatore Francesco Emanuele Oliviero on 2021-03-09 [id 1291])

Dear authors, I would like to address your comment

Although referee is correct that the random matrix techniques we used are standard, we want to highlight that several results are new, as, to our knowledge, using Poisson spacing to compute the various quantities of this paper has not been done before, and these results are new

This is exact, I agree with your point. My remark was about the Weingarten part, not about the integration with respect to the spectra of the different ensembles (which one can say is not a random matrix computation).

---

## Round 3 · Referee Report · Karol Życzkowski · 2021-3-14

Strengths

*

Weaknesses

*

Report

*

Requested changes

*

---

## Round 3 · Referee Report · Ion Nechita · 2021-3-16

Strengths

-

Weaknesses

-

Report

-

Requested changes

-

---

## Editorial Decision

published